# The Price of Privacy for Low-rank Factorization

**Jalaj Upadhyay**
Johns Hopkins University
Baltimore, MD - 21201, USA.
jalaj@jhu.edu

## Abstract

In this paper, we study what price one has to pay to release *differentially private low-rank factorization* of a matrix. We consider various settings that are close to the real world applications of low-rank factorization: (i) the manner in which matrices are updated (row by row or in an arbitrary manner), (ii) whether matrices are distributed or not, and (iii) how the output is produced (once at the end of all updates, also known as *one-shot algorithms* or continually). Even though these settings are well studied without privacy, surprisingly, there are no private algorithm for these settings (except when a matrix is updated row by row). We present the first set of differentially private algorithms for all these settings.

Our algorithms when private matrix is updated in an arbitrary manner promise differential privacy with respect to two stronger privacy guarantees than previously studied, use space and time *comparable* to the non-private algorithm, and achieve *optimal accuracy*. To complement our positive results, we also prove that the space required by our algorithms is optimal up to logarithmic factors. When data matrices are distributed over multiple servers, we give a non-interactive differentially private algorithm with communication cost independent of dimension. In concise, we give algorithms that incur *optimal cost across all parameters of interest*. We also perform experiments to verify that all our algorithms perform well in practice and outperform the best known algorithm until now for large range of parameters.

## 1 Introduction

Low-rank factorization (LRF) of matrices is a fundamental component used in many applications, such as clustering [15, 19, 43], data mining [5], recommendation systems [20], information retrieval [49, 53], learning distributions [2, 34], and web search [1, 36]. In these applications, given an $m \times n$ matrix $\mathbf{A}$, a common approach is to first compute three matrices: a diagonal positive semidefinite matrix $\widetilde{\boldsymbol{\Sigma}}_k \in \mathbb{R}^{k \times k}$ and two matrices, $\widetilde{\mathbf{U}}_k \in \mathbb{R}^{m \times k}$ and $\widetilde{\mathbf{V}}_k \in \mathbb{R}^{n \times k}$, with orthonormal columns. The requirement then is that the product $\mathbf{B} := \widetilde{\mathbf{U}} \widetilde{\boldsymbol{\Sigma}} \widetilde{\mathbf{V}}^{\mathsf{T}}$ is as close to $\mathbf{A}$ as possible. More formally,

**Problem 1.** $(\alpha, \beta, \gamma, k)$-LRF. *Given parameters $0 < \alpha, \beta < 1$, $\gamma$, a matrix $\mathbf{A} \in \mathbb{R}^{m \times n}$ matrix, the target rank $k$, compute a rank-$k$ matrix factorization $\widetilde{\mathbf{U}}_k, \widetilde{\boldsymbol{\Sigma}}_k,$ and $\widetilde{\mathbf{V}}_k$ such that*

$$\mathsf{Pr}\left[ \|\mathbf{A} - \widetilde{\mathbf{U}}_k \widetilde{\boldsymbol{\Sigma}}_k \widetilde{\mathbf{V}}_k^{\mathsf{T}}\|_F \leq (1 + \alpha)\|\mathbf{A} - [\mathbf{A}]_k\|_F + \gamma \right] \geq 1 - \beta,$$

*where $\|\cdot\|_F$ denotes the Frobenius norm, and $[\mathbf{A}]_k$ is the best rank-$k$ approximation of $\mathbf{A}$. We refer to the parameter $\gamma$ as the* additive error *and to $\alpha$ as the* multiplicative error.

Practical matrices are often large, distributed over many servers and are dynamically updated [41, 42] and hence many works have considered these settings in order to reduce latency, synchronization issues and resource overhead [6, 9, 16, 13, 14, 17, 27, 40, 44, 48, 52]. Moreover, these applications use confidential dataset and use of *ad hoc* mechanism can lead to serious privacy leaks [47]. Therefore,

| Updates | Comments | Privacy | Additive Error | Reference |
|---------|----------|---------|----------------|-----------|
| Turnstile | One-shot | $\mathbf{A} - \mathbf{A}' = \mathbf{u}\mathbf{v}^{\mathsf{T}}$ | $\widetilde{O}((\sqrt{mk}\alpha^{-1} + \sqrt{kn})\varepsilon^{-1})$ | Theorem 1 |
| Turnstile | One-shot | $\|\mathbf{A} - \mathbf{A}'\|_F = 1$ | $\widetilde{O}((\sqrt{mk}\alpha^{-2} + \sqrt{kn})\varepsilon^{-1})$ | Theorem 2 |
| Turnstile | Continually | $\mathbf{A} - \mathbf{A}' = \mathbf{u}\mathbf{v}^{\mathsf{T}}$ | $\widetilde{O}((\sqrt{mk}\alpha^{-1} + \sqrt{kn})\varepsilon^{-1}\log T)$ | Theorem 4 |
| Turnstile | Continually | $\|\mathbf{A} - \mathbf{A}'\|_F = 1$ | $\widetilde{O}((\sqrt{mk}\alpha^{-1} + \sqrt{kn})\varepsilon^{-1}\log T)$ | Theorem 4 |
| Row wise | One-shot | $\|\mathbf{A} - \mathbf{A}'\|_F = 1$ | $\widetilde{O}\left((\alpha\varepsilon)^{-1}\sqrt{nk}\right)$ | Corollary 1 |
| – | Local | $\|\mathbf{A} - \mathbf{A}'\|_F = 1$ | $\widetilde{O}\left(k\alpha^{-2}\epsilon^{-1}\sqrt{m}\right)$ | Theorem 5 |

Table 1: Our Results for $(\varepsilon, \Theta(n^{-\log n}))$-Differentially Private Algorithms ($T$: stream length, $m \geq n$).

for any practical deployment [3, 25], one would like to simultaneously maintain strong privacy guarantee and minimize space requirements, communication and computational costs.

Unfortunately, existing private algorithm for LRF do not consider these settings (except in central model when matrices are received row by row [24]). For example, known algorithms either use multiple pass over the data matrix [29, 30, 31, 35] or cannot handle arbitrary updates [24]. Similarly, known algorithms that continually release output are for *monotonic functions* [23], thereby excluding Problem 1. Private algorithms like [24, 30] that can be extended to distributed setting use multiple rounds of interaction, large communication cost, and/or result in trivial error bounds. Moreover, known private algorithms are inefficient compared to non-private algorithms: $O(mnk)$ time and $O(mn)$ space compared to time linear in the sparsity of the matrix and $O((m+n)k/\alpha)$ space [9, 13]. In fact, for rank-$k$ matrices, Musco and Woodruff [45] state that Problem 1 is equivalent to the well studied *matrix completion* for which one can have $\widetilde{O}(n \cdot \mathsf{poly}(k))$ time non private algorithm [32]. Under same assumptions, private algorithm takes $O(mnk)$ time [30]. This motivates the central thesis of this paper: What is the price of privacy for non-trivial private algorithms?

## 1.1 Overview of the Results

We give a unified approach and first set of algorithms for solving Problem 1 in various settings: (i) when private matrix is updated row by row or in arbitrary manner, (ii) when private matrix is distributed or not, and (iii) when the output is produced once at the end of all updates, also known as *one-shot algorithms* or continually. We show that *one does not have to pay the price of privacy* (more than what is required in terms of additive error and space). On a high level, we show the following:

1. When a private matrix is streamed, we propose differentially private algorithms with respect to two stronger privacy guarantees than previously studied. We also show that these algorithms can be extended to continual release model. Our algorithms uses basic linear algebra. This makes them easy to code, and therefore, optimize.

2. We complement our positive results with a matching lower bound on the space required. Our algorithms are also time efficient and achieve *optimal accuracy*.

3. In the distributed setting, we give a non-interactive differentially private algorithm with communication cost independent of dimension.

All our results are summarized in Table 1.

## 2 Preliminaries

In this paper, we give algorithms that are private under the notion of differential privacy. Differential privacy has emerged as a *de facto* notion of privacy over the last few years. Formally, it is defined as follows:

**Definition 1** (($\epsilon, \delta$)-differential privacy). *A randomized algorithm $\mathfrak{M}$ gives $(\varepsilon, \delta)$-differential privacy, if for all neighboring datasets $\mathbf{A}$ and $\mathbf{A}'$, and all measurable sets $S$ in the range of $\mathfrak{M}$, $\Pr[\mathfrak{M}(\mathbf{A}) \in S] \leq \exp(\varepsilon)\Pr[\mathfrak{M}(\mathbf{A}') \in S] + \delta$, where the probability is over the coin tosses of $\mathfrak{M}$.*

We consider two stronger privacy guarantees than previously studied: $\mathsf{Priv}_1$ and $\mathsf{Priv}_2$. In $\mathsf{Priv}_1$, we call two matrices $\mathbf{A}$ and $\mathbf{A}'$ neighboring if $\mathbf{A} - \mathbf{A}' = \mathbf{u}\mathbf{v}^{\mathsf{T}}$ for some unit vectors $\mathbf{u}$ and $\mathbf{v}$. In $\mathsf{Priv}_2$, we consider two matrices $\mathbf{A}$ and $\mathbf{A}'$ neighboring if $\|\mathbf{A} - \mathbf{A}'\|_F \leq 1$.

Our algorithm relies heavily on some results from the theory of random projections.

**Definition 2.** *A distribution $\mathcal{D}_R$ of $t \times m$ matrices satisfies $(\alpha, \delta)$-subspace embedding for generalized regression if it has the following property: for any matrices $\mathbf{P} \in \mathbb{R}^{m \times n}$ and $\mathbf{Q} \in \mathbb{R}^{m \times n'}$ such that $\mathsf{rank}(\mathbf{P}) \leq r$, with probability $1 - \delta$ over $\mathbf{\Phi} \sim \mathcal{D}_R$, if $\widetilde{\mathbf{X}} = \mathrm{argmin}_{\mathbf{X}} \|\mathbf{\Phi}(\mathbf{PX} - \mathbf{Q})\|_F$ and $\widehat{\mathbf{X}} = \mathrm{argmin}_{\mathbf{X} \in \mathbb{R}^{n \times n'}} \|\mathbf{PX} - \mathbf{Q}\|_F$, then $\|\mathbf{P}\widetilde{\mathbf{X}} - \mathbf{Q}\|_F \leq (1 + \alpha)\|\mathbf{P}\widehat{\mathbf{X}} - \mathbf{Q}\|_F$.*

**Definition 3.** *A distribution $\mathcal{D}_A$ over $v \times m$ matrices satisfies $(\alpha, \delta)$-affine subspace embedding if it has the following property: for any matrices $\mathbf{D} \in \mathbb{R}^{m \times n}$ and $\mathbf{E} \in \mathbb{R}^{m \times n'}$ such that $\mathsf{rank}(\mathbf{D}) \leq r$, with probability $1 - \delta$ over $\mathbf{S} \sim \mathcal{D}_A$, simultaneously for all $\mathbf{X} \in \mathbb{R}^{n \times n'}$, $\|\mathbf{S}(\mathbf{DX} - \mathbf{E})\|_F^2 = (1 \pm \alpha)\|\mathbf{DX} - \mathbf{E}\|_F^2$.*

An example distribution $\mathcal{D}_R$ with $t = O(\alpha^{-2} \log(1/\delta))$ is the distribution of random matrices whose entries are sampled i.i.d. from $\mathcal{N}(0, 1/t)$.

## 3  A Meta Low Space Differentially Private Algorithm

Our aim in this section is to present a unified algorithmic approach in the form of meta algorithm (see, Algorithm 1). This serves the purpose of illustrating the key ideas. For example, since one of our goals is private algorithms under turnstile model, we are restricted to only use linear sketches [38], but what we show is that advance yet inexpensive post-processing combined with careful analysis can lead to a small error differentially private LRF.

**Our Techniques.** Our algorithm is based on two observations: **(i)** there is a way to *maintain differentially private sketches* of $\mathbf{A}$ (henceforth, we call such sketches *noisy sketches*) that incurs sub-optimal accuracy in terms of additive error and **(ii)** one can apply *post-processing* to these noisy sketches to obtain optimal additive error.

To illustrate point **(i)** and why we need post-processing, consider the following vanilla algorithm for approximating the right singular vector: compute $\mathbf{B} = \mathbf{\Phi A} + \mathbf{N}_1$, where $\mathbf{\Phi}$ satisfies certain embedding property (for example, [44, Theorem 1]) and $\mathbf{N}_1 \sim \mathcal{N}(0, \rho_1^2)^{\widetilde{O}(n^2) \times n}$ for $\rho_1$ as defined in Figure 1. The output is $[\mathbf{B}]_k$, the best rank-$k$ approximation of $\mathbf{B}$. This already gives a good approximation. Let $m \gg n^2$ and let $[\widetilde{\mathbf{U}}]_k [\widetilde{\mathbf{\Sigma}}]_k [\widehat{\mathbf{V}}]_k^{\mathsf{T}}$ be the singular value decomposition of $[\mathbf{B}]_k$. Then by embedding property of $\mathbf{\Phi}$ [44, Theorem 1],

$$
\begin{aligned}
\|\mathbf{A} - \mathbf{A}[\widetilde{\mathbf{V}}]_k[\widetilde{\mathbf{V}}]_k^{\mathsf{T}}\|_F &\leq \|(\mathbf{A} + \mathbf{\Phi}^\dagger \mathbf{N}_1) - (\mathbf{A} + \mathbf{\Phi}^\dagger \mathbf{N}_1)[\widetilde{\mathbf{V}}]_k[\widetilde{\mathbf{V}}]_k^{\mathsf{T}}\|_F + \|\mathbf{\Phi}^\dagger \mathbf{N}_1 + \mathbf{\Phi}^\dagger \mathbf{N}_1[\widetilde{\mathbf{V}}]_k[\widetilde{\mathbf{V}}]_k^{\mathsf{T}}\|_F \\
&\leq (1 - \alpha)^{-1}\|\mathbf{B}(\mathbb{I} - [\widetilde{\mathbf{V}}]_k[\widetilde{\mathbf{V}}]_k^{\mathsf{T}})\|_F + O(\|\mathbf{\Phi}^\dagger \mathbf{N}_1 + \mathbf{\Phi}^\dagger \mathbf{N}_1[\widetilde{\mathbf{V}}]_k[\widetilde{\mathbf{V}}]_k^{\mathsf{T}}\|_F) \\
&\leq (1 - \alpha)\|\mathbf{B}(\mathbb{I} - [\mathbf{V}]_k[\mathbf{V}]_k^{\mathsf{T}})\|_F + O(\|\mathbf{\Phi}^\dagger \mathbf{N}_1 + \mathbf{\Phi}^\dagger \mathbf{N}_1[\widetilde{\mathbf{V}}]_k[\widetilde{\mathbf{V}}]_k^{\mathsf{T}}\|_F) \\
&\leq (1 + \alpha)(1 - \alpha)^{-1}\|\mathbf{A} - [\mathbf{A}]_k\|_F + O(\|\mathbf{\Phi}^\dagger \mathbf{N}_1\|_F + \|\mathbf{\Phi}^\dagger \mathbf{N}_1[\widetilde{\mathbf{V}}]_k[\widetilde{\mathbf{V}}]_k^{\mathsf{T}}\|_F).
\end{aligned}
$$

The term in $O(\cdot)$ can be bounded using the embedding property of $\mathbf{\Phi}$, but this incurs large error. The question is whether we can further improve it to get optimal additive error. We show that it is possible using careful post-processing (point **(ii)** above). That is, we can extract top-$k$ singular components of the input matrix $\mathbf{A}$ from sketches that are appropriately perturbed to preserve differential privacy. The underlying idea is as follows: suppose we know the singular value decomposition of $[\mathbf{A}]_k := [\mathbf{U}]_k[\mathbf{\Sigma}]_k[\mathbf{V}]_k^{\mathsf{T}}$. Then for finding a matrix $\mathbf{B}$ such that $\mathbf{B} \approx \mathbf{A}$, it suffices to compute $\widetilde{\mathbf{U}}$ that approximates $[\mathbf{U}]_k$, $\widetilde{\mathbf{\Sigma}}$ that approximates $[\mathbf{\Sigma}]_k$, and $\widetilde{\mathbf{V}}$ that approximates $[\mathbf{V}]_k$, and set $\mathbf{B} := \widetilde{\mathbf{U}}\widetilde{\mathbf{\Sigma}}\widetilde{\mathbf{V}}^{\mathsf{T}}$. However, this over simplistic overview does not guarantee privacy. In the rest of this exposition, we give a brief overview of how we turn this simplistic overview to a private algorithm.

**Challenges in computing differentially private low-rank factorization.** The two traditional methods to preserve privacy—input perturbation and output perturbation—do not provide both privacy and small additive error. For example, if we use output perturbation to compute the sketches, $\mathbf{Y}_c = \mathbf{A\Phi} + \mathbf{N}$ and $\mathbf{Y}_r = \mathbf{\Psi A} + \mathbf{N}'$ for appropriate sketching matrices $\mathbf{\Phi}$ and $\mathbf{\Psi}$ and noise matrices $\mathbf{N}$ and $\mathbf{N}'$, and use known random projection results, then we get an additive error term that can be arbitrarily large (more specifically, depends on the Frobenius norm of $\mathbf{A}$ and has the form $\|\mathbf{N}\mathcal{L}_{\mathbf{A}}\mathbf{N}'\|_F$ for some linear function $\mathcal{L}_{\mathbf{A}}$ of $\mathbf{A}$). More precisely, we can show that $\min_{r(\mathbf{X}) \leq k} \|\mathbf{Y}_c \mathbf{X} \mathbf{Y}_r - \mathbf{A}\|_F \leq \|\mathbf{A} - [\mathbf{A}]_k\|_F + \|\mathbf{N}\mathcal{L}_{\mathbf{A}}\mathbf{N}'\|_F + \|\mathbf{N}\mathcal{L}_{\mathbf{A}}\mathbf{\Psi A}\|_F + \|\mathbf{A\Phi}\mathcal{L}_{\mathbf{A}}\mathbf{N}'\|_F$.

---

**Algorithm 1** PRIVATE-OPTIMAL-LRF($\mathbf{A}; (\epsilon, \delta); \alpha; k$)

---

1: **Set** $\eta = \max\{k, \alpha^{-1}\}$, $t = O(\eta\alpha^{-1}\log(k/\delta))$, $v = O(\eta\alpha^{-2}\log(k/\delta))$ and $\sigma_{\min} = 16\log(1/\delta)\sqrt{t(1+\alpha)(1-\alpha)^{-1}\ln(1/\delta)}/\varepsilon$, $\rho_1 = \sqrt{(1+\alpha)\ln(1/\delta)}/\varepsilon$, $\rho_2 = \sqrt{(1+\alpha)}\rho_1$. Sample $\mathbf{\Phi} \sim \mathcal{N}(0, 1/t)^{(m+n)\times m}$, $\mathbf{\Psi} \sim \mathcal{N}(0, 1/t)^{t\times m}$, $\mathbf{S} \sim \mathcal{N}(0, 1/v)^{v\times m}$, $\mathbf{T} \sim \mathcal{N}(0, 1/v)^{v\times(m+n)}$ with every entry sampled i.i.d. from $\mathcal{N}(0, 1)$. Sample $\mathbf{N}_1 \sim \mathcal{N}(0, \rho_1^2)^{t\times(m+n)}$ and $\mathbf{N}_2 \sim \mathcal{N}(0, \rho_2^2)^{v\times v}$. Keep $\mathbf{N}_1, \mathbf{N}_2, \mathbf{\Phi}$ private.
2: **Set** $\widehat{\mathbf{A}} = (\mathbf{A} \quad \sigma_{\min}\mathbb{I}_m)$ by padding $\sigma_{\min}\mathbb{I}_m$ to the columns of $\mathbf{A}$, where $\mathbb{I}_m$ denotes an $m \times m$ identity matrix. Compute $\mathbf{Y}_c = \widehat{\mathbf{A}}\mathbf{\Phi}$, $\mathbf{Y}_r = \mathbf{\Psi}\widehat{\mathbf{A}} + \mathbf{N}_1$, and $\mathbf{Z} = \mathbf{S}\widehat{\mathbf{A}}\mathbf{T}^\mathsf{T} + \mathbf{N}_2$.
3: **Compute:** $\mathbf{U} \in \mathbb{R}^{m\times t}$ whose columns are orthonormal basis for the column space of $\mathbf{Y}_c$ and matrix $\mathbf{V} \in \mathbb{R}^{t\times(m+n)}$ whose rows are the orthonormal basis for the row space of $\mathbf{Y}_r$.
4: **Compute:** SVD of $\mathbf{SU} := \widetilde{\mathbf{U}}_s\widetilde{\mathbf{\Sigma}}_s\widetilde{\mathbf{V}}_s^\mathsf{T} \in \mathbb{R}^{v\times t}$ and a SVD of $\mathbf{VT}^\mathsf{T} := \widetilde{\mathbf{U}}_t\widetilde{\mathbf{\Sigma}}_t\widetilde{\mathbf{V}}_t^\mathsf{T} \in \mathbb{R}^{t\times v}$.
5: **Compute:** SVD of $\widetilde{\mathbf{V}}_s\widetilde{\mathbf{\Sigma}}_s^\dagger[\widetilde{\mathbf{U}}_s^\mathsf{T}\mathbf{Z}\widetilde{\mathbf{V}}_t]_k\widetilde{\mathbf{\Sigma}}_t^\dagger\widetilde{\mathbf{U}}_t^\mathsf{T}$. Let it be be $\mathbf{U}'\mathbf{\Sigma}'\mathbf{V}'^\mathsf{T}$.
6: **Output:** $\widetilde{\mathbf{U}} = \mathbf{U}\mathbf{U}'$, diagonal matrix $\widetilde{\mathbf{\Sigma}} = \mathbf{\Sigma}'$, and $\widetilde{\mathbf{V}} = \mathbf{V}^\mathsf{T}\mathbf{V}'$.

---

While $\min_{r(\mathbf{X})\leq k} \|\mathbf{Y}_c\mathbf{X}\mathbf{Y}_r - \mathbf{A}\|_F$ can be lower bounded using the techniques we use in this paper, the additive term $\|\mathbf{N}\mathcal{L}_\mathbf{A}\mathbf{N}'\|_F$ can have large Frobenius norm.

On the other hand, input perturbation of $\mathbf{A}$ followed by a multiplication by Gaussian matrices $\mathbf{\Omega}_1$ and $\mathbf{\Omega}_2$ as in [8, 57, 58] can leak private data due to a subtle reason. Every row of $\mathbf{\Omega}_1\mathbf{A}$ (and columns of $\mathbf{A}\mathbf{\Omega}_2$) has a multivariate Gaussian distribution if the determinant of $\mathbf{A}^\mathsf{T}\mathbf{A}$ ($\mathbf{A}\mathbf{A}^\mathsf{T}$, respectively) is non zero. If $m < n$, one can prove that computing $\mathbf{A}\mathbf{\Omega}_1$ preserves privacy, but, since, $\mathbf{A}$ is not a full-column rank matrix, the multivariate Gaussian distribution is not defined. The trick to consider the subspace orthogonal to the kernel space of $\mathbf{A}$ [8] does not work because span of $\mathbf{A}$ and $\mathbf{A}'$ may not coincide for neighboring matrices $\mathbf{A}$ and $\mathbf{A}'$. If the span do not coincide, then one can easily differentiate the two cases with high probability, violating differential privacy. In fact, until this work, it was not even clear whether using input perturbation yields low rank approximation (see the comment after Theorem IV.2 and discussion in Section V in Blocki *et al.* [8])!

**Our Algorithm.** We use input perturbation with a careful choice of parameter to one of the sketches and output perturbation to the other two sketches and show that it incur optimal additive error and preserve privacy. The intuitive reason why this incurs small additive error is the fact that only one of the sketches, $\mathbf{Y}_r$ or $\mathbf{Y}_c$, undergoes output perturbation, so there is no term like $\|\mathbf{N}\mathcal{L}_\mathbf{A}\mathbf{N}'\|_F$ as above. This allows us to show that $\mathbf{Y}_c$ and $\mathbf{Y}_r$ (or equivalently, their orthonormal bases $\mathbf{U}$ and $\mathbf{V}$ as formed in Algorithm 1) approximates the span of $[\mathbf{U}]_k$ and $[\mathbf{V}]_k$ up to a small additive error.

Once we have extracted a "good enough" $\mathbf{U}$ and $\mathbf{V}$, our problem reduces to computing $\operatorname{argmin}_{rk(\mathbf{X})\leq k} \|\mathbf{A} - \mathbf{U}\mathbf{X}\mathbf{V}\|_F$. This would require storing the whole matrix $\mathbf{A}$, something that we wish to avoid. To avoid storing the whole $\mathbf{A}$, we use the fact that $\mathbf{S}$ and $\mathbf{T}$ are sampled from a distribution of random matrices with a property that, for all appropriate $\mathbf{X}$, $\|\mathbf{A} - \mathbf{U}\mathbf{X}\mathbf{V}\|_F \approx \|\mathbf{S}(\mathbf{A} - \mathbf{U}\mathbf{X}\mathbf{V})\mathbf{T}^\mathsf{T}\|_F$. In other words, without privacy, $\operatorname{argmin}_{rk(\mathbf{X})\leq k} \|\mathbf{S}(\mathbf{A} - \mathbf{U}\mathbf{X}\mathbf{V})\mathbf{T}^\mathsf{T}\|_F$ can be used to get a "good" approximation of $[\mathbf{\Sigma}]_k$. The exact method to perform and analyze the approximation of $[\mathbf{\Sigma}]_k$ is slightly more involved because we only have access to the noisy version of $\mathbf{SAT}$, i.e., $\mathbf{Z}$ (in fact, this is one of the places we need careful post processing to output an approximation to $\mathbf{\Sigma}_k$ under a rotation and a small additive error).Finally, we arrive at the main result stated below for the case when $m \leq n$ (the result when $m > n$ can be derived by just swapping $m$ and $n$).

**Theorem 1** (Main result). *Let $m, n, k \in \mathbb{N}$ and $\alpha, \varepsilon, \delta$ be the input parameters (with $m \leq n$). Let $\kappa$, $\eta$, and $\sigma_{\min}$ be as defined in Algorithm 1. Given an $m \times n$ matrix $\mathbf{A}$ with $\mathsf{nn}(\mathbf{A})$ non-zero entries, let $(\mathbf{A} \quad \mathbf{0})$ be a matrix formed by appending an all zero $m \times m$ matrix to $\mathbf{A}$. Then PRIVATE-OPTIMAL-LRF (Algorithm 1) is $(3\varepsilon, 3\delta)$ differentially private under* $\mathsf{Priv}_1$ *and outputs a factorization $\widetilde{\mathbf{U}}, \widetilde{\mathbf{\Sigma}}, \widetilde{\mathbf{V}}$ such that*

1. *With probability $9/10$ over the random coins of* PRIVATE-SPACE-OPTIMAL-LRF,

$$\|(\mathbf{A} \quad \mathbf{0}) - \widetilde{\mathbf{U}}\widetilde{\mathbf{\Sigma}}\widetilde{\mathbf{V}}^\mathsf{T}\|_F \leq (1+\alpha)\|\mathbf{A} - [\mathbf{A}]_k\|_F + O(\sigma_{\min}\sqrt{m} + \varepsilon^{-1}\sqrt{kn\ln(1/\delta)}).$$

2. *The space used by* PRIVATE-SPACE-OPTIMAL-LRF *is $O((m+n)\eta\alpha^{-1}\log(k/\delta))$.*

*Proof Sketch.* The proof of Theorem 1 is presented in the supplementary material. Here, we give a brief sketch of part 1 (for $m \leq n$) to illustrate the key points. The intuition that there is no term like

$\|\mathbf{N}\mathcal{L_A}\mathbf{N}'\|_F$ does not directly yield optimal additive error. This is because, even if we do not get an additive error term with large value like $\|\mathbf{N}\mathcal{L_A}\mathbf{N}'\|_F$, if not analyzed precisely, one can either get a non-analytic expression for the error terms or one that is difficult to analyze. To get analytic expressions for all the error terms that are also easier to analyze, we introduce two carefully chosen optimization problems (equation (3)) so that the intermediate terms in our analysis satisfy certain properties (see the proof sketch below for exact requirements). Let $\widehat{\mathbf{A}}$ be as defined in Figure 1. Part 1 follows from the following chain of inequalities and bounding $\|\widehat{\mathbf{A}}\mathbf{\Phi}\mathcal{L_A}\mathbf{N}_1\|_F$:

$$\|\mathbf{M}_k - (\mathbf{A} \quad \mathbf{0})\|_F \le \|\mathbf{M}_k - \widehat{\mathbf{A}}\|_F + O(\sigma_{\min}\sqrt{m})$$
$$\le (1+\alpha)\|\widehat{\mathbf{A}} - [\widehat{\mathbf{A}}]_k\|_F + \|\widehat{\mathbf{A}}\mathbf{\Phi}\mathcal{L_A}\mathbf{N}_1\|_F + O(\sigma_{\min}\sqrt{m})$$
$$\le (1+\alpha)\|\mathbf{A} - [\mathbf{A}]_k\|_F + \|\widehat{\mathbf{A}}\mathbf{\Phi}\mathcal{L_A}\mathbf{N}_1\|_F + O(\sigma_{\min}\sqrt{m}), \qquad (1)$$

where the matrix $\mathcal{L_A}$ satisfies the following properties: (a) $\|\widehat{\mathbf{A}}\mathbf{\Phi}\mathcal{L_A}\mathbf{\Psi}\widehat{\mathbf{A}} - \widehat{\mathbf{A}}\|_F \le (1+\alpha)\|\widehat{\mathbf{A}} - [\widehat{\mathbf{A}}]_k\|_F$, (b) $\mathcal{L_A}$ has rank at most $k$, and (c) $\mathbf{\Psi}\mathbf{A}\mathbf{\Phi}\mathcal{L_A}$ is a rank-$k$ projection matrix. We use subadditivity of norm to prove the first inequality and Weyl's perturbation theorem [7] to prove the third inequality. Proving the second inequality is the technically involved part. For this, we need to find a candidate $\mathcal{L_A}$. We first assume we have such a candidate $\mathcal{L_A}$ with all the three properties. Once we have such an $\mathcal{L_A}$, we can prove part (b) as follows:

$$\min_{\text{rk}(\mathbf{X}) \le k} \|\mathbf{U}\mathbf{X}\mathbf{V} - \mathbf{B}\|_F \le \|\widehat{\mathbf{A}}\mathbf{\Phi}\mathcal{L_A}\mathbf{\Psi}\widehat{\mathbf{A}} - \widehat{\mathbf{A}}\|_F + \|\widehat{\mathbf{A}}\mathbf{\Phi}\mathcal{L_A}\mathbf{N}_1\|_F + \|\mathbf{S}^\dagger\mathbf{N}_1(\mathbf{T}^\dagger)^\mathsf{T}\|_F$$
$$\le (1+\alpha)\|\widehat{\mathbf{A}} - [\widehat{\mathbf{A}}]_k\|_F + \|\widehat{\mathbf{A}}\mathbf{\Phi}\mathcal{L_A}\mathbf{N}_1\|_F + \|\mathbf{S}^\dagger\mathbf{N}_2(\mathbf{T}^\mathsf{T})^\dagger\|_F, \qquad (2)$$

where $\mathbf{B} = \mathbf{A} + \mathbf{S}^\dagger\mathbf{N}_1(\mathbf{T}^\dagger)^\mathsf{T}$. The first inequality follows from the subadditivity of Frobenius norm, the fact that $\mathbf{U}$ and $\mathbf{V}$ are orthonormal bases of $\mathbf{Y}_c$ and $\mathbf{Y}_r$, and property (b) to exploit that minimum on the left hand side is over rank-$k$ matrices. We then use the approximation guarantee of property (a) to get the second inequality. Using the fact that $\mathbf{S}$ and $\mathbf{T}$ are Gaussian matrices, we can lower bound the left hand side of equation (2) up to an additive term as follows:

$$\|(\mathbf{A} \quad \mathbf{0}) - \widetilde{\mathbf{U}}\widetilde{\mathbf{\Sigma}}\widetilde{\mathbf{V}}^\mathsf{T}\|_F - \|\mathbf{S}^\dagger\mathbf{N}_1(\mathbf{T}^\mathsf{T})^\dagger\|_F \le (1+\alpha)^3 \min_{\text{rk}(\mathbf{X}) \le k} \|\mathbf{U}\mathbf{X}\mathbf{V} - \mathbf{B}\|_F,$$

where $\widetilde{\mathbf{U}}, \widetilde{\mathbf{\Sigma}}$, and $\widetilde{\mathbf{V}}$ are as in Algorithm 1. We upper bound the right hand side of equation (2) by using Markov's inequality combined with the fact that both $\mathbf{S}$ and $\mathbf{T}$ are Gaussian matrices and $\mathcal{L_A}$ satisfies property (c). Scaling the value of $\alpha$ by a constant gives part 1. So all that remains is to find a candidate matrix $\mathcal{L_A}$. We construct such an $\mathcal{L_A}$ using the following two optimization problems:

$$\mathsf{Prob}_1 : \min_{\mathbf{X}} \|\mathbf{\Psi}(\widehat{\mathbf{A}}\mathbf{\Phi}([\widehat{\mathbf{A}}]_k\mathbf{\Phi})^\dagger\mathbf{X} - \widehat{\mathbf{A}})\|_F \quad \text{and} \quad \mathsf{Prob}_2 : \min_{\mathbf{X}} \|\widehat{\mathbf{A}}\mathbf{\Phi}([\widehat{\mathbf{A}}]_k\mathbf{\Phi})^\dagger\mathbf{X} - \widehat{\mathbf{A}}\|_F. \quad (3)$$

We prove that a solution to $\mathsf{Prob}_1$ gives us a candidate $\mathcal{L_A}$. This completes the proof. $\qquad\square$

**From** $\mathsf{Priv}_1$ **to** $\mathsf{Priv}_2$**.** If we try to use the idea described above to prove differential privacy under $\mathsf{Priv}_2$, we end up with an additive error that depends linearly on $\min\{m, n\}$. This is because we need to perturb the input matrix by a noise proportional to $\min\{\sqrt{km}, \sqrt{kn}\}$ to preserve differential privacy under $\mathsf{Priv}_2$. We show that by maintaining noisy sketches $\mathbf{Y} = \mathbf{A}\mathbf{\Phi} + \mathbf{N}_1$ and $\mathbf{Z} = \mathbf{S}\mathbf{A} + \mathbf{N}_2$ for appropriately chosen noise matrices $\mathbf{N}_1$ and $\mathbf{N}_2$ and sketching matrices $\mathbf{\Phi}$ and $\mathbf{S}$, followed by some post processing, we can have an optimal error differentially private algorithm under $\mathsf{Priv}_2$. Here, we require $\mathbf{S}$ to satisfy the same property as in the case of $\mathsf{Priv}_1$. However, the lack of symmetry between $\mathbf{S}$ and $\mathbf{\Phi}$ requires us to decouple the effects of noise matrices to get a tight bound on the additive error. In total, we get an efficient $(\epsilon, \delta)$-differentially private algorithm that uses $O((m\alpha^{-1} + n)k\alpha^{-1})$ space and outputs $(\alpha, 99/100, \gamma, k)$-LRF for $\gamma = \widetilde{O}((\sqrt{km}\alpha^{-2} + \sqrt{kn})\sqrt{\log(1/\delta)/\epsilon^{-2}})$.

## 4  Differentially Private Algorithms for Streaming Matrices

We next give more details of our result when matrices are streamed. Unless specified, for the ease of presentation, we assume that $k \ge 1/\alpha, \delta = \Theta(n^{-\log n})$, and $\widetilde{O}(\cdot)$ hides a poly $\log n$ factor. To capture the scenarios where data matrices are constantly updated, we consider the *turnstile update model* (see the survey [46] for further motivations). Formally, in a turnstile update model, a matrix

| | Privacy Notion | Additive Error | Space Required | Streaming |
|---|---|---|---|---|
| **This work** | $\mathbf{A} - \mathbf{A}' = \mathbf{u}\mathbf{v}^\mathsf{T}$ | $\widetilde{O}((\sqrt{km}\alpha^{-1} + \sqrt{kn})\varepsilon^{-1})$ | $\widetilde{O}((m+n)k\alpha^{-1})$ | Turnstile |
| **This work** | $\|\mathbf{A} - \mathbf{A}'\|_F = 1$ | $\widetilde{O}((\sqrt{km}\alpha^{-2} + \sqrt{kn})\varepsilon^{-1})$ | $\widetilde{O}((m\alpha^{-1} + n)k\alpha^{-1})$ | Turnstile |
| Hardt-Roth [30] | $\mathbf{A} - \mathbf{A}' = \mathbf{e}_s\mathbf{v}^\mathsf{T}$ | $\widetilde{O}\left(\left(\sqrt{km} + kc\sqrt{n}\right)\varepsilon^{-1}\right)$ | $O(mn)$ | $\times$ |
| Upadhyay [57] | $\mathbf{A} - \mathbf{A}' = \mathbf{e}_s\mathbf{v}^\mathsf{T}$ | $\widetilde{O}\left(\left(k^2\sqrt{n+m}\right)\varepsilon^{-1}\right)$ | $\widetilde{O}((m+n)k\alpha^{-1})$ | Row-wise |
| Lower Bounds | All of the above | $\Omega\left(\sqrt{km} + \sqrt{kn}\right)$ [30] | $\Omega((m+n)k\alpha^{-1})$ | Turnstile |

Table 2: Comparison of Results ($\|\mathbf{u}\|_2, \|\mathbf{v}\|_2 = 1$, $\mathbf{e}_s$: standard basis, $k \leq 1/\alpha$).

$\mathbf{A} \in \mathbb{R}^{m \times n}$ is initialized to an all zero-matrix and is updated by a sequence of triples $\{i, j, \Delta\}$, where $1 \leq i \leq m, 1 \leq j \leq n$, and $\Delta \in \mathbb{R}$. Each update results in a change in the $(i,j)$-th entry of $\mathbf{A}$ as follows: $\mathbf{A}_{i,j} \leftarrow \mathbf{A}_{i,j} + \Delta$.

An algorithm is *differentially private under turnstile update model* if, for all possible matrices updated in the turnstile update model and runs of the algorithm, the output of the algorithm is $(\varepsilon, \delta)$-differentially private. A straightforward application of known privacy techniques to make known space-optimal non-private algorithms [9] differentially private incurs a large additive error.

In other words, it is an *open question* whether we can solve Problem 1 with good accuracy while preserving differential privacy and receiving the matrix in the turnstile update model? We resolve this question positively. We say two data streams are neighboring if they are formed by neighboring matrices. We show the following:

**Theorem 2.** *Let $\mathbf{A}$ be an $m \times n$ matrix streamed in a turnstile update model. Then there is an efficient $(\varepsilon, \delta)$-differentially private algorithm under $\mathsf{Priv}_1$ that uses $\widetilde{O}((m+n)k\alpha^{-1})$ space and computes $(\alpha, 99/100, \gamma, k)$-$\mathsf{LRF}$, where $\gamma = \widetilde{O}((\sqrt{mk\alpha^{-1}} + \sqrt{kn})/\varepsilon)$. There is also an efficient $(\epsilon, \delta)$-differentially private algorithm under $\mathsf{Priv}_2$ that computes an $(\alpha, 99/100, \gamma, k)$-$\mathsf{LRF}$, where $\gamma = \widetilde{O}((\sqrt{mk\alpha^{-2}} + \sqrt{kn})/\varepsilon)$.*

Before we argue the tightness of Theorem 2 with respect to both space and additive error, we compare our result with previous works. All the private algorithms prior to this work compute a low rank approximation of either the matrix $\mathbf{A}$ or its covariance $\mathbf{A}^\mathsf{T}\mathbf{A}$. One can compute a factorization from their output at the expense of an extra $O(mn^2)$ time and $O(mn)$ space (Dwork *et al.* [24] requires an extra $O(n^3)$ time and $O(n^2)$ space to output an $\mathsf{LRF}$ of $\mathbf{A}^\mathsf{T}\mathbf{A}$). Some works like [11, 35, 30, 29] compute $\mathsf{LRF}$ under the spectral norm instead of Frobenius norm.

In other words, Hardt and Roth [30] and Upadhyay [57] study a problem closest to ours (the differences being that they do not consider turnstile updates and output a low rank matrix). Therefore, we compare Theorem 2 only with these two results. We do not make any assumptions on the private matrix. This allows us to cover matrices of all form and relaxations in an unified manner. We next compare the accuracy, privacy guarantees, space, and time required in more detail (see Table 2).

Both Hardt and Roth [30] and Upadhyay [57] give rank-$O(k)$ approximation instead of rank-$k$ approximation, incur a multiplicative error of $\sqrt{1 + k/p}$, where $p$ is an oversampling parameter (typically, $p = \Theta(k)$), and $m \leq n$. Therefore, for a reasonable comparison, we consider Theorem 2 when $\alpha = \Theta(1)$ and $m \leq n$. Our additive error is smaller than Upadhyay [57] by a factor of $\widetilde{O}(k^{3/2})$. To make a reasonable comparison with Hardt and Roth [30], we consider their result without incoherence assumption: which roughly says that no single row of the matrix is significantly correlated with any of the right singular vectors of the matrix. Then Hardt and Roth [30, Theorem 4.2 and 4.7] results in an additive error $\widetilde{O}((\sqrt{km} + ck\sqrt{n})\varepsilon^{-1})$, where $c$ is the maximum entry in their projection matrix. In other words, we improve Hardt and Roth [30] by an $\widetilde{O}(c\sqrt{k})$ factor.

Our algorithms are more efficient than previous algorithms in terms of space and time even though earlier algorithms output a rank-$O(k)$ matrix and cannot handle updates in the turnstile model. Upadhyay [57] takes more time than Hardt and Roth [30]. The algorithm of Hardt and Roth [30] uses $O(mn)$ space since it is a private version of Halko *et al.* [28] and has to store the entire matrix: both the stages of Halko *et al.* [28] require the matrix explicitly. One of the motivations mentioned in Hardt and Roth [30] is sparse private incoherent matrices (see the discussion in Hardt and Roth [30, Sec 1.1]), but their algorithm uses this only to reduce the additive error and not the running time.

On the other hand, our algorithms use sublinear space and almost matches the running time of most efficient non-private algorithm in the turnstile model [9, 13].

Our privacy guarantees are also more general than previous works, who consider two matrices $\mathbf{A}$ and $\mathbf{A}'$ neighboring either if $\mathbf{A} - \mathbf{A}' = \mathbf{e}_i \mathbf{e}_j^\mathsf{T}$ [29, 31, 33] or $\mathbf{A} - \mathbf{A}' = \mathbf{e}_i \mathbf{v}^\mathsf{T}$ for some unit vector $\mathbf{v}$ [24, 30, 57], depending on whether a user's data is an entry of the matrix or a row of the matrix. It is easy to see that these privacy guarantees are special case of $\mathsf{Priv}_1$ and $\mathsf{Priv}_2$.

**Tightness of Additive Error.** Hardt and Roth [30] showed a lower bound of $\Omega(\sqrt{kn} + \sqrt{km})$ on additive error by showing a reduction to the linear reconstruction attack [18]. In other words, any algorithm that outputs a low rank matrix with additive error $o(\sqrt{kn} + \sqrt{km})$ cannot be differentially private! This lower bound holds even when the private algorithm can access the private matrix any number of times. Our results show that one can match the lower bound for constant $\alpha$, a setting considered in Hardt and Roth [30], up to a small logarithmic factor, while allowing access to the private matrix only in the turnstile model.

**Space Lower Bound and Optimality of the Algorithm Under** $\mathsf{Priv}_1$**.** Our algorithms use same space as non-private algorithm up to a logarithmic factor, which is known to be optimal for $\gamma = 0$ [12]. However, we incur a non-zero additive error, $\gamma$, which is inevitable [30], and it is not clear if we can achieve better space algorithm when $\gamma \neq 0$.

We complement Theorem 2 with a lower bound on the space required for low-rank approximation with non-trivial additive error. Our result holds for any randomized algorithm; therefore, also hold for any private algorithm. This we believe makes our result of independent interest.

**Theorem 3.** *The space required by any randomized algorithm to solve* $(\alpha, 1/6, O(m+n), k)$*-LRF in the turnstile update model is* $\Omega((n+m)k\alpha^{-1})$.

Any differentially private incurs an additive error $\Omega(\sqrt{km} + \sqrt{kn})$. Moreover, known differentially private low-rank approximation [30] set $\alpha = \sqrt{2} - 1$. This thus prove optimality for all $k \geq 3$.

**Under Bounded Norm Assumptions.** In some practical applications, matrices are more structured. One such special case is when the rows of private matrix $\mathbf{A}$ have bounded norm and one would like to approximate $\mathbf{A}^\mathsf{T}\mathbf{A}$. This problem was studied by Dwork *et al.* [24]. We consider the matrices are updated by row-wise: all the updates at time $\tau \leq T$ are of the form $\{i_\tau, \mathbf{A}^{(\tau)}\}$, where $1 \leq i_\tau \leq m$, $\mathbf{A}^{(\tau)} \in \mathbb{R}^n$, and $i_\tau \neq i_{\tau'}$ for all $\tau \neq \tau'$. We show the following by using $\mathbf{A}^\mathsf{T}\mathbf{A}$ as the input matrix:

**Corollary 1.** *Given an* $\mathbf{A} \in \mathbb{R}^{m \times n}$ *updated by inserting one row at a time such that every row has a bounded norm 1 and* $m > n$. *Then there is an* $(\varepsilon, \delta)$*-differentially private algorithm under* $\mathsf{Priv}_2$ *that uses* $\widetilde{O}(nk\alpha^{-2})$ *space, and outputs a rank-$k$ matrix* $\mathbf{B}$ *such that* $\|\mathbf{A}^\mathsf{T}\mathbf{A} - \mathbf{B}\|_F \leq (1+\alpha)\|\mathbf{A}^\mathsf{T}\mathbf{A} - [\mathbf{A}^\mathsf{T}\mathbf{A}]_k\|_F + \widetilde{O}(\sqrt{nk}/(\alpha\varepsilon))$.

We do not violate the lower bound of Dwork *et al.* [24] because their lower bound is valid when $\alpha = 0$, which is not possible for low space algorithms due to Theorem 3. Dwork *et al.* [24] bypassed their lower bounds under a stronger assumption known as *singular value separation*: the difference between $k$-th singular value and all $k'$-th singular values for $k' > k$ is at least $\omega(\sqrt{n})$. In other words, our result shows that we do not need singular value separation while using significantly less space—$\widetilde{O}(nk/\alpha^2)$ as compared to $O(n^2)$—if we are ready to pay for a small multiplicative error.

**Adapting to Continual Release Model.** Until now, we gave algorithms that produce the output only at the end of the stream. There is a related model called $(\varepsilon, \delta)$*-differential privacy under $T$-continual release* [23]. In this model, the server receives a stream of length $T$ and produces an output after every update, such that every output is $(\varepsilon, \delta)$-differentially private. We modify our meta algorithm to work in this model by using the fact that we only store noisy linear sketches of the private matrix during the updates and low-rank factorization is computed through post-processing on only the noisy sketches. That is, we can use the generic transformation [23] to maintain the sketch of the updates. A factorization for any time range can be done by aggregating the sketches for the specified range using range queries. This gives the *first instance* of algorithm that provides differentially private continual release of LRF. We show the following.

**Theorem 4.** *Let* $\mathbf{A} \in \mathbb{R}^{m \times n}$ *be the private matrix streamed over $T$ time epochs. Then there is an* $(\varepsilon, \delta)$*-differentially private algorithm under* $\mathsf{Priv}_1$ *that outputs a rank-$k$ factorization under the continual release for $T$ time epochs such that* $\gamma = \widetilde{O}(\varepsilon^{-1}(\sqrt{mk\alpha^{-1}} + \sqrt{kn})\log T)$.

# 5 Noninteractive Local Differentially Private PCA

Till now, we have considered a single server that receives the private matrix in a streamed manner. We next consider another variant of differential privacy known as *local differential privacy* (LDP) [21, 22, 26, 59]. In the *local* model, each individual applies a differentially private algorithm locally to their data and shares only the output of the algorithm—called a report—with a server that aggregates users' reports. A multi-player protocol is $\varepsilon$-LDP if for all possible inputs and runs of the protocol, the transcript of player $i$'s interactions with the server is $\varepsilon$-LDP.

One can study two variants of local differential privacy depending on whether the server and the users interact more than once or not. In the interactive variant, the server sends several messages, each to a subset of users. In the noninteractive variant, the server sends a single message to all the users at the start of the protocol and sends no message after that. Smith, Thakurta, and Upadhyay [54] argued that noninteractive locally private algorithms are ideal for implementation.

The natural extension of Problem 1 in the local model is when the matrix is distributed among the users such that every user has one row of the matrix and users are responsible for the privacy of their row vector. Unfortunately, known private algorithms (including the results presented till now) do not yield non trivial additive error in the local model. For example, if we convert Theorem 2 to the local model, we end up with an additive error $\widetilde{O}(\sqrt{kmn})$. This is worse than the trivial bound of $O(\sqrt{mn})$, for example, when $\mathbf{A} \in \{0,1\}^{m \times n}$, a trivial output of all zero matrix incurs an error at most $O(\sqrt{mn})$. In fact, existing lower bounds in the local model suggests that one is likely to incur an error which is $O(\sqrt{m})$ factor worse than in the central model, where $m$ is the number of users. However, owing to the result of Dwork *et al.* [24], we can hope to achieve non-trivial result for differentially private principal component analysis. This problem has been studied without privacy under the *row-partition model* [6, 39, 37, 9, 27, 50, 51, 55]). We exploit the fact that our meta algorithm only stores differentially private sketches of the input matrix to give a noninteractive algorithm for low-rank principal component analysis (PCA) under local differential privacy. This produces an $(\epsilon, \delta)$-locally differentially private algorithm; however, it is non-interactive. We then use the generic transformation of Bun *et al.* [10] to get the following result.

**Theorem 5.** *Let $m, n \in \mathbb{N}$ and $\alpha, \varepsilon, \delta$ be the input parameters. Let $k$ be the desired rank of the factorization and $\eta = \max\{k, \alpha^{-1}\}$. Let $v = O(\eta \alpha^{-2} \log(k/\delta))$. Given a private input matrix $\mathbf{A} \in \mathbb{R}^{m \times n}$ distributed in a row-wise manner amongst $m$ users, there is an efficient $\varepsilon$-local differentially private algorithm under $\mathsf{Priv}_2$ that uses $O(v^2)$ words of communications from users to central server and outputs a rank-k orthonormal matrix $\mathbf{U}$ such that with probability $9/10$,*

$$\|\mathbf{A} - \mathbf{U}\mathbf{U}^{\mathsf{T}}\mathbf{A}\|_F \le (1 + O(\alpha))\|\mathbf{A} - [\mathbf{A}]_k\|_F + O\left(v\sqrt{m\log(1/\delta)}/\epsilon\right).$$

# 6 Discussion on Neighboring Relation

The two privacy guarantees considered in this paper have natural reasons to be considered. $\mathsf{Priv}_1$ generalizes the earlier privacy guarantees and captures the setting where any two matrices differ in only one spectrum. Since $\mathsf{Priv}_1$ is defined in terms of the spectrum of matrices, $\mathsf{Priv}_1$ captures one of the natural privacy requirements in all the applications of LRF. $\mathsf{Priv}_2$ is stronger than $\mathsf{Priv}_1$. To motivate the definition of $\mathsf{Priv}_2$, consider a graph, $\mathcal{G} := (\mathcal{V}, \mathcal{E})$ that stores career information of people in a set $\mathcal{P}$ since their graduation. The vertex set $\mathcal{V}$ is the set of all companies. An edge $e = (u, v) \in \mathcal{E}$ has weight $\sum_{p \in \mathcal{P}}(t_{p,e}/t_p)$, where $t_{p,e}$ is the time for which the person $p$ held a job at $v$ after leaving his/her job at $u$, and $t_p$ is the total time lapsed since his/her graduation. Graphs like $\mathcal{G}$ are useful because the weight on every edge $e = (u, v)$ depends on the number of people who changed their job status from $u$ to $v$ (and the time they spent at $v$). Therefore, data analysts might want to mine such graphs for various statistics. In the past, graph statistics have been extensively studied for static graph under *edge-level privacy* (see, for e.g., [24, 30, 29, 56, 57]): the presence or absence of a person corresponds to a change in a single edge. On the other hand, in graphs like $\mathcal{G}$, presence or absence of a person would be reflected on many edges. If we use earlier results on edge-level privacy to such graphs, it would lead to either a large additive error or a loss in privacy parameters $\varepsilon, \delta$. $\mathsf{Priv}_2$ is an attempt to understand whether we can achieve any non-trivial guarantee on the additive error without depreciating the privacy parameters.

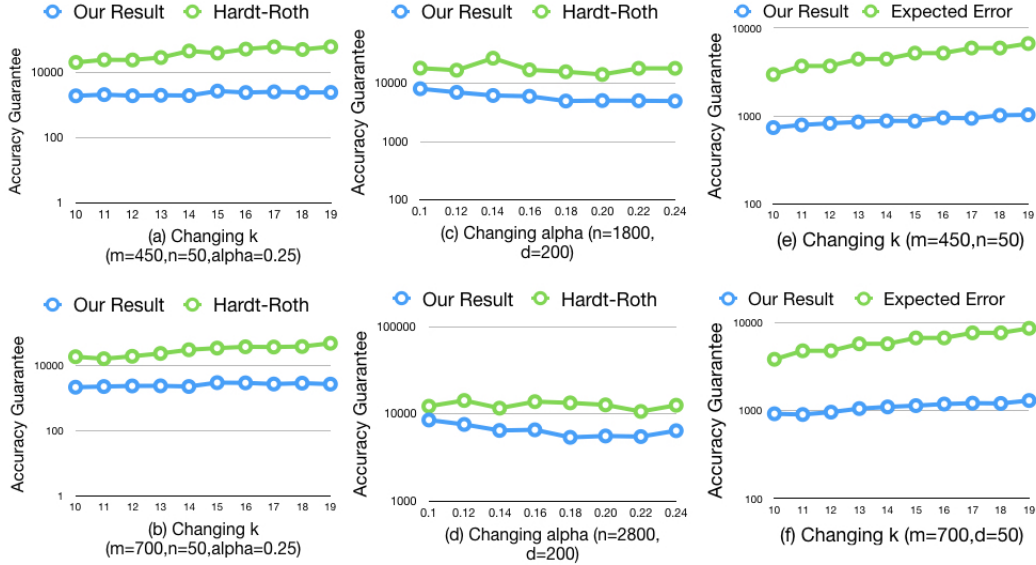

Figure 1: Empirical Evaluation of Additive Error of Our Algorithm.

# 7 Empirical Evaluation of Our Algorithms

In this section, we give a glimpse of our experimental evaluations of additive error and compare it with the best known results. The details and discussion of our empirical evaluations is in supplementary materials. Two important parameters in our bounds are $k$ and $\alpha$ – Hardt and Roth [30] consider a constant $\alpha$. Therefore, we analyze the additive error with respect to the change in $\alpha$ in order to better understand the effect of differential privacy on low space low-rank approximation of matrices. The result of our experiment is presented in Figure 1 ((a)-(d)) with the scale of $y$-axis (accuracy) in logarithmic to better illustrate the accuracy improvement shown by our algorithm. In both these experiments, we see that the additive error incurred by our algorithm is less than the additive error incurred by Hardt and Roth [30]. We note that the matrices are highly incoherent as all the entries are sampled i.i.d. We also consider the role of $k$ in our locally-private algorithm. The results of our experiment in presented in Figure 1 ((e)-(f)). The error of our algorithm is consistently less than the expected error.

# 8 Conclusion

In this paper, we study differentially private low-rank approximation in various settings of practical importance. We give first algorithms with optimal accuracy, space requirements, and runtime for all of these settings. Our results relies crucially on *careful analysis* and our algorithms heavily exploit advance yet inexpensive *post-processing*. Prior to this work, only two known private algorithms for Problem 1 use any form of post-processing for LRF: Hardt and Roth [30] uses simple pruning of entries of a matrix formed in the intermediate step, while Dwork *et al.* [24] uses best rank-$k$ approximation of the privatized matrix. These post-processing either make the algorithm suited only for static matrices or are expensive.

There are few key take aways from this paper: (i) maintaining a differentially private sketches of row space and column space of a matrix already give a sub-optimal accuracy, but this can be significantly improved by careful inexpensive post-processing, and (ii) the structural properties of linear sketches can be carefully exploited to get tight bound on the error. Prior to this work, it was not clear whether the techniques we use in this paper yields low rank approximation (see the comment after Theorem IV.2 and discussion in Section V in Blocki *et al.* [8]). Therefore, we believe our techniques will find use in many related private algorithms as evident by the recent result of Arora et al. [4].

**Acknowledgements.** The author would like to thank Adam Smith for useful feedback on this paper. This research was supported in part by NSF BIGDATA grant IIS-1447700, NSF BIGDATA grant IIS-154648, and NSF BIGDATA grant IIS-1838139.

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
