[Supplementary Material]

# Supplementary Material to "The Price of Privacy for Low-rank Factorization"

## A    Related Work

Low-rank approximation (LRA), where the goal is to output a matrix $\mathbf{B}$ such that $\|\mathbf{A} - \mathbf{B}\|_F$ is close to the optimal LRA, of large data-matrices has received a lot of attention in the recent past in the private as well as the non-private setting. In what follows, we give a brief exposition of works most relevant to this work.

In the non-private setting, previous works have either used random projection or random sampling (at a cost of a small additive error) to give low-rank approximation [3, 21, 31, 33, 27, 43, 55, 82, 85, 86]. Subsequent works [23, 24, 26, 28, 66, 71, 79, 86] achieved a run-time that depends linearly on the input sparsity of the matrix. In a series of works, Clarkson and Woodruff [21, 23] showed space lower bounds and almost matching space algorithms. Recently, Boutsidis *et al.* [17] gave the first space-optimal algorithm for low-rank approximation under turnstile update mode, but they do not optimize for run-time. A optimal space algorithm was recently proposed by Yurtsever *et al.* [99]. Distributed PCA algorithms in the row partition model and arbitrary partition model has been long studied [8, 64, 61, 17, 44, 83, 84, 91].

In the private setting, low-rank approximation LRA has been studied under a privacy guarantee called differential privacy. Differential privacy was introduced by Dwork *et al.* [36]. Since then, many algorithms for preserving differential privacy have been proposed in the literature [38]. All these mechanisms have a common theme: they perturb the output before responding to queries. Recently, Blocki *et al.* [13] and Upadhyay [92] took a complementary approach. They perturb the input reversibly and then perform a random projection of the perturbed matrix.

Blum *et al.* [14] first studied the problem of differentially private LRA in the Frobenius norm. This was improved by Hardt and Roth [47] under the low coherence assumption. Upadhyay [93] later made it a single-pass. Differentially private LRA has been studied in the spectral norm as well by many works [20, 57, 48, 46]. Kapralov and Talwar [57] and Chaudhary *et al.* [20] studied the spectral LRA of a matrix by giving a matching upper and lower bounds for privately computing the top $k$ eigenvectors of a matrix with pure differential privacy (i.e., $\delta = 0$). In subsequent works Hardt and Roth [48] and Hardt and Price [46] improved the approximation guarantee with respect to the spectral norm by using *robust private subspace iteration* algorithm. Recently, Dwork *et al.* [40] gave a tighter analysis of Blum *et al.* [14] to give an optimal approximation to the right singular space, i.e., they gave a LRA for the covariance matrix. Dwork *et al.* [37] first considered streaming algorithms with privacy under the model of *pan-privacy*, where the internal state is known to the adversary. They gave private analogues of known sampling based streaming algorithms to answer various counting tasks. This was followed by results on online private learning [40, 51, 90].

## B    Discussion on Privacy Guarantees, Assumptions, and Relaxation.

The two privacy guarantees considered in this paper have natural reasons to be considered. $\mathsf{Priv}_1$ generalizes the earlier privacy guarantees and captures the setting where any two matrices differ in only one spectrum. Since $\mathsf{Priv}_1$ is defined in terms of the spectrum of matrices, $\mathsf{Priv}_1$ captures one of the natural privacy requirements in all the applications of LRF. $\mathsf{Priv}_2$ is more stronger than $\mathsf{Priv}_1$. To motivate the definition of $\mathsf{Priv}_2$, consider a graph, $\mathcal{G} := (\mathcal{V}, \mathcal{E})$ that stores career information of people in a set $\mathcal{P}$ since their graduation. The vertex set $\mathcal{V}$ is the set of all companies. An edge $e = (u, v) \in \mathcal{E}$ has weight $\sum_{p \in \mathcal{P}}(t_{p,e}/t_p)$, where $t_{p,e}$ is the time for which the person $p$ held a job at $v$ after leaving his/her job at $u$, and $t_p$ is the total time lapsed since his/her graduation. Graphs like $\mathcal{G}$ are useful because the weight on every edge $e = (u, v)$ depends on the number of people who changed their job status from $u$ to $v$ (and the time they spent at $v$). Therefore, data analysts might

want to mine such graphs for various statistics. In the past, graph statistics have been extensively studied for static graph under *edge-level privacy* (see, for e.g., [40, 47, 46, 92, 93]): the presence or absence of a person corresponds to a change in a single edge. On the other hand, in graphs like $\mathcal{G}$, presence or absence of a person would be reflected on many edges. If we use earlier results on edge-level privacy to such graphs, it would lead to either a large additive error or a loss in privacy parameters $\varepsilon, \delta$. $\mathsf{Priv}_2$ is an attempt to understand whether we can achieve any non-trivial guarantee on the additive error without depreciating the privacy parameters.

Our choice to *not make any assumptions* such as symmetry, incoherence, or a bound on the Frobenius norm of the input matrix, as made in previous works [40, 47, 46, 57], allows us to give results as general as possible and cover many practical scenarios not covered by previous works. For example, this allows us to have a unified algorithmic approach for all the settings mentioned earlier in the introduction (see Corollary 2). Moreover, by not making any assumption on the Frobenius norm or coherence of the matrix, we are able to cover many general cases not covered by previous results. For example, adjacency matrices corresponding to graph $\mathcal{G}$ can have arbitrary large Frobenius norm. Such graphs can be distributed over all the companies and updated every time a person changes his job, and the effect is (possibly) reflected on many edges. On the other hand, one would like to protect private information about an individual in $\mathcal{G}$ or data packets in communication network. There are practical learning application such as word2vec where private data is used to build a relationship graph on objects (such as words-to-words, queries-to-urls, etc) also share all these properties. Other applications like recommendation systems, clustering, and network analysis, have their input datasets store dynamically changing private information in the form of $m \times n$ matrices. For example, online video streaming websites that stores users' streaming history, both store private information in the form of matrix which is updated in a turnstile manner and has arbitrary large Frobenius norm. These data are useful for their respective businesses since they can be used for various analyses [30, 62, 77].

On the other hand, in some practical applications, matrices are more structured. One such special case is when the rows of private matrix $\mathbf{A}$ have bounded norm and one would like to approximate $\mathbf{A}^\mathsf{T}\mathbf{A}$. This problem was studied by Dwork *et al.* [40] (see Appendix N.3 for their formal problem statement). We consider the matrices are updated by inserting one row at a time: all the updates at time $\tau \le T$ are of the form $\{i_\tau, \mathbf{A}^{(\tau)}\}$, where $1 \le i_\tau \le m$, $\mathbf{A}^{(\tau)} \in \mathbb{R}^n$, and $i_\tau \ne i_{\tau'}$ for all $\tau \ne \tau'$. We get the following corollary by using $\mathbf{A}^\mathsf{T}\mathbf{A}$ as the input matrix:

**Corollary 2.** *(Theorem 15, informal). Given an $\mathbf{A} \in \mathbb{R}^{m \times n}$ updated by inserting one row at a time such that every row has a bounded norm $1$ and $m > n$. Then there is an $(\varepsilon, \delta)$-differentially private algorithm under $\mathsf{Priv}_2$ that uses $\widetilde{O}(nk\alpha^{-2})$ space, and outputs a rank-$k$ matrix $\mathbf{B}$ such that*
$$\|\mathbf{A}^\mathsf{T}\mathbf{A} - \mathbf{B}\|_F \le (1 + \alpha)\|\mathbf{A}^\mathsf{T}\mathbf{A} - [\mathbf{A}^\mathsf{T}\mathbf{A}]_k\|_F + \widetilde{O}\left(\sqrt{nk}/(\alpha\varepsilon)\right).$$

We do not violate the lower bound of Dwork *et al.* [40] because their lower bound is valid when $\alpha = 0$, which is not possible for low space algorithms due to Theorem 2. Dwork *et al.* [40] bypassed their lower bounds under a stronger assumption known as *singular value separation*: the difference between $k$-th singular value and all $k'$-th singular values for $k' > k$ is at least $\omega(\sqrt{n})$. In other words, our result shows that we do not need singular value separation while using significantly less space—$\widetilde{O}(nk/\alpha^2)$ as compared to $O(n^2)$—if we are ready to pay for a small multiplicative error (this also happens for both the problems studied by Blocki *et al.* [13] though the authors do not mention it explicitly). There are scenarios where our algorithm perform better, such as when $\|\mathbf{A}^\mathsf{T}\mathbf{A} - [\mathbf{A}^\mathsf{T}\mathbf{A}]_k\|_F = O(\sqrt{kn})$ with small singular value separation. For example, a class of rank-$ck$ matrices (for $c \ge 2$) with singular values $\sigma_1, \cdots, \sigma_{ck}$, such that $\sigma_k^2 = \sigma_{k'}^2 + O(\sqrt{n/k})$ for $k \le k' \le ck$, does not satisfy singular value separation. Moreover, $\|\mathbf{A}^\mathsf{T}\mathbf{A} - [\mathbf{A}^\mathsf{T}\mathbf{A}]_k\|_F = O(\sqrt{kn})$ and $\|\mathbf{A}^\mathsf{T}\mathbf{A}\|_F = O(\sqrt{m})$. In this case, we have $\|\mathbf{A}^\mathsf{T}\mathbf{A} - \mathbf{B}\|_F \le O((\alpha\varepsilon)^{-1}\sqrt{kn})$. On the other hand, Dwork *et al.* [40] guarantees $\|\mathbf{A}^\mathsf{T}\mathbf{A} - \mathbf{B}\|_F \le O(\varepsilon^{-1}k\sqrt{n})$. That is, we improve on Dwork *et al.* [40] when $\alpha = \Theta(1)$ for a large class of matrices. Some examples include the matrices considered in the literature of matrix completion (see [74] for details).

## C    Notations and Preliminaries

**Notations** We give a brief exposition of notations and linear algebra to the level required to understand this paper. We refer the readers to standard textbook on this topic for more details [12]. We let $\mathbb{N}$ to

denote the set of natural numbers and $\mathbb{R}$ to denote the set of real numbers. For a real number $x \in \mathbb{R}$, we denote by $|x|$ the absolute value of $x$. We use boldface lowercase letters to denote vectors, for example, $\mathbf{x}$, and $\mathbf{x}_1, \ldots, \mathbf{x}_n$ to denote the entries of $\mathbf{x}$, and bold-face capital letters to denote matrices, for example, $\mathbf{A}$. For two vectors $\mathbf{x}$ and $\mathbf{y}$, we denote by $\langle \mathbf{x}, \mathbf{y} \rangle = \sum_i \mathbf{x}_i \mathbf{y}_i$ the inner product of $\mathbf{x}$ and $\mathbf{y}$. We let $\mathbf{e}_1, \ldots, \mathbf{e}_n$ denote the standard basis vectors in $\mathbb{R}^n$, i.e., $\mathbf{e}_i$ has entries $0$ everywhere except for the position $i$ where the entry is $1$. We denote by $(\mathbf{A} \quad \mathbf{B})$ the matrix formed by appending the matrix $\mathbf{A}$ columnwise with the matrix $\mathbf{B}$. We use the notation $\mathbb{I}_n$ to denote the identity matrix of order $n$ and $\mathbf{0}^{m \times n}$ the all-zero $m \times n$ matrix. Where it is clear from the context, we drop the subscript.

For a matrix $\mathbf{A} \in \mathbb{R}^{m \times n}$, we denote by $\mathbf{A}_{ij}$ the $(i, j)$-th entry of $\mathbf{A}$. We denote by $\mathsf{vec}(\mathbf{A})$ the vector of length $mn$ formed by the entries of the matrix $\mathbf{A}$, i.e., for an $m \times n$ matrix $\mathbf{A}$, the $((i-1)n+j)$-th entry of $\mathsf{vec}(\mathbf{A})$ is $\mathbf{A}_{ij}$, where $1 \le i \le m, 1 \le j \le n$. The *transpose* of a matrix $\mathbf{A}$ is a matrix $\mathbf{B}$ such that $\mathbf{B}_{ij} = \mathbf{A}_{ji}$. We use the notation $\mathbf{A}^\mathsf{T}$ to denote the transpose of a matrix $\mathbf{A}$. For a matrix $\mathbf{A}$, we denote the best $k$-rank approximation of $\mathbf{A}$ by $[\mathbf{A}]_k$, its Frobenius norm by $\|\mathbf{A}\|_F$, and by $\Delta_k := \|\mathbf{A} - [\mathbf{A}]_k\|_F$. For a matrix $\mathbf{A}$, we use the symbol $\mathsf{r}(\mathbf{A})$ to denote its *rank* and $\det(\mathbf{A})$ to denote its *determinant*.

We give a brief exposition of notations and linear algebra to the level required to understand this paper. We refer the readers to standard textbook on this topic for more details [12]. We let $\mathbb{N}$ to denote the set of natural numbers and $\mathbb{R}$ to denote the set of real numbers. For a real number $x \in \mathbb{R}$, we denote by $|x|$ the absolute value of $x$. We use boldface lowercase letters to denote vectors, for example, $\mathbf{x}$, and $\mathbf{x}_1, \ldots, \mathbf{x}_n$ to denote the entries of $\mathbf{x}$, and bold-face capital letters to denote matrices, for example, $\mathbf{A}$. For two vectors $\mathbf{x}$ and $\mathbf{y}$, we denote by $\langle \mathbf{x}, \mathbf{y} \rangle = \sum_i \mathbf{x}_i \mathbf{y}_i$ the inner product of $\mathbf{x}$ and $\mathbf{y}$. We let $\mathbf{e}_1, \ldots, \mathbf{e}_n$ denote the standard basis vectors in $\mathbb{R}^n$, i.e., $\mathbf{e}_i$ has entries $0$ everywhere except for the position $i$ where the entry is $1$. We denote by $(\mathbf{A} \quad \mathbf{b})$ the matrix formed by appending the matrix $\mathbf{A}$ with the vector $\mathbf{b}$. We use the notation $\mathbb{I}_n$ to denote the identity matrix of order $n$ and $\mathbf{0}^{m \times n}$ the all-zero $m \times n$ matrix. Where it is clear from the context, we drop the subscript.

For a $m \times n$ matrix $\mathbf{A}$, we denote by $\mathbf{A}_{ij}$ the $(i, j)$-th entry of $\mathbf{A}$. We denote by $\mathsf{vec}(\mathbf{A})$ the vector of length $mn$ formed by the entries of the matrix $\mathbf{A}$, i.e., for an $m \times n$ matrix $\mathbf{A}$, the $((i-1)n+j)$-th entry of $\mathsf{vec}(\mathbf{A})$ is $\mathbf{A}_{ij}$, where $1 \le i \le m, 1 \le j \le n$. The *transpose* of a matrix $\mathbf{A}$ is a matrix $\mathbf{B}$ such that $\mathbf{B}_{ij} = \mathbf{A}_{ji}$. We use the notation $\mathbf{A}^\mathsf{T}$ to denote the transpose of a matrix $\mathbf{A}$. For a matrix $\mathbf{A}$, we denote the best $k$-rank approximation of $\mathbf{A}$ by $[\mathbf{A}]_k$ and its Frobenius norm by $\|\mathbf{A}\|_F$. For a matrix $\mathbf{A}$, we use the symbol $\mathsf{r}(\mathbf{A})$ to denote its *rank* and $\det(\mathbf{A})$ to denote its *determinant*. A matrix $\mathbf{A}$ is a *non-singular matrix* if $\det(\mathbf{A}) \ne 0$.

## C.1  Definitons

**Definition 2.** *(Differentially Private Low Rank Factorizataion in the General Turnstile Update Model). Let the private matrix $\mathbf{A} \in \mathbb{R}^{m \times n}$ initially be all zeroes. At every time epoch, the curator receives an update in the form of the tuple $\{i, j, \Delta\}$, where $1 \le i \le m, 1 \le j \le n$, and $\Delta \in \mathbb{R}$—each update results in a change in the $(i, j)$-th entry of the matrix $\mathbf{A}$ as follows: $\mathbf{A}_{i,j} \leftarrow \mathbf{A}_{i,j} + \Delta$. Given parameters $0 < \alpha, \beta < 1$ and $\gamma$, and the target rank $k$, the curator is required to output a differentially-private rank-$k$ matrix factorization $\widetilde{\mathbf{U}}_k, \widetilde{\mathbf{\Sigma}}_k$, and $\widetilde{\mathbf{V}}_k$ of $\mathbf{A}$ at the end of the stream, such that, with probability at least $1 - \beta$,*

$$\|\mathbf{A} - \widetilde{\mathbf{U}}_k \widetilde{\mathbf{\Sigma}}_k \widetilde{\mathbf{V}}_k^\mathsf{T}\|_F \le (1 + \alpha)\Delta_k + \gamma,$$

*where $\Delta_k := \|\mathbf{A} - [\mathbf{A}]_k\|_F$ with $[\mathbf{A}]_k$ being the best rank-$k$ approximation of $\mathbf{A}$.*

**Definition 3.** *(Differentially Private Low Rank Factorizataion in the Continual Release Model). Let the private matrix $\mathbf{A} \in \mathbb{R}^{m \times n}$ initially be all zeroes. At every time epoch $t$, the curator receives an update in the form of the tuple $\{i, j, \Delta\}$, where $1 \le i \le m, 1 \le j \le n$, and $\Delta \in \mathbb{R}$—each update results in a change in the $(i, j)$-th entry of the matrix $\mathbf{A}$ as follows: $\mathbf{A}_{i,j}^{(t)} \leftarrow \mathbf{A}_{i,j}^{(t-1)} + \Delta$. Given parameters $0 < \alpha, \beta < 1$ and $\gamma$, and the target rank $k$, the curator is required to output a differentially-private rank-$k$ matrix factorization $\widetilde{\mathbf{U}}_k^{(t)}, \widetilde{\mathbf{\Sigma}}_k^{(t)}$, and $\widehat{\mathbf{V}}_k^{(t)}$ of $\mathbf{A}$ at every time epoch, such that, with probability at least $1 - \beta$,*

$$\|\mathbf{A}^{(t)} - \widetilde{\mathbf{U}}_k^{(t)} \widetilde{\mathbf{\Sigma}}_k^{(t)} (\widetilde{\mathbf{V}}_k^{(t)})^\mathsf{T}\|_F \le (1 + \alpha)\Delta_k^{(t)} + \gamma,$$

*where $\Delta_k^{(t)} := \|\mathbf{A}^{(t)} - [\mathbf{A}^{(t)}]_k\|_F$ with $[\mathbf{A}^{(t)}]_k$ being the best rank-$k$ approximation of $\mathbf{A}^{(t)}$.*

**Definition 4.** *(Local Differentially Private Principal Component Analysis Problem). Given an $m \times n$ matrix $\mathbf{A}$ distributed among $m$ users where each user holds a row of the matrix $\mathbf{A}$, a rank parameter $k$, and an accuracy parameter $0 < \alpha < 1$, design a local differentially private algorithm which, upon termination, outputs an orthonormal matrix $\mathbf{U}$ such that*

$$\|\mathbf{A} - \mathbf{U}\mathbf{U}^\mathsf{T}\mathbf{A}\|_F \leq (1 + \alpha)\|\mathbf{A} - [\mathbf{A}]_k\|_F + \gamma$$

*with probability at least $1 - \beta$, and the communication cost of the algorithm is as small as possible. Furthermore, the algorithm should satisfies $(\varepsilon, \delta)$-LDP.*

In this paper, we use various concepts and results from the theory of random projections, more specifically the Johnson-Lindenstrauss transform and its variants.

**Definition 5.** *Let $\alpha, \delta > 0$. A distribution $\mathcal{D}$ over $t \times n$ random matrices satisfies $(\alpha, \beta)$-Johnson Lindenstrauss property (JLP) if, for any unit vector $\boldsymbol{x} \in \mathbb{R}^n$, we have*

$$\mathsf{Pr}_{\boldsymbol{\Phi} \sim \mathcal{D}}[\|\boldsymbol{\Phi}\boldsymbol{x}\|_2^2 \in (1 \pm \alpha)] \geq 1 - \beta.$$

**Definition 6.** *A distribution $\mathcal{D}_R$ of $t \times m$ matrices satisfies $(\alpha, \delta)$-subspace embedding for generalized regression if it has the following property: for any matrices $\mathbf{P} \in \mathbb{R}^{m \times n}$ and $\mathbf{Q} \in \mathbb{R}^{m \times n'}$ such that $\mathsf{r}(\mathbf{P}) \leq r$, with probability $1 - \delta$ over $\boldsymbol{\Phi} \sim \mathcal{D}_R$, if*

$$\widetilde{\mathbf{X}} = \operatorname*{argmin}_{\mathbf{X}} \|\boldsymbol{\Phi}(\mathbf{P}\mathbf{X} - \mathbf{Q})\|_F \quad and \quad \widehat{\mathbf{X}} = \operatorname*{argmin}_{\mathbf{X} \in \mathbb{R}^{n \times n'}} \|\mathbf{P}\mathbf{X} - \mathbf{Q}\|_F,$$

*then $\|\mathbf{P}\widetilde{\mathbf{X}} - \mathbf{Q}\|_F \leq (1 + \alpha)\|\mathbf{P}\widehat{\mathbf{X}} - \mathbf{Q}\|_F$.*

**Definition 7.** *A distribution $\mathcal{D}_A$ over $v \times m$ matrices satisfies $(\alpha, \delta)$-affine subspace embedding if it has the following property: for any matrices $\mathbf{D} \in \mathbb{R}^{m \times n}$ and $\mathbf{E} \in \mathbb{R}^{m \times n'}$ such that $\mathsf{r}(\mathbf{D}) \leq r$, with probability $1 - \delta$ over $\mathbf{S} \sim \mathcal{D}_A$, simultaneously for all $\mathbf{X} \in \mathbb{R}^{n \times n'}$, $\|\mathbf{S}(\mathbf{D}\mathbf{X} - \mathbf{E})\|_F^2 = (1 \pm \alpha)\|\mathbf{D}\mathbf{X} - \mathbf{E}\|_F^2$.*

We use the symbol $\mathcal{D}_R$ to denote a distribution that satisfies $(\alpha, \delta)$-subspace embedding for generalized regression and $\mathcal{D}_A$ to denote a distribution that satisfies $(\alpha, \delta)$-affine subspace embedding.

## C.2 Linear Algebra

A matrix is called a *diagonal matrix* if the non-zero entries are all along the principal diagonal. An $m \times m$ matrix $\mathbf{A}$ is a *unitary matrix* if $\mathbf{A}^\mathsf{T}\mathbf{A} = \mathbf{A}\mathbf{A}^\mathsf{T} = \mathbb{I}_m$. Additionally, if the entries of the matrix $\mathbf{A}$ are real, then such a matrix is called an *orthogonal matrix*. For an $m \times m$ matrix $\mathbf{A}$, the *trace* of $\mathbf{A}$ is the sum of its diagonal elements. We use the symbol $\mathrm{Tr}(\mathbf{A})$ to denote the trace of matrix $\mathbf{A}$. We use the symbol $\det(\mathbf{A})$ to denote the determinant of matrix $\mathbf{A}$.

Let $\mathbf{A}$ be an $m \times m$ matrix. Its *singular values* are the eigenvalues of the matrix $\sqrt{\mathbf{A}^\mathsf{T}\mathbf{A}}$. The eigenvalues of the matrix $\sqrt{\mathbf{A}^\mathsf{T}\mathbf{A}}$ are real because $\mathbf{A}^\mathsf{T}\mathbf{A}$ is a symmetric matrix and has a well-defined *spectral decomposition*[1] [12].

The *singular-value decomposition* (SVD) of an $m \times n$ rank-$r$ matrix $\mathbf{A}$ is a decomposition of $\mathbf{A}$ as a product of three matrices, $\mathbf{A} = \mathbf{U}\boldsymbol{\Sigma}\mathbf{V}^\mathsf{T}$ such that $\mathbf{U} \in \mathbb{R}^{m \times r}$ and $\mathbf{V} \in \mathbb{R}^{n \times r}$ have orthonormal columns and $\boldsymbol{\Sigma} \in \mathbb{R}^{r \times r}$ is a diagonal matrix with singular values of $\mathbf{A}$ on its diagonal. One can equivalently write it in the following form:

$$\mathbf{A} = \sum_{i=1}^{\mathsf{r}(\mathbf{A})} \sigma_i \mathbf{u}_i \mathbf{v}_i^\mathsf{T},$$

where $\mathbf{u}_i$ is the $i$-th column of $\mathbf{U}$, $\mathbf{v}_i$ is the $i$-th column of $\mathbf{V}$, and $\sigma_i$ is the $i$-th diagonal entry of $\Sigma$.

One can derive a lot of things from the singular value decomposition of a matrix. For example,

1. The *Moore-Penrose pseudo-inverse* of a matrix $\mathbf{A} = \mathbf{U}\mathbf{\Sigma}\mathbf{V}^{\mathsf{T}}$ is denoted by $\mathbf{A}^{\dagger}$ and has a SVD $\mathbf{A}^{\dagger} = \mathbf{V}\mathbf{\Sigma}^{\dagger}\mathbf{U}^{\mathsf{T}}$, where $\mathbf{\Sigma}^{\dagger}$ consists of inverses of only non-zero singular values of $\mathbf{A}$. In other words,

$$\mathbf{A}^{\dagger} = \sum_{i=1}^{r(\mathbf{A})} \frac{1}{\sigma_i} \mathbf{u}_i \mathbf{v}_i^{\mathsf{T}},$$

where $r(\mathbf{A})$ is the number of non-zero singular values of $\mathbf{A}$.

2. Let $\sigma_1 \geq \cdots \geq \sigma_k \geq \cdots \geq \sigma_{r(\mathbf{A})}$ be the singular values of $\mathbf{A}$. Then

$$[\mathbf{A}]_k = \sum_{i=1}^{k} \sigma_i \mathbf{u}_i \mathbf{v}_i^{\mathsf{T}},$$

3. The trace of a matrix $\mathbf{A}$ can be represented in form of the singular values of $\mathbf{A}$ as follows: $\mathrm{Tr}(\mathbf{A}) = \sum_i \sigma_i$. Similarly, the determinant of a matrix $\mathbf{A}$ is $\det(\mathbf{A}) = \prod_i \sigma_i$. Moreover, the Frobenius norm of $\mathbf{A}$ is $\sum_i \sigma_i^2$.

We use few unitary matrices, which we define next. Let $N$ be a power of 2. A Walsh-Hadamard matrix of order $N$ is an $N \times N$ matrix formed recursively as follows:

$$\mathbf{W}_N = \frac{1}{\sqrt{2}} \begin{pmatrix} \mathbf{W}_{N/2} & \mathbf{W}_{N/2} \\ \mathbf{W}_{N/2} & -\mathbf{W}_{N/2} \end{pmatrix} \qquad \mathbf{W}_1 := (1).$$

A Walsh-Hadamard matrix is a unitary matrix. We often drop the subscript wherever it is clear from the context. A *randomized Walsh-Hadamard matrix* is a matrix product of a Walsh-Hadamard matrix and a random diagonal matrix with entries $\pm 1$ picked according to the following probability distribution:

$$\mathsf{Pr}[X = 1] = \mathsf{Pr}[X = -1] = 1/2.$$

A *discrete Fourier matrix* of order $n$ is an $n \times n$ matrix such that the $(i, j)$-th entry is $\omega^{(i-1)(j-1)}$, where $\omega$ is the $n$-th root of unity, i.e., $\omega = e^{-2\pi\iota/n}$.

## C.3  Gaussian Distribution

Given a random variable $x$, we denote by $\mathcal{N}(\mu, \rho^2)$ the fact that $x$ has a normal Gaussian distribution with mean $\mu$ and variance $\rho^2$. The Gaussian distribution is invariant under affine transformation, i.e., if $X \sim \mathcal{N}(\mu_x, \sigma_x)$ and $Y \sim \mathcal{N}(\mu_y, \sigma_y)$, then $Z = aX + bY$ has the distribution $Z \sim \mathcal{N}(a\mu_x + b\mu_y, a\sigma_x^2 + b\sigma_y^2)$. This is also called the *rotational invariance* of Gaussian distribution. By simple computation, one can verify that the tail of a standard Gaussian variable decays exponentially. More specifically, for a random variable $X \sim \mathcal{N}(0, 1)$, we have $\mathsf{Pr}\left[|X| > t\right] \leq 2e^{-t^2/2}$.

Our proof uses an analysis of multivariate Gaussian distribution. The multivariate Gaussian distribution is a generalization of univariate Gaussian distribution. Let $\mu$ be an $N$-dimensional vector. An $N$-dimensional multivariate random variable, $\mathbf{x} \sim \mathcal{N}(\mu, \mathbf{\Lambda})$, where $\mathbf{\Lambda} = \mathbb{E}[(\mathbf{x}-\mu)(\mathbf{x}-\mu)^{\mathsf{T}}]$ is the $N \times N$ covariance matrix, has the probability density function given by $\mathsf{PDF}_{\mathbf{X}}(\mathbf{x}) := \frac{e^{-\mathbf{x}^{\mathsf{T}}\mathbf{\Lambda}^{\dagger}\mathbf{x}/2}}{\sqrt{(2\pi)^{r(\mathbf{\Sigma})}\det(\mathbf{\Lambda})}}$. If $\mathbf{\Lambda}$ has a non-trivial kernel space, then the multivariate distribution is undefined. However, in this paper, all our covariance matrices have only trivial kernel. Multivariate Gaussian distributions is invariant under affine transformation, i.e., if $\mathbf{y} = \mathbf{A}\mathbf{x} + \mathbf{b}$, where $\mathbf{A} \in \mathbb{R}^{M \times N}$ is a rank-$M$ matrix and $\mathbf{b} \in \mathbb{R}^M$, then $\mathbf{y} \sim \mathcal{N}(\mathbf{A}\mu + \mathbf{b}, \mathbf{A}\mathbf{\Lambda}\mathbf{A}^{\mathsf{T}})$.

# D  Basic Results Used in This Paper

Our proofs uses various concepts and known results about random projections, pseudo-inverse of matrices and gaussian distribution. In this section, we cover them up to the level of exposition required to understand this paper. We refer to the excellent book by Bhatia [12] for more exposition on pseudo-inverses, and Woodruff [98].

## D.1 Random Projections.

Random projection has been used in computer science for a really long time. Some partial application includes metric and graph embeddings [16, 65], computational speedups [86, 95], machine learning [9, 87], nearest-neighbor search [50, 80], and compressed sensing [10].

In this paper, we use random projections that satisfy Definition 6 and Definition 7. An example distribution $\mathcal{D}_R$ with $t = O(\alpha^{-2} \log(1/\delta))$ is the distribution of random matrices whose entries are sampled i.i.d. from $\mathcal{N}(0, 1/t)$. Recently, Clarkson and Woodruff [22] proposed other distribution of random matrices that satisfies $(\alpha, \delta)$-*subspace embedding for generalized regression* and $(\alpha, \delta)$-*affine subspace embedding*. They showed the following:

**Lemma 1.** *([22, Lem 41, Lem 46]) There is a distribution $\mathcal{D}_R$ over $\mathbb{R}^{t \times m}$ such that satisfies*

**(i)** *if $t = O(\alpha^{-2} \log^2 m)$, then for $\mathbf{\Phi} \sim \mathcal{D}_R$ and any $m \times n$ matrix $\mathbf{Q}$, $\|\mathbf{\Phi Q}\|_F^2 = (1 \pm \alpha)\|\mathbf{Q}\|_F^2$, and*

**(ii)** *if $t = O(r/\alpha \log(r/\delta))$, then $\mathcal{D}_R$ satisfies $(\alpha, \delta)$-subspace embedding for generalized regression for $\mathbf{P}$ and $\mathbf{Q}$.*

*Further, for any matrix $\mathbf{L} \in \mathbb{R}^{m \times n}$, $\mathbf{\Phi L}$ can be computed in $O(\mathrm{nn}(\mathbf{L}) + tn \log t)$ time. Here $\mathbf{P}$, $\mathbf{Q}$, and $r$ are as in Definition 6.*

**Lemma 2.** *[22, Thm 39, Thm 42]) There exists a distribution $\mathcal{D}_A$ over $\mathbb{R}^{v \times m}$ such that*

**(i)** *if $v = \Theta(\alpha^{-2})$, then for $\mathbf{S} \sim \mathcal{D}_A$ and any $m \times d$ matrix $\mathbf{D}$, $\|\mathbf{SD}\|_F^2 = (1 \pm \alpha)\|\mathbf{D}\|_F^2$.*

**(ii)** *if $v = O(p/\alpha^2 \log(p/\delta))$, then $\mathcal{D}_A$ satisfies $(\alpha, \delta)$-affine embedding for $\mathbf{D}$ and $\mathbf{E}$.*

*Further, for any matrix $\mathbf{L} \in \mathbb{R}^{m \times n}$, $\mathbf{SL}$ can be computed in $O(\mathrm{nn}(\mathbf{L}) + nv \log v)$ time. Here $\mathbf{E}$, $\mathbf{D}$, and $p$ are as in Definition 7.*

In the theorems above, $\mathbf{\Phi}$ and $\mathbf{S}$ are oblivious to the matrices $\mathbf{P}, \mathbf{Q}, \mathbf{D}$, and $\mathbf{E}$. That is, we design the distribution $\mathcal{D}_A$ over linear maps such that for any fixed matrices $\mathbf{D}, \mathbf{E}$, if we chose $\mathbf{S} \sim \mathcal{D}_A$, then $\mathbf{S}$ is an $(\alpha, \beta)$-affine embedding for $\mathbf{D}, \mathbf{E}$. Similarly, we design the distribution $\mathcal{D}_R$ over linear maps such that for any fixed matrices $\mathbf{P}, \mathbf{Q}$, if we chose $\mathbf{\Phi} \sim \mathcal{D}_A$, then $\mathbf{\Phi}$ is an $(\alpha, \beta)$ embedding for $\mathbf{P}, \mathbf{Q}$.

## D.2 Differential privacy

Differential privacy is a very robust guarantee of privacy which makes confidential data available widely for accurate analysis while still preserving the privacy of individual data. Achieving these two requirements at the same time seems paradoxical. On one hand, we do not wish to leak information about an individual. On the other hand, we want to answer the query on the entire database as accurately as possible. This makes designing differentially private mechanisms challenging.

### D.2.1 Robustness of Differential Privacy

One of the key features of differential privacy is that it is preserved under arbitrary post-processing, i.e., an analyst, without additional information about the private database, cannot compute a function that makes an output less differentially private. In other words,

**Lemma 3.** *(Dwork et al. [35]). Let $\mathfrak{M}(\mathbf{D})$ be an $(\alpha, \beta)$-differential private mechanism for a database $\mathbf{D}$, and let $h$ be any function, then any mechanism $\mathfrak{M}' := h(\mathfrak{M}(\mathbf{D}))$ is also $(\alpha, \beta)$-differentially private for the same set of tasks.*

*Proof.* Let $\mathfrak{M}$ be a differentially private mechanism. Let $\mathrm{range}(\mathfrak{M})$ denote the range the of $\mathfrak{M}$. Let $R$ be the range of the function $h(\cdot)$. Without loss of generality, we assume that $h(\cdot) : \mathrm{range}(\mathfrak{M}) \to \mathcal{R}$ is a deterministic function. This is because any randomized function can be decomposed into a convex combination of deterministic function, and a convex combination of differentially private mechanisms is differentially private. Fix any pair of neighbouring data-sets $\mathbf{DB}$ and $\widetilde{\mathbf{DB}}$ and an event $S \subseteq \mathcal{R}$.

Let $T = \{y \in \text{range}(\mathfrak{M}) : f(r) \in S\}$. Then

$$\Pr[f(\mathfrak{M}(\mathbf{DB})) \in S] = \Pr[\mathfrak{M}(\mathbf{DB}) \in T]$$
$$\leq \exp(\alpha)\Pr[\mathfrak{M}(\widetilde{\mathbf{DB}}) \in T] + \beta$$
$$= \exp(\alpha)\Pr[f(\mathfrak{M}(\widetilde{\mathbf{DB}})) \in S] + \beta.$$

$\square$

### D.2.2 Composition

Before we begin, we discuss what does it mean by the term "composition" of differentially private mechanism. The composition that we consider covers the following two cases:

1. Repeated use of differentially private mechanism on the same database.
2. Repeated use of differentially private mechanism on different database that might contain information relating to a particular individual.

The first case covers the case when we wish to use the same mechanism multiple times while the second case covers the case of cumulative loss of privacy of a single individual whose data might be spread across many databases.

It is easy to see that the composition of pure differentially private mechanisms yields another pure differentially private mechanism, i.e., composition of an $(\alpha_1, 0)$-differentially private and an $(\alpha_2, 0)$-differentially private mechanism results in an $(\alpha_1 + \alpha_2, 0)$-differentially private mechanism. In other words, the privacy guarantee depreciates linearly with the number of compositions. In the case of approximate differential privacy, we can improve on the degradation of $\alpha$ parameter at the cost of slight depreciation of the $\beta$ factor. We use this strengthening in our proofs. In our proofs of differential privacy, we prove that each row of the published matrix preserves $(\alpha_0, \beta_0)$-differential privacy for some appropriate $\alpha_0, \beta_0$, and then invoke a composition theorem by Dwork, Rothblum, and Vadhan [39] to prove that the published matrix preserves $(\alpha, \beta)$-differential privacy. The following theorem is the composition theorem that we use.

**Theorem 4.** (Dwork *et al.* [39]). *Let* $\alpha_0, \beta_0 \in (0, 1)$, *and* $\beta' > 0$. *If* $\mathfrak{M}_1, \cdots, \mathfrak{M}_\ell$ *are each* $(\alpha, \beta)$-*differential private mechanism, then the mechanism* $\mathfrak{M}(\mathbf{D}) := (\mathfrak{M}_1(\mathbf{D}), \cdots, \mathfrak{M}_\ell(\mathbf{D}))$ *releasing the concatenation of each algorithm is* $(\alpha', \ell\beta + 0 + \beta')$-*differentially private for* $\alpha' < \sqrt{2\ell \ln(1/\beta')}\alpha_0 + 2\ell\alpha_0^2$.

A proof of this theorem could be found in [38, Chapter 3].

**Gaussian Mechanism.** The Gaussian variant of the Laplace mechanism was proven to preserve differential privacy by Dwork *et al.* [35] in a follow-up work. Let $f(\cdot)$ be a function from a class of $\Delta$-sensitive functions. The Gaussian mechanism is

$$\mathfrak{M}(\mathbf{D}, f(\cdot), \alpha) := f(\mathbf{D}) + (X_1, \cdots, X_k), \text{ where } X_i \sim \mathcal{N}\left(0, \frac{\Delta^2}{\epsilon^2}\log(1.25/\delta)\right).$$

Dwork *et al.* [35] proved the following.

**Theorem 5.** *(Gaussian mechanism [35].) Let* $\boldsymbol{x}, \boldsymbol{y} \in \mathbb{R}^n$ *be any two vectors such that* $\|\boldsymbol{x} - \boldsymbol{y}\|_2 \leq c$. *Let* $\rho = c\varepsilon^{-1}\sqrt{\log(1/\delta)}$ *and* $\mathbf{g} \sim \mathcal{N}(0, \rho^2)^n$ *be a vector with each entries sampled i.i.d. Then for any* $\mathbf{s} \subset \mathbb{R}^n$, $\Pr[\boldsymbol{x} + \mathbf{g} \in \mathbf{s}] \leq e^\varepsilon \Pr[\boldsymbol{y} + \mathbf{g} \in \mathbf{s}] + \delta$.

### D.3 Properties of Gaussian distribution.

We need the following property of a random Gaussian matrices.

**Fact 1.** *([54, 86]) Let* $\mathbf{P} \in \mathbb{R}^{m \times n}$ *be a matrix of rank $r$ and* $\mathbf{Q} \in \mathbb{R}^{m \times n'}$ *be an* $m \times n'$ *matrix. Let* $\mathcal{D}$ *be a distribution of matrices over* $\mathbb{R}^{t \times n}$ *with entries sampled i.i.d. from* $\mathcal{N}(0, 1/t)$. *Then there exists a* $t = O(r/\alpha \log(r/\beta))$ *such that* $\mathcal{D}$ *is an* $(\alpha, \beta)$-*subspace embedding for generalized regression.*

**Lemma 4.** *Let* $\mathbf{N} \sim \mathcal{N}(0, \rho^2)^{m \times n}$ *Gaussian matrix. Then with probability* 99/100, $\|\mathbf{N}\|_F = O(\rho\sqrt{mn})$.

*Proof.* The lemma follows from the following computation.

$$\mathbb{E}[\|\mathbf{C\Phi}\|_F^2] = \mathbb{E}\left[\sum_{i,j}(\widetilde{\mathbf{N}}_1)_{ij}^2\right] = \sum_{i,j}\mathbb{E}[(\widetilde{\mathbf{N}}_1)_{ij}^2] = mn\rho^2.$$

The result follows using Markov's inequality. □

## D.4 Properties of pseudo-inverse of a matrix.

We need the following results about product of pseudo-inverse in the proof of Lemma 5 and Lemma 22.

**Fact 2.** *If $\mathbf{A}$ has a left-inverse, then $\mathbf{A}^\dagger = (\mathbf{A}^\mathsf{T}\mathbf{A})^{-1}\mathbf{A}^\mathsf{T}$ and if $\mathbf{A}$ has right-inverse, then $\mathbf{A}^\dagger = \mathbf{A}^\mathsf{T}(\mathbf{A}\mathbf{A}^\mathsf{T})^{-1}$.*

**Theorem 6.** *Let $\mathbf{A}$ and $\mathbf{B}$ be conforming matrices and either,*

1. *$\mathbf{A}$ has orthonormal columns (i.e., $\mathbf{A}^\mathsf{T}\mathbf{A}$ is an identity matrix) or,*

2. *$\mathbf{B}$ has orthonormal rows (i.e., $\mathbf{B}\mathbf{B}^\mathsf{T}$ is an identity matrix),*

3. *$\mathbf{A}$ has all columns linearly independent (full column rank) and $\mathbf{B}$ has all rows linearly independent (full row rank) or,*

4. *$\mathbf{B} = \mathbf{A}^\mathsf{T}$ (i.e., $\mathbf{B}$ is the conjugate transpose of $\mathbf{A}$),*

*then $(\mathbf{AB})^\dagger = \mathbf{B}^\dagger\mathbf{A}^\dagger$.*

We use the following variant of Pythagorean theorem in the proof of Lemma 6.

**Theorem 7.** *(Pythagorean theorem). Let $\mathbf{A}$ and $\mathbf{B}$ be two matrices such that $\mathbf{A}^\mathsf{T}\mathbf{B}$ is an zero matrix. Then for any $\mathbf{C} = \mathbf{A} + \mathbf{B}$, we have $\|\mathbf{C}\|_F^2 = \|\mathbf{A}\|_F^2 + \|\mathbf{B}\|_F^2$.*

## D.5 Linear Algebraic Results Used in This Paper

We also need the following results for the privacy proof.

**Theorem 8.** *(Lidskii Theorem [12]). Let $\mathbf{A}, \mathbf{B}$ be $n \times n$ Hermittian matrices. Then for any choice of indices $1 \le i_1 \le \cdots \le i_k \le n$,*

$$\sum_{j=1}^{k}\lambda_{i_j}(\mathbf{A}+\mathbf{B}) \le \sum_{j=1}^{k}\lambda_{i_j}(\mathbf{A}) + \sum_{j=1}^{k}\lambda_{i_j}(\mathbf{B}),$$

*where $\{\lambda_i(\mathbf{A})\}_{i=1}^n$ are the eigen-values of $\mathbf{A}$ in decreasing order.*

**Theorem 9.** *(Weyl's Pertubation Theorem [12]). For any $m \times n$ matrices $\mathbf{P}, \mathbf{Q}$, we have $|\sigma_i(\mathbf{P}+\mathbf{Q}) - \sigma_i(\mathbf{P})| \le \|\mathbf{Q}\|_2$, where $\sigma_i(\cdot)$ denotes the $i$-th singular value and $\|\mathbf{Q}\|_2$ is the spectral norm of the matrix $\mathbf{Q}$.*

We use the notation $\mathsf{Rad}(p)$ to denote a distribution with support $\pm 1$ such that $+1$ is sampled with probability $p$ and $-1$ is sampled with probability $1 - p$. An $n \times n$ Walsh-Hadamard matrix $\mathbf{H}_n$ is constructed recursively as follows:

$$\mathbf{H}_n = \begin{pmatrix} \mathbf{H}_{n/2} & \mathbf{H}_{n/2} \\ \mathbf{H}_{n/2} & -\mathbf{H}_{n/2} \end{pmatrix} \text{ and } \mathbf{H}_1 := 1.$$

A randomized Walsh-Hadamard matrix $\mathbf{W}_n$ is formed by multiplying $\mathbf{H}_n$ with a diagonal matrix whose diagonal entries are picked i.i.d. from $\mathsf{Rad}(1/2)$. We drop the subscript $n$ where it is clear from the context. A subsampled randomized Hadamard matrix is construct by multiplying $\mathbf{\Pi}_{1..r}$ from the left to a randomized Hadamard matrix, where $\mathbf{\Pi}_{1..r}$ is the matrix formed by the first $r$ rows of a random permutation matrix.

**Lemma 5.** *Let $\mathbf{S}$ be a $v \times m$ subsampled randomized Hadamard matrix, where $v \le m$ and $\mathbf{N} \in \mathbb{R}^{v \times n}$. Then we have,*

$$\|\mathbf{S}^\dagger\mathbf{N}_2\|_F = \|\mathbf{N}_2\|_F.$$

*Proof.* One way to look at the action of $\mathbf{S}$ when it is a subsampled Hadamard transform is that it is a product of matrices $\mathbf{W}$ and $\mathbf{\Pi}_{1..r}$, where $\mathbf{\Pi}_{1..r}$ is the matrix formed by the first $r$ rows of a random permutation matrix and $\mathbf{W}$ is a randomized Walsh-Hadamard matrix formed by multiplying a Walsh-Hadamard matrix with a diagonal matrix whose non-zero entries are picked i.i.d. from $\mathsf{Rad}(1/2)$.

Since $\mathbf{WD}$ has orthonormal rows, $\mathbf{S}^\dagger = (\mathbf{\Pi}_{1..v}\mathbf{WD})^\dagger = (\mathbf{WD})^\mathsf{T}(\mathbf{\Pi}_{1..v})^\dagger$. This implies

$$\|\mathbf{S}^\dagger\mathbf{N}\|_F = \|(\mathbf{\Pi}_{1..v}\mathbf{WD})^\dagger\mathbf{N}\|_F = \|(\mathbf{WD})^\mathsf{T}\mathbf{\Pi}_{1..v}^\dagger\mathbf{N}\|_F$$
$$= \|\mathbf{\Pi}_{1..v}^\dagger\mathbf{N}\|_F.$$

Using the fact that $\mathbf{\Pi}_{1..v}$ is a full row rank matrix and $\widehat{\mathbf{\Pi}}_{1..v}\widehat{\mathbf{\Pi}}_{1..v}^\mathsf{T}$ is an identity matrix, we have $\widehat{\mathbf{\Pi}}_{1..v}^\dagger = \widehat{\mathbf{\Pi}}_{1..v}^\mathsf{T}(\widehat{\mathbf{\Pi}}_{1..v}\mathbf{\Pi}_{1..v}^\mathsf{T})^{-1} = \widehat{\mathbf{\Pi}}_{1..v}^\mathsf{T}$. The result follows. $\qquad\square$

We reprove the following theorem of Boutsidis *et al.* [17]. Our proof allows us a tighter control on the intermediate results which helps us to get a tighter bound on the additive error.

**Theorem 10.** *Let $\mathcal{D}_R$ be an $(\alpha, \delta)$-subspace embedding for generalized regression (Definition 6). Then with probability $1 - 2\delta$ over $\mathbf{\Phi}^\mathsf{T} \sim \mathcal{D}_R$ and $\mathbf{\Psi} \sim \mathcal{D}_R$, for any arbitrary $m \times n$ matrix $\mathbf{A}$,*

$$\min_{\mathbf{X}, r(\mathbf{X})\leq k}\|\mathbf{A}\mathbf{\Phi}\mathbf{X}\mathbf{\Psi}\mathbf{A} - \mathbf{A}\|_F \leq (1+\alpha)^2\|\mathbf{A} - [\mathbf{A}]_k\|_F. \tag{4}$$

Our proof uses two optimization problems and uses the solution to those optimization problem in a clever way. We feel that our proof is simpler. It also has explicit solutions to the two optimization problems, which makes it easy to extend to the case of private low-rank factorization and get a tight bound.

*Proof.* Let $[\mathbf{A}]_k = \mathbf{U}_k\mathbf{\Sigma}_k\mathbf{V}_k^\mathsf{T}$. We will use Lemma 1 to prove the theorem. Set $\mathbf{\Phi} = \mathbf{\Phi}^\mathsf{T}, \mathbf{P} = [\mathbf{A}]_k^\mathsf{T}$, $\mathbf{Q} = \mathbf{A}^\mathsf{T}$. Then for $\widetilde{\mathbf{X}} = \mathrm{argmin}_\mathbf{X}\|\mathbf{\Phi}^\mathsf{T}([\mathbf{A}]_k^\mathsf{T}\mathbf{X} - \mathbf{A}^\mathsf{T})\|_F$, we have with probability $1 - \delta$ over $\mathbf{\Phi}^\mathsf{T} \sim \mathcal{D}_R$,

$$\|[\mathbf{A}]_k^\mathsf{T}\widetilde{\mathbf{X}} - \mathbf{A}^\mathsf{T}\|_F \leq (1+\alpha)\min_\mathbf{X}\|[\mathbf{A}]_k^\mathsf{T}\mathbf{X} - \mathbf{A}^\mathsf{T}\|_F$$
$$\leq (1+\alpha)\|[\mathbf{A}]_k^\mathsf{T} - \mathbf{A}^\mathsf{T}\|_F$$

where the last inequality follows by setting $\mathbf{X} = \mathbf{U}_k\mathbf{U}_k^\mathsf{T}$. Here $\widetilde{\mathbf{X}} = (\mathbf{\Phi}^\mathsf{T}[\mathbf{A}]_k^\mathsf{T})^\dagger(\mathbf{A}\mathbf{\Phi})^\mathsf{T}$. Since Frobenius norm is preserved under transpose, we have by substituting the value of $\widetilde{\mathbf{X}}$,

$$\|\mathbf{A}\mathbf{\Phi}([\mathbf{A}]_k\mathbf{\Phi})^\dagger[\mathbf{A}]_k - \mathbf{A}\|_F \leq (1+\alpha)\|\mathbf{A} - [\mathbf{A}]_k\|_F. \tag{5}$$

We now use Lemma 1 on the following regression problem:

$$\min_\mathbf{X}\|\mathbf{W}\mathbf{X} - \mathbf{A}\|_F, \quad \text{where} \quad \mathbf{W} = \mathbf{A}\mathbf{\Phi}([\mathbf{A}]_k\mathbf{\Phi})^\dagger.$$

Let $\widehat{\mathbf{X}} = \mathrm{argmin}_\mathbf{X}\|\mathbf{\Psi}(\mathbf{W}\mathbf{X} - \mathbf{A})\|_F$. Since $[\mathbf{A}]_k$ has rank $k$, Lemma 1 and equation (5) gives with probability $1 - \delta$ over $\mathbf{\Psi} \sim \mathcal{D}_R$

$$\|\mathbf{W}\widehat{\mathbf{X}} - \mathbf{A}\|_F = \|\mathbf{A}\mathbf{\Phi}([\mathbf{A}]_k\mathbf{\Phi})^\dagger\widehat{\mathbf{X}} - \mathbf{A}\|_F$$
$$\leq (1+\alpha)\min_\mathbf{X}\|(\mathbf{A}\mathbf{\Phi})([\mathbf{A}]_k\mathbf{\Phi})^\dagger\mathbf{X} - \mathbf{A}\|_F$$
$$\leq (1+\alpha)\|\mathbf{A}\mathbf{\Phi}([\mathbf{A}]_k\mathbf{\Phi})^\dagger[\mathbf{A}]_k - \mathbf{A}\|_F$$
$$\leq (1+\alpha)^2\|\mathbf{A} - [\mathbf{A}]_k\|_F.$$

Substituting the value of $\widehat{\mathbf{X}} = (\mathbf{\Psi}\mathbf{W})^\dagger\mathbf{\Psi}\mathbf{A}$, with probability $1 - 2\delta$ over $\mathbf{\Phi}^\mathsf{T}, \mathbf{\Psi} \sim \mathcal{D}_R$, we have

$$\|\mathbf{A}\mathbf{\Phi}([\mathbf{A}]_k\mathbf{\Phi})^\dagger(\mathbf{\Psi}\mathbf{A}\mathbf{\Phi}([\mathbf{A}]_k\mathbf{\Phi})^\dagger)^\dagger\mathbf{\Psi}\mathbf{A} - \mathbf{A}\|_F \leq (1+\alpha)^2\|\mathbf{A} - [\mathbf{A}]_k\|_F. \tag{6}$$

Since $([\mathbf{A}]_k\mathbf{\Phi})^\dagger(\mathbf{\Psi}\mathbf{A}\mathbf{\Phi}([\mathbf{A}]_k\mathbf{\Phi})^\dagger)^\dagger$ has rank at most $k$, this completes the proof because $\mathbf{\Phi}([\mathbf{A}]_k\mathbf{\Phi})^\dagger)^\dagger\mathbf{\Psi}$ is a rank-$k$ matrix. $\qquad\square$

We prove the following key lemma, which can be seen as a generalization of one of the previous results of Clarkson and Woodruff [21]. This lemma would be required in proving all our results.

**Lemma 6.** *Let $\mathbf{R}$ be a matrix with orthonormal rows and $\mathbf{C}$ have orthonormal columns. Then*

$$\min_{\mathbf{X}, r(\mathbf{X})=k} \|\mathbf{CXR} - \mathbf{F}\|_F = \|\mathbf{C}[\mathbf{C}^\mathsf{T}\mathbf{FR}^\mathsf{T}]_k\mathbf{R} - \mathbf{F}\|_F.$$

*Proof.* For any matrix $\mathbf{Y}$ of appropriate dimension, we have $\langle \mathbf{F} - \mathbf{CC}^\mathsf{T}\mathbf{F}, \mathbf{CC}^\mathsf{T}\mathbf{F} - \mathbf{CYR} \rangle = 0$. This is because $\mathbf{F} - \mathbf{CC}^\mathsf{T}\mathbf{F} = (\mathbb{I} - \mathbf{CC}^\mathsf{T})\mathbf{F}$ lies in space orthogonal to $\mathbf{C}(\mathbf{C}^\mathsf{T}\mathbf{F} - \mathbf{YR})$. By Theorem 7,

$$\|\mathbf{F} - \mathbf{CYR}\|_F^2 = \|\mathbf{F} - \mathbf{CC}^\mathsf{T}\mathbf{F}\|_F^2 + \|\mathbf{CC}^\mathsf{T}\mathbf{F} - \mathbf{CYR}\|_F^2$$
$$= \|\mathbf{F} - \mathbf{CC}^\mathsf{T}\mathbf{F}\|_F^2 + \|\mathbf{C}^\mathsf{T}\mathbf{F} - \mathbf{YR}\|_F^2, \tag{7}$$

where the second equality follows from the properties of unitary matrices.

Again, for any matrix $\mathbf{Y}$ of appropriate dimensions, we have $\langle \mathbf{C}^\mathsf{T}\mathbf{FR}^\mathsf{T}\mathbf{R} - \mathbf{YR}, \mathbf{C}^\mathsf{T}\mathbf{F} - \mathbf{C}^\mathsf{T}\mathbf{FR}^\mathsf{T}\mathbf{R} \rangle = 0$. This is because $\mathbf{C}^\mathsf{T}\mathbf{FR}^\mathsf{T}\mathbf{R} - \mathbf{YR} = (\mathbf{C}^\mathsf{T}\mathbf{FR}^\mathsf{T} - \mathbf{Y})\mathbf{R}$ lies in the space spanned by $\mathbf{R}$, and $\mathbf{C}^\mathsf{T}\mathbf{F} - \mathbf{C}^\mathsf{T}\mathbf{FR}^\mathsf{T}\mathbf{R} = \mathbf{C}^\mathsf{T}\mathbf{F}(\mathbb{I} - \mathbf{R}^\mathsf{T}\mathbf{R})$ lies in the orthogonal space. By Theorem 7, we have

$$\|\mathbf{C}^\mathsf{T}\mathbf{F} - \mathbf{YR}\|_F^2 = \|\mathbf{C}^\mathsf{T}\mathbf{F} - \mathbf{C}^\mathsf{T}\mathbf{FR}^\mathsf{T}\mathbf{R}\|_F^2 + \|\mathbf{C}^\mathsf{T}\mathbf{FR}^\mathsf{T}\mathbf{R} - \mathbf{YR}\|_F^2 \tag{8}$$

Since $\|\mathbf{C}^\mathsf{T}\mathbf{F} - \mathbf{C}^\mathsf{T}\mathbf{FR}^\mathsf{T}\mathbf{R}\|_F^2$ is independent of $\mathbf{Y}$, we just bound the term $\|\mathbf{C}^\mathsf{T}\mathbf{FR}^\mathsf{T}\mathbf{R} - \mathbf{YR}\|_F^2$. Substituting $\mathbf{Y} = [\mathbf{CFR}]_k$ and using the fact that multiplying $\mathbf{R}$ from the right does not change the Frobenius norm and $[\mathbf{C}^\mathsf{T}\mathbf{FR}^\mathsf{T}]_k$ is the best $k$-rank approximation to the matrix $\mathbf{C}^\mathsf{T}\mathbf{FR}^\mathsf{T}$, for all rank-$k$ matrices $\mathbf{Z}$, we have

$$\|\mathbf{C}^\mathsf{T}\mathbf{FR}^\mathsf{T}\mathbf{R} - [\mathbf{C}^\mathsf{T}\mathbf{FR}^\mathsf{T}]_k\mathbf{R}\|_F^2 \leq \|\mathbf{C}^\mathsf{T}\mathbf{FR}^\mathsf{T}\mathbf{R} - \mathbf{ZR}\|_F^2. \tag{9}$$

Combining equation (9) with equation (8) and Theorem 7, we have

$$\|\mathbf{C}^\mathsf{T}\mathbf{F} - [\mathbf{CFR}]_k\mathbf{R}\|_F^2 \leq \|\mathbf{C}^\mathsf{T}\mathbf{F} - \mathbf{C}^\mathsf{T}\mathbf{FR}^\mathsf{T}\mathbf{R}\|_F^2 + \|\mathbf{C}^\mathsf{T}\mathbf{FR}^\mathsf{T}\mathbf{R} - \mathbf{ZR}\|_F^2 = \|\mathbf{C}^\mathsf{T}\mathbf{F} - \mathbf{ZR}\|_F^2. \tag{10}$$

Combining equation (10) with equation (7), the fact that $\mathbf{C}$ has orthonormal columns, and Theorem 7, we have

$$\|\mathbf{F} - \mathbf{C}[\mathbf{CFR}]_k\mathbf{R}\|_F^2 \leq \|\mathbf{F} - \mathbf{CC}^\mathsf{T}\mathbf{F}\|_F^2 + \|\mathbf{C}^\mathsf{T}\mathbf{F} - \mathbf{ZR}\|_F^2$$
$$= \|\mathbf{F} - \mathbf{CC}^\mathsf{T}\mathbf{F}\|_F^2 + \|\mathbf{CC}^\mathsf{T}\mathbf{F} - \mathbf{CZR}\|_F^2$$
$$= \|\mathbf{F} - \mathbf{CZR}\|_F^2.$$

This completes the proof of Lemma 6. $\qquad\qquad\square$

# E   Low Space Differentially private Low-rank Factorization

In this section, we give our basic low space algorithms for various granularity of privacy. These algorithms serve as a meta algorithm on which we built algorithms in various model of computations, like streaming model under turnstile update, continual release model, and local model. In Appendix E.1, we analyze the algorithm presented earlier. In Appendix E.2, we give our low space differentially private algorithm under $\mathsf{Priv}_2$, a stronger privacy guarantee.

## E.1   Low Space Differentially Private Low-rank Factorization Under $\mathsf{Priv}_1$

In this section, we analyze the algorithm presented earlier. For the ease of the readers, we first restate Theorem 3 here.

**Restatement of Theorem 3.** Let $m, n, k \in \mathbb{N}$ and $\alpha, \varepsilon, \delta$ be the input parameters. Let $s = \max\{m, n\}, u = \min\{m, n\}, \kappa = (1 + \alpha)/(1 - \alpha), \eta = \max\{k, \alpha^{-1}\}$, and $\sigma_{\min} = 16\log(1/\delta)\sqrt{t\kappa\ln(1/\delta)}/\varepsilon$. Given an $m \times n$ matrix $\mathbf{A}$, PRIVATE-SPACE-OPTIMAL-LRF, described in Figure 1, outputs a $k$-rank factorization $\widetilde{\mathbf{U}}, \widetilde{\mathbf{\Sigma}}$, and $\widetilde{\mathbf{V}}$, such that

1. PRIVATE-SPACE-OPTIMAL-LRF is $(3\varepsilon, 3\delta)$ differentially private under $\mathsf{Priv}_1$.

2. Let $\mathbf{M}_k = \widetilde{\mathbf{U}}\widetilde{\mathbf{\Sigma}}\widetilde{\mathbf{V}}^\mathsf{T}$. Then with probability $9/10$ over the random coins of PRIVATE-SPACE-OPTIMAL-LRF,

$$\|(\mathbf{A} \quad \mathbf{0}) - \mathbf{M}_k\|_F \le (1 + \alpha)\|\mathbf{A} - [\mathbf{A}]_k\|_F + O(\sigma_{\min}\sqrt{u} + \varepsilon^{-1}\sqrt{ks\ln(1/\delta)}),$$

3. The space used by PRIVATE-SPACE-OPTIMAL-LRF is $O((m + n)\eta\alpha^{-1}\log(k/\delta))$.

4. The total computational time is $O\left(\mathsf{nn}(\mathbf{A})\log(1/\delta) + \frac{(u^2 + s\eta)\eta\log^2(k/\delta)}{\alpha^2} + \frac{\eta^3\log^3(k/\delta)}{\alpha^5}\right)$.

First note that if we do not aim for run time efficiency, then we can simply use Gaussian random matrices instead of sampling $\mathbf{\Phi}, \Psi \sim \mathcal{D}_R$ and $\mathbf{S}, \mathbf{T} \sim \mathcal{D}_A$ as per Lemma 1 and Lemma 2. This would simplify the privacy proof as we will see later. Secondly, the probability of success can be amplified to get a high probability bound by standard techniques. We leave these details as they are standard arguments.

*Proof of Theorem 3.* Part 3 follows immediately by setting the values of $t$ and $v$. Part 4 of Theorem 3 requires some computation. More precisely, we have the following.

1. Computing $\mathbf{Y}_c$ requires $O(\mathsf{nn}(\mathbf{A})\log(1/\delta)) + m(m + n)t$ time and computing $\mathbf{Y}_r$ requires $O(\mathsf{nn}(\mathbf{A})\log(1/\delta)) + (m + n)t^2$ time.

2. Computing $\mathbf{U}$ and $\mathbf{V}$ requires $O(nt^2 + mt^2) = O((m + n)\eta^2\alpha^{-2}\log^2(k/\delta))$ time.

3. Computing a SVD of matrices $\mathbf{SU}$ and $\mathbf{TV}^\mathsf{T}$ requires $vt^2 + tv^2 = O(k^3\alpha^{-5}\log^2(k/\delta))$.

4. Computing $\mathbf{Z}$ requires $O(\mathsf{nn}(\mathbf{A})\log(1/\delta) + \eta^2/\alpha^2\log^2(k/\delta) + n\eta^2\alpha^{-2}\log^2(k/\delta))$ Computation of $[\widetilde{\mathbf{U}}_s^\mathsf{T}\mathbf{Z}\widetilde{\mathbf{V}}]_k$ requires $O(\mathsf{nn}(\mathbf{A})\log(1/\delta)) + tv^2 = O(\mathsf{nn}(\mathbf{A})\log(1/\delta) + k^3\alpha^{-5}\log^3(k/\delta))$ time.

5. Computation of the last SVD requires $O((m + n)\eta^2\alpha^{-2}\log^2(k/\delta))$ time.

Combining all these terms, we have our claim on the running time.

### E.1.1   Correctness Proof of PRIVATE-SPACE-OPTIMAL-LRF

We now prove the correctness guarantee of PRIVATE-SPACE-OPTIMAL-LRF. In what follows, we analyze the case when $m \le n$. The case when $n \le m$ follows analogously due to the symmetry of PRIVATE-SPACE-OPTIMAL-LRF. First note that appending $\mathbf{A}$ with an all zero matrix $\mathbf{0}^{m \times m}$ has no effect on its $k$-rank approximation, i.e., we can analyze the approximation guarantee for $\widehat{\mathbf{A}} := (\mathbf{A} \quad \mathbf{0})$ instead of $\mathbf{A}$. Let $\mathbf{M}_k$ be as defined in Theorem 3. We break our proof in three main steps.

**(i)** Lower bound $\|\mathbf{M}_k - \widehat{\mathbf{A}}\|_F$ by $\|\mathbf{M}_k - (\mathbf{A} \quad \mathbf{0})\|_F$ up to an additive term (Lemma 7).

**(ii)** Relate $\|\mathbf{M}_k - \widehat{\mathbf{A}}\|_F$ and $\|\widehat{\mathbf{A}} - [\widehat{\mathbf{A}}]_k\|_F$ (Lemma 8).

**(iii)** Upper bound $\|\widehat{\mathbf{A}} - [\widehat{\mathbf{A}}]_k\|_F$ by a term linear in $\|\mathbf{A} - [\mathbf{A}]_k\|_F$ up to an additive term (Lemma 9).

Part 2 of Theorem 3 follows by combining these three items together.

**Performing step (i).** We start by proving a bound on $\|\mathbf{M}_k - \widehat{\mathbf{A}}\|_F$ by $\|\mathbf{M}_k - \mathbf{A}\|_F$ and a small additive term. The following lemma provides such a bound.

**Lemma 7.** *Let* $\mathbf{A}$ *be an* $m \times n$ *input matrix, and let* $\widehat{\mathbf{A}} = (\mathbf{A} \quad \sigma_{\min}\mathbb{I}_m)$ *for* $\sigma_{\min}$ *defined in Theorem 3. Denote by* $\mathbf{M}_k := \widetilde{\mathbf{U}}\widetilde{\mathbf{\Sigma}}\widetilde{\mathbf{V}}^\mathsf{T}$ *the output of* PRIVATE-OPTIMAL-SPACE-LRF. *Then*

$$\|\mathbf{M}_k - (\mathbf{A} \quad \mathbf{0})\|_F \le \|\mathbf{M}_k - \widehat{\mathbf{A}}\|_F + \sigma_{\min}\sqrt{m}.$$

*Proof.* The lemma is immediate from the following observation: $\|\mathbf{M}_k - (\mathbf{A} \quad \mathbf{0})\|_F - \sigma_{\min}\|\mathbb{I}_m\|_F \leq \|\mathbf{M}_k - (\mathbf{A} \quad \mathbf{0}) - (\mathbf{0} \quad \sigma_{\min}\mathbb{I})\|_F = \|\mathbf{M}_k - \widehat{\mathbf{A}}\|_F$, where the first inequality follows from the sub-additivity of the Frobenius norm. $\qquad\square$

**Performing step (ii).** This is the most involved part of the proof and uses multiple lemmas as follows.

**Lemma 8.** *Let* $\widehat{\mathbf{A}} = (\mathbf{A} \quad \sigma_{\min}\mathbb{I})$ *and denote by* $\mathbf{M}_k := \widetilde{\mathbf{U}}\widetilde{\mathbf{\Sigma}}\widetilde{\mathbf{V}}^\mathsf{T}$. *Let* $\widehat{\mathbf{\Phi}} = t^{-1}\mathbf{\Omega}\mathbf{\Phi}$. *Then with probability* $1 - O(\delta)$ *over the random coins of the algorithm* PRIVATE-SPACE-OPTIMAL-LRF,

$$\|\mathbf{M}_k - \widehat{\mathbf{A}}\|_F \leq (1+\alpha)\|\widehat{\mathbf{A}} - [\widehat{\mathbf{A}}]_k\|_F + 2\|\mathbf{S}^\dagger\mathbf{N}(\mathbf{T}^\mathsf{T})^\dagger\|_F + \|\widehat{\mathbf{A}}\widehat{\mathbf{\Phi}}([\widehat{\mathbf{A}}]_k\widehat{\mathbf{\Phi}})^\dagger(\mathbf{\Psi}\widehat{\mathbf{A}}\widehat{\mathbf{\Phi}}([\widehat{\mathbf{A}}]_k\widehat{\mathbf{\Phi}})^\dagger)^\dagger\mathbf{N}_1\|_F.$$

*Proof.* Let $\mathbf{B} = \widehat{\mathbf{A}} + \mathbf{S}^\dagger\mathbf{N}(\mathbf{T}^\dagger)^\mathsf{T}$ and $\widehat{\mathbf{\Phi}} = \mathbf{\Omega}\mathbf{\Phi}$. We first use the relation between $\min_{\mathbf{X},r(\mathbf{X})\leq k}\|\widehat{\mathbf{A}}\widehat{\mathbf{\Phi}}\mathbf{X}\mathbf{\Psi}\widehat{\mathbf{A}} - \widehat{\mathbf{A}}\|$ and $(1+\alpha)\|\widehat{\mathbf{A}} - [\widehat{\mathbf{A}}]_k\|_F$ from the proof of Theorem 10. Using Lemma 1 and Fact 1, if we set $\mathbf{A} = \widehat{\mathbf{A}}$ in equation (6), then with probability $1 - 3\delta$ over $\widehat{\mathbf{\Phi}}^\mathsf{T}, \mathbf{\Psi} \sim \mathcal{D}_R$, $\|\widehat{\mathbf{A}}\widehat{\mathbf{\Phi}}([\widehat{\mathbf{A}}]_k\widehat{\mathbf{\Phi}})^\dagger(\mathbf{\Psi}\widehat{\mathbf{A}}\widehat{\mathbf{\Phi}}([\widehat{\mathbf{A}}]_k\widehat{\mathbf{\Phi}})^\dagger)^\dagger\mathbf{\Psi}\widehat{\mathbf{A}} - \widehat{\mathbf{A}}\|_F \leq (1+\alpha)^2\|\widehat{\mathbf{A}} - [\widehat{\mathbf{A}}]_k\|_F$. Now define a rank-$k$ matrix $\mathbf{P}_k := ([\widehat{\mathbf{A}}]_k\widehat{\mathbf{\Phi}})^\dagger(\mathbf{\Psi}\widehat{\mathbf{A}}\widehat{\mathbf{\Phi}}([\widehat{\mathbf{A}}]_k\widehat{\mathbf{\Phi}})^\dagger)^\dagger$. Let us consider the following optimization problem:

$$\min_{\substack{\mathbf{X}\\r(\mathbf{X})\leq k}}\|\mathbf{Y}_c\mathbf{X}\mathbf{Y}_r - \mathbf{B}\|_F.$$

Since $\mathbf{P}_k$ is a rank-$k$ matrix, using the subadditivity of the Frobenius norm, we have

$$\min_{\substack{\mathbf{X},\\r(\mathbf{X})\leq k}}\|\mathbf{Y}_c\mathbf{X}\mathbf{Y}_r - \mathbf{B}\|_F \leq \|\mathbf{Y}_c\mathbf{P}_k\mathbf{Y}_r - \mathbf{B}\|_F$$

$$= \|\mathbf{Y}_c\mathbf{P}_k\mathbf{Y}_r - (\widehat{\mathbf{A}} + \mathbf{S}^\dagger\mathbf{N}(\mathbf{T}^\dagger)^\mathsf{T})\|_F$$
$$\leq \|\mathbf{Y}_c\mathbf{P}_k\mathbf{Y}_r - \widehat{\mathbf{A}}\|_F + \|\mathbf{S}^\dagger\mathbf{N}(\mathbf{T}^\dagger)^\mathsf{T}\|_F$$
$$= \|\widehat{\mathbf{A}}\widehat{\mathbf{\Phi}}\mathbf{P}_k(\mathbf{\Psi}\widehat{\mathbf{A}} + \mathbf{N}_1) - \widehat{\mathbf{A}}\|_F + \|\mathbf{S}^\dagger\mathbf{N}(\mathbf{T}^\dagger)^\mathsf{T}\|_F$$
$$\leq \|\widehat{\mathbf{A}}\widehat{\mathbf{\Phi}}\mathbf{P}_k\mathbf{\Psi}\widehat{\mathbf{A}} - \widehat{\mathbf{A}}\|_F + \|\mathbf{S}^\dagger\mathbf{N}_2(\mathbf{T}^\dagger)^\mathsf{T}\|_F + \|\widehat{\mathbf{A}}\widehat{\mathbf{\Phi}}\mathbf{P}_k\mathbf{N}_1\|_F$$
$$\leq (1+\alpha)^2\|\widehat{\mathbf{A}} - [\widehat{\mathbf{A}}]_k\|_F + \|\mathbf{S}^\dagger\mathbf{N}_2(\mathbf{T}^\dagger)^\mathsf{T}\|_F + \|\widehat{\mathbf{A}}\widehat{\mathbf{\Phi}}\mathbf{P}_k\mathbf{N}_1\|_F \quad (11)$$

Let $\mathbf{S}_2 = \widehat{\mathbf{A}}\widehat{\mathbf{\Phi}}\mathbf{P}_k\mathbf{N}_1$. By definition, $\mathbf{V}$ is a matrix whose rows are an orthonormal basis for the row space of $\mathbf{Y}_r$ and $\mathbf{U}$ is a matrix whose columns are an orthonormal basis for the column space of $\mathbf{Y}_c$. Therefore,

$$\min_{\substack{\mathbf{Y}\\r(\mathbf{Y})\leq k}}\|\mathbf{U}\mathbf{Y}\mathbf{V} - \mathbf{B}\|_F \leq \min_{\substack{\mathbf{X}\\r(\mathbf{X})\leq k}}\|\mathbf{Y}_c\mathbf{X}\mathbf{V} - \mathbf{B}\|_F \leq \min_{\substack{\mathbf{X}\\r(\mathbf{X})\leq k}}\|\mathbf{Y}_c\mathbf{X}\mathbf{Y}_r - \mathbf{B}\|_F. \quad (12)$$

Combining equation (11) and equation (12), we have

$$\min_{\substack{\mathbf{Y}\\r(\mathbf{Y})\leq k}}\|\mathbf{U}\mathbf{Y}\mathbf{V} - \mathbf{B}\|_F \leq (1+\alpha)^2\|\widehat{\mathbf{A}} - [\widehat{\mathbf{A}}]_k\|_F + \|\mathbf{S}^\dagger\mathbf{N}_2(\mathbf{T}^\dagger)^\mathsf{T}\|_F + \|\mathbf{S}_2\|_F. \quad (13)$$

**Claim 1.** *Let* $\mathbf{U}, \mathbf{V}, \mathbf{B}, \mathbf{A}, \mathbf{S}, \mathbf{T}$ *and* $\mathbf{N}_2$ *be as above. Let* $\mathcal{D}_A$ *be a distribution that satisfies* $(\alpha, \delta)$-*affine embedding. Let* $\widetilde{\mathbf{X}} = \text{argmin}_{\mathbf{X},r(\mathbf{X})=k}\|\mathbf{S}(\mathbf{U}\mathbf{X} - \mathbf{B})\|_F$. *Then with probability* $1 - O(\delta)$ *over* $\mathbf{S}, \mathbf{T}^\mathsf{T} \sim \mathcal{D}_A$,

$$\|(\mathbf{U}\widetilde{\mathbf{X}}\mathbf{V} - \mathbf{B})\|_F \leq (1+\alpha)^4\|\widehat{\mathbf{A}} - [\widehat{\mathbf{A}}]_k\|_F + 4\|\mathbf{S}^\dagger\mathbf{N}_2(\mathbf{T}^\dagger)^\mathsf{T}\|_F + 4\|\mathbf{S}_2\|_F.$$

*Proof.* Set $p = t$, $\mathbf{D} = \mathbf{U}$ and $\mathbf{E} = \mathbf{B}$ in the statement of Lemma 2. Let us restrict our attention to matrices $\mathbf{X}$ with rank at most $k$ and denote by

$$\widehat{\mathbf{X}} = \text{argmin}_{\mathbf{X},r(\mathbf{X})\leq k}\|\mathbf{U}\mathbf{X}\mathbf{V} - \mathbf{B}\|_F \quad \text{and} \quad \widetilde{\mathbf{X}} = \text{argmin}_{\mathbf{X},r(\mathbf{X})\leq k}\|\mathbf{S}(\mathbf{U}\mathbf{X}\mathbf{V} - \mathbf{B})\mathbf{T}^\mathsf{T}\|_F.$$

Then we have with probability $1 - 3\delta$ over $\mathbf{S} \sim \mathcal{D}_A$,

$$\min_{\substack{\mathbf{X}\\r(\mathbf{X})=k}}\|\mathbf{U}\mathbf{X}\mathbf{V} - \mathbf{B}\|_F = \|\mathbf{U}\widehat{\mathbf{X}}\mathbf{V} - \mathbf{B}\|_F \geq (1+\alpha)^{-1/2}\|\mathbf{S}(\mathbf{U}\widehat{\mathbf{X}}\mathbf{V} - \mathbf{B})\|_F. \quad (14)$$

Substituting $\mathbf{D} = \mathbf{V}^{\mathsf{T}}$, $\mathbf{X} = (\mathbf{SU}\widehat{\mathbf{X}})^{\mathsf{T}}$ and $\mathbf{E} = (\mathbf{SB})^{\mathsf{T}}$ in the statement of Lemma 2, with probability $1 - 4\delta$,

$$
\begin{aligned}
(1+\alpha)^{-1/2}\|\mathbf{S}(\mathbf{U}\widehat{\mathbf{X}}\mathbf{V} - \mathbf{B})\|_F &= (1+\alpha)^{-1/2}\|\mathbf{V}^{\mathsf{T}}(\mathbf{SU}\widehat{\mathbf{X}})^{\mathsf{T}} - (\mathbf{SB})^{\mathsf{T}}\|_F \\
&\geq (1+\alpha)^{-1}\|\mathbf{T}(\mathbf{V}^{\mathsf{T}}(\mathbf{SU}\widehat{\mathbf{X}})^{\mathsf{T}} - (\mathbf{SB})^{\mathsf{T}})\|_F \\
&= (1+\alpha)^{-1}\|\mathbf{S}(\mathbf{U}\widehat{\mathbf{X}}\mathbf{V} - \mathbf{B})\mathbf{T}^{\mathsf{T}}\|_F \\
&\geq (1+\alpha)^{-1}\min_{\substack{\mathbf{X} \\ r(\mathbf{X}) \leq k}}\|\mathbf{S}(\mathbf{U}\mathbf{X}\mathbf{V} - \mathbf{B})\mathbf{T}^{\mathsf{T}}\|_F \\
&= (1+\alpha)^{-1}\|\mathbf{S}(\mathbf{U}\widetilde{\mathbf{X}}\mathbf{V} - \mathbf{B})\mathbf{T}^{\mathsf{T}}\|_F \\
&\geq (1+\alpha)^{-2}\|(\mathbf{U}\widetilde{\mathbf{X}}\mathbf{V} - \mathbf{B})\|_F. \qquad (15)
\end{aligned}
$$

Combining equation (15) with equation (13), with probability $1 - O(\delta)$ over the random coins of PRIVATE-OPTIMAL-SPACE-LRF,

$$
\|(\mathbf{U}\widetilde{\mathbf{X}}\mathbf{V} - \mathbf{B})\|_F \leq (1+\alpha)^4\|\widehat{\mathbf{A}} - [\widehat{\mathbf{A}}]_k\|_F + 4(\|\mathbf{S}^{\dagger}\mathbf{N}_2(\mathbf{T}^{\dagger})^{\mathsf{T}}\|_F + \|\mathbf{S}_2\|_F) \qquad (16)
$$

as $\alpha \in (0,1)$. This completes the proof of Claim 1. $\qquad\square$

To finalize the proof, we need to compute

$$
\widetilde{\mathbf{X}} = \operatorname*{argmin}_{\substack{\mathbf{X} \\ r(\mathbf{X}) \leq k}}\|\mathbf{S}(\mathbf{U}\mathbf{X}\mathbf{V} - \mathbf{B})\mathbf{T}^{\mathsf{T}}\|_F.
$$

We use Lemma 6 to compute $\widetilde{\mathbf{X}}$. Recall $\mathbf{SU} = \mathbf{U}_s\boldsymbol{\Sigma}_s\mathbf{V}_s^{\mathsf{T}}$ and $\mathbf{TV}^{\mathsf{T}} = \mathbf{U}_t\boldsymbol{\Sigma}_t\mathbf{V}_t^{\mathsf{T}}$. Using Lemma 6 with $\mathbf{C} = \widetilde{\mathbf{U}}_s$, $\mathbf{R} = \widetilde{\mathbf{V}}_t^{\mathsf{T}}$ and $\mathbf{F} = \mathbf{Z} = \mathbf{S}\widehat{\mathbf{A}}\mathbf{T}^{\mathsf{T}} + \mathbf{N}_2$, we get

$$
[\widetilde{\mathbf{U}}_s^{\mathsf{T}}\mathbf{SBT}^{\mathsf{T}}\widetilde{\mathbf{V}}_t]_k = \operatorname*{argmin}_{\substack{\mathbf{X} \\ r(\mathbf{X}) \leq k}}\|\widetilde{\mathbf{U}}_s\mathbf{X}\widetilde{\mathbf{V}}_t^{\mathsf{T}} - \mathbf{SBT}^{\mathsf{T}}\|_F
$$

This implies that $\operatorname{argmin}_{\mathbf{X}, r(\mathbf{X}) \leq k}\|\mathbf{S}(\mathbf{U}\mathbf{X}\mathbf{V} - \mathbf{B})\mathbf{T}^{\mathsf{T}}\|_F$ has closed form

$$
\widetilde{\mathbf{X}} = \widetilde{\mathbf{V}}_s\widetilde{\boldsymbol{\Sigma}}_s^{\dagger}[\widetilde{\mathbf{U}}_s^{\mathsf{T}}\mathbf{Z}\widetilde{\mathbf{V}}_t]_k\widetilde{\boldsymbol{\Sigma}}_t^{\dagger}\widetilde{\mathbf{U}}_t^{\mathsf{T}} \qquad (17)
$$

Recall, $\widetilde{\mathbf{X}} = \widetilde{\mathbf{V}}_s\widetilde{\boldsymbol{\Sigma}}_s^{\dagger}[\widetilde{\mathbf{U}}_s^{\mathsf{T}}\mathbf{Z}\widetilde{\mathbf{V}}_t]_k\widetilde{\boldsymbol{\Sigma}}_t^{\dagger}\widetilde{\mathbf{U}}_t^{\mathsf{T}} = \mathbf{U}'\boldsymbol{\Sigma}'\mathbf{V}'^{\mathsf{T}}$. Substituting equation (17) in equation (16) and the fact that $\mathbf{B} = \widehat{\mathbf{A}} + \mathbf{S}^{\dagger}\mathbf{N}_2(\mathbf{T}^{\dagger})^{\mathsf{T}}$, we have

$$
\begin{aligned}
\|\mathbf{U}\mathbf{U}'\boldsymbol{\Sigma}'(\mathbf{V}^{\mathsf{T}}\mathbf{V}')^{\mathsf{T}} - \widehat{\mathbf{A}}\|_F - \|\mathbf{S}^{\dagger}\mathbf{N}_2(\mathbf{T}^{\mathsf{T}})^{\dagger}\|_F &\leq \|\mathbf{U}\mathbf{U}'\boldsymbol{\Sigma}'(\mathbf{V}^{\mathsf{T}}\mathbf{V}')^{\mathsf{T}} - \mathbf{B}\|_F \\
&\leq (1+\alpha)^6\|\widehat{\mathbf{A}} - [\widehat{\mathbf{A}}]_k\|_F + O(\|\mathbf{S}^{\dagger}\mathbf{N}_2(\mathbf{T}^{\mathsf{T}})^{\dagger}\|_F + \|\mathbf{S}_2\|_F).
\end{aligned}
$$

This in particular implies that

$$
\|\mathbf{U}\mathbf{U}'\boldsymbol{\Sigma}'(\mathbf{V}^{\mathsf{T}}\mathbf{V}')^{\mathsf{T}} - \widehat{\mathbf{A}}\|_F \leq (1+\alpha)^6\|\widehat{\mathbf{A}} - [\widehat{\mathbf{A}}]_k\|_F + O(\|\mathbf{S}^{\dagger}\mathbf{N}_2(\mathbf{T}^{\mathsf{T}})^{\dagger}\|_F + \|\mathbf{S}_2\|_F).
$$

Scaling the value of $\alpha$ by a constant completes the proof of Lemma 8. $\qquad\square$

**Performing step (iii).** In order to complete the proof, we compute an upper bound on $\|\widehat{\mathbf{A}} - [\widehat{\mathbf{A}}]_k\|_F$. For this, we need the Weyl's perturbation theorem (Theorem 9).

**Lemma 9.** *Let $d$ be the maximum of the rank of $\mathbf{A}$ and $\widehat{\mathbf{A}}$. Let $\sigma_1, \cdots, \sigma_d$ be the singular values of $\mathbf{A}$ and $\sigma_1', \cdots, \sigma_d'$ be the singular values of $\widehat{\mathbf{A}}$. Then $|\sigma_i - \sigma_i'| \leq \sigma$ for all $1 \leq i \leq d$.*

*Proof.* The lemma follows from the basic application of Theorem 9. We can write $\widehat{\mathbf{A}} = (\mathbf{A} \quad \mathbf{0}) + (\mathbf{0} \quad \sigma_{\min}\mathbb{I}_m)$. The lemma follows since, by construction, all the singular values of $(\mathbf{0} \quad \sigma_{\min}\mathbb{I}_m)$ are $\sigma_{\min}$. $\qquad\square$

To compute the additive error, we need to bound $\|\mathbf{S}^\dagger \mathbf{N}_2 (\mathbf{T}^\mathsf{T})^\dagger\|_F$ and $\|\mathbf{S}_2\|_F$. This is done by the following two lemmas.

**Claim 2.** *Let $\mathcal{D}_R$ be a distribution that satisfies $(\alpha, \delta)$-subspace embedding for generalized regression. Let $\mathbf{S}_2$ be as defined above. Then with probability $99/100$ over $\widehat{\mathbf{\Phi}} \sim \mathcal{D}_R$, $\|\mathbf{S}_2\|_F = \rho_1 \sqrt{kn(1+\alpha)}$.*

*Proof.* Let $\mathbf{G} = \mathbf{\Psi} \mathbf{A} \mathbf{\Phi}([\widehat{\mathbf{A}}]_k \mathbf{\Phi})^\dagger$. $\mathbf{G}$ is an $m \times k$ matrix. When $\alpha \leq 1$, $\mathbf{G}$ has rank $k$. This implies that there exist a $t \times k$ matrix $\widehat{\mathbf{U}}$ with orthonormal columns such that $\mathbf{G}\mathbf{G}^\dagger = \widehat{\mathbf{U}}\widehat{\mathbf{U}}^\mathsf{T}$. Therefore, $\mathbf{\Psi}\mathbf{S}_2 = \mathbf{G}\mathbf{G}^\dagger \mathbf{N}_2 = \widehat{\mathbf{U}}\widehat{\mathbf{U}}^\mathsf{T} \mathbf{N}_1$. From the second claim of Lemma 1 and the choice of the parameter $t$, $\|\mathbf{S}_2\|_F^2 \leq (1+\alpha)\|\widehat{\mathbf{U}}\widehat{\mathbf{U}}^\mathsf{T} \mathbf{N}_1\|_F^2$. Since every entries of $\mathbf{N}_1$ are picked i.i.d. and $\widehat{\mathbf{U}}\widehat{\mathbf{U}}^\mathsf{T}$ is an orthonormal projection onto a $k$-dimensional subspace, we have $\|\mathbf{S}_2\|_F = O(\rho_1 \sqrt{kn(1+\alpha)})$. $\square$

The following claim follows from Lemma 5.

**Claim 3.** *Let $\mathbf{S}, \mathbf{T} \sim \mathcal{D}_A$. Then for any matrix $\mathbf{N}_2$ of appropriate dimension, $\|\mathbf{S}^\dagger \mathbf{N}_2 (\mathbf{T}^\mathsf{T})^\dagger\|_F = \|\mathbf{N}_2\|_F$.*

*Proof.* Let $\mathbf{C} = \mathbf{S}^\dagger \mathbf{N}_2 (\mathbf{T}^\mathsf{T})^\dagger$. Then $\mathbf{S}\mathbf{C}\mathbf{T}^\mathsf{T} = \mathbf{S}\mathbf{S}^\dagger \mathbf{N}_2 (\mathbf{T}\mathbf{T}^\dagger)^\mathsf{T}$. Now $\mathbf{S}\mathbf{S}^\dagger$ (similarly, $\mathbf{T}\mathbf{T}^\dagger$) is a projection unto a random subspace of dimension $k$. Since every entries of $\mathbf{N}_2$ is picked i.i.d. from $\mathcal{N}(0, \rho^2)$, $\mathbf{S}\mathbf{C}\mathbf{T}^\mathsf{T} = \widetilde{\mathbf{N}}_2$, where $\widetilde{\mathbf{N}}_1$ is an $v \times v$ matrix with every entries picked i.i.d. from $\mathcal{N}(0, \rho_2^2)$. Using Lemma 4, this implies that

$$\mathbb{E}[\|\mathbf{S}\mathbf{C}\mathbf{T}^\mathsf{T}\|_F^2] = \mathbb{E}\left[\|\widetilde{\mathbf{N}}_2\|_F^2\right] = \sum_{i,j} \mathbb{E}[(\widetilde{\mathbf{N}}_2)_{ij}^2] = v^2 \rho_2^2.$$

The result follows using Markov's inequality and the fact that $\|\mathbf{S}\mathbf{C}\mathbf{T}^\mathsf{T}\|_F^2 = (1+\alpha)^2 \|\mathbf{C}\|_F^2$ and $\alpha \leq 1$. $\square$

The above claim implies that $\|\mathbf{N}_2\|_F = O(\rho_2 v)$ with probability $99/100$.

Since $\|\widehat{\mathbf{A}} - [\widehat{\mathbf{A}}]_k\|_F^2 \leq \sum_{i > k} \sigma_i'^2$ and $\|\mathbf{A} - \mathbf{A}_k\|_F^2 \leq \sum_{i > k} \sigma_i^2$, combining Lemma 7, Lemma 8, Claim 2, Claim 3, Lemma 9, and Lemma 4, we have the final utility bound.

**Lemma 10.** *Let $\rho_1, \rho_2$, and $\sigma_{\min}$ be as defined in Theorem 3. With probability $99/100$ over the coins of the algorithm* PRIVATE-OPTIMAL-SPACE-LRF, *the output of* PRIVATE-OPTIMAL-SPACE-LRF *satisfies*

$$\| (\mathbf{A} \quad \mathbf{0}) - \mathbf{M}_k \|_F \leq (1+\alpha)\|\mathbf{A} - [\mathbf{A}]_k\|_F + O(\sigma_{\min}\sqrt{m} + \rho_1 \sqrt{kn(1+\alpha)} + \rho_2 v).$$

Now observe that $v = O((\eta/\alpha^2) \log(k/\delta)) \ll \min\{m, n\}$ and $1 + \alpha \leq 2$. If former is not the case, then there is no reason to do a random projection. Therefore, the term $\rho_2 v$ is subsumed by the rest of the term. The result follows by setting the values of $\rho_1$ and $\sigma_{\min}$. $\square$

### E.1.2 Privacy Proof of PRIVATE-SPACE-OPTIMAL-LRF

Our privacy result can be restated as the following lemma.

**Lemma 11.** *If $\sigma_{\min}, \rho_1$ and $\rho_2$ be as in Theorem 3, then the algorithm presented in Figure 1,* ALGORITHM 2, *is $(3\varepsilon, 3\delta)$-differentially private.*

We prove the lemma when $m \leq n$. The case for $m \geq n$ is analogous after inverting the roles of $\widehat{\mathbf{\Phi}}$ and $\mathbf{\Psi}$. Let $\mathbf{A}$ and $\mathbf{A}'$ be two neighboring matrices, i.e., $\mathbf{E} = \mathbf{A} - \mathbf{A}' = \mathbf{u}\mathbf{v}^\mathsf{T}$. Then $\widehat{\mathbf{A}}$ and $\widehat{\mathbf{A}}'$, constructed by OPTIMAL-SPACE-PRIVATE-LRF, has the following property: $\widehat{\mathbf{A}}' = \widehat{\mathbf{A}} + (\mathbf{E} \quad \mathbf{0})$.

**Claim 4.** *If $\rho_1 = \frac{\sqrt{(1+\alpha)\ln(1/\delta)}}{\varepsilon}$ and $\rho_2 = \frac{(1+\alpha)\sqrt{\ln(1/\delta)}}{\varepsilon}$, then publishing $\mathbf{Y}_r$ and $\mathbf{Z}$ preserves $(2\varepsilon, 2\delta)$-differential privacy.*

*Proof.* We use the second claims of Lemma 1 and Lemma 2, i.e., $\|\mathbf{SD}\|_F^2 = (1 \pm \alpha)\|\mathbf{D}\|_F^2$ and $\|\mathbf{\Psi D}\|_F^2 = (1 \pm \alpha)\|\mathbf{D}\|_F^2$ for all $\mathbf{D}$, where $\mathbf{S} \sim \mathcal{D}_A$ and $\mathbf{\Psi} \sim \mathcal{D}_R$. Let $\mathbf{A}$ and $\mathbf{A}'$ be two neighboring matrices such that $\mathbf{E} = \mathbf{A} - \mathbf{A}' = \mathbf{uv}^\mathsf{T}$. Then $\|\mathbf{S}(\mathbf{E} \quad \mathbf{0})\mathbf{T}^\mathsf{T}\|_F^2 \leq (1+\alpha)\|(\mathbf{E} \quad \mathbf{0})\mathbf{T}^\mathsf{T}\|_F^2 \leq (1+\alpha)^2$. Publishing $\mathbf{Z}$ preserves $(\varepsilon, \delta)$-differential privacy follows from considering the vector form of the matrix $\mathbf{S}\widehat{\mathbf{A}}\mathbf{T}^\mathsf{T}$ and $\mathbf{N}_2$ and applying Theorem 5. Similarly, we use Theorem 5 and the fact that, for any matrix $\mathbf{C}$ of appropriate dimension, $\|\mathbf{\Psi C}\|^2 \leq (1+\alpha)\|\mathbf{C}\|_F^2$, to prove that publishing $\mathbf{\Psi}\widehat{\mathbf{A}} + \mathbf{N}_1$ preserves differential privacy. $\qquad \square$

We next prove that $\mathbf{Y}_c$ is $(\varepsilon, \delta)$-differentially private. This would complete the proof of Lemma 11 by combining Lemma 3 and Theorem 4 with the above claim. Let $\mathbf{A} - \mathbf{A}' = \mathbf{E} = \mathbf{uv}^\mathsf{T}$ and let $\widehat{\mathbf{v}} = (\mathbf{v} \quad \mathbf{0}^m)$. Then $\widehat{\mathbf{A}} - \widehat{\mathbf{A}}' = \mathbf{u}\widehat{\mathbf{v}}^\mathsf{T}$. Since $\mathbf{\Phi}^\mathsf{T}$ is sampled from $\mathcal{D}_R$, we have $\|\mathbf{\Phi}^\mathsf{T}\mathbf{W}\|_F^2 = (1+\alpha)\|\mathbf{W}\|_F^2$ for any matrix $\mathbf{W}$ with probability $1 - \delta$ (second claim of Lemma 1). Therefore, $\mathbf{uv}^\mathsf{T}\mathbf{\Phi} = (1+\alpha)^{1/2}\mathbf{u}\widehat{\mathbf{v}}^\mathsf{T} = \widetilde{\mathbf{u}}\widetilde{\mathbf{v}}^\mathsf{T}$ for some unit vectors $\mathbf{u}, \widetilde{\mathbf{v}}$ and $\widetilde{\mathbf{u}} = (1+\alpha)^{1/2}\mathbf{u}$. We now show that $\widehat{\mathbf{A}}\mathbf{\Phi}\mathbf{\Omega}_1$ preserves privacy. We prove that each row of the published matrix preserves $(\varepsilon_0, \delta_0)$-differential privacy for some appropriate $\varepsilon_0, \delta_0$, and then invoke Theorem 4 to prove that the published matrix preserves $(\varepsilon, \delta)$-differential privacy.

It may seem that the privacy of $\mathbf{Y}_c$ follows from the result of Blocki *et al.* [13], but this is not the case because of the following reasons.

1. The definition of neighboring matrices considered in this paper is different from that of Blocki *et al.* [13]. To recall, Blocki *et al.* [13] considered two matrices neighboring if they differ in at most one row by a unit norm. In our case, we consider two matrices are neighboring if they have the form $\mathbf{uv}^\mathsf{T}$ for unit vectors $\mathbf{u}, \mathbf{v}$.

2. We multiply the Gaussian matrix to a random projection of $\widehat{\mathbf{A}}$ and not to $\mathbf{A}$ as in the case of Blocki *et al.* [13], i.e., to $\widehat{\mathbf{A}}\mathbf{\Phi}$ and not to $\widehat{\mathbf{A}}$.

If we do not care about the run time efficiency of the algorithm, then we can set $\widehat{\mathbf{\Phi}} := \mathbf{\Omega}$ instead of $\widehat{\mathbf{\Phi}} := \mathbf{\Omega}\mathbf{\Phi}$. In this case, we would not need to deal with the second issue mentioned above.

We first give a brief overview of how to deal with these issues here. The first issue is resolved by analyzing $(\widehat{\mathbf{A}} - \widehat{\mathbf{A}}')\mathbf{\Phi}$. We observe that this expression can be represented in the form of $\widetilde{\mathbf{u}}\widetilde{\mathbf{v}}^\mathsf{T}$, where $\widetilde{\mathbf{u}} = (1+\alpha)^{1/2}\mathbf{u}$ for some $\|\mathbf{u}\|_2 = 1, \|\widetilde{\mathbf{v}}\|_2 = 1$. The second issue can be resolved by observing that $\mathbf{\Phi}$ satisfies $(\alpha, \delta)$-JLP because of the choice of $t$. Since the rank of $\widehat{\mathbf{A}}$ and $\widehat{\mathbf{A}}\mathbf{\Phi}$ are the same, the singular values of $\widehat{\mathbf{A}}\mathbf{\Phi}$ are within a multiplicative factor of $(1 \pm \alpha)^{1/2}$ of the singular values of $\mathbf{\Phi}$ with probability $1 - \delta$ due to Sarlos [86]. Therefore, we scale the singular values of $\widehat{\mathbf{A}}$ appropriately.

We now return to the proof. Denote by $\widehat{\mathbf{A}} = (\mathbf{A} \quad \sigma_{\min}\mathbb{I}_m)$ and by $\widehat{\mathbf{A}}' = (\mathbf{A}' \quad \sigma_{\min}\mathbb{I}_m)$, where $\mathbf{A} - \mathbf{A}' = \mathbf{uv}^\mathsf{T}$. Then $\widehat{\mathbf{A}}' - \widehat{\mathbf{A}} = (\mathbf{uv}^\mathsf{T} \quad \mathbf{0})$. Let $\mathbf{U_C}\mathbf{\Sigma_C}\mathbf{V_C^\mathsf{T}}$ be the SVD of $\mathbf{C} = \widehat{\mathbf{A}}\mathbf{\Phi}$ and $\widetilde{\mathbf{U}}_\mathbf{C}\widetilde{\mathbf{\Sigma}}_\mathbf{C}\widetilde{\mathbf{V}}_\mathbf{C}^\mathsf{T}$ be the SVD of $\widetilde{\mathbf{C}} = \widehat{\mathbf{A}}'\mathbf{\Phi}$. From above discussion, we know that if $\mathbf{A} - \mathbf{A}' = \mathbf{uv}^\mathsf{T}$, then $\mathbf{C} - \widetilde{\mathbf{C}} = (1+\alpha)^{1/2}\widetilde{\mathbf{u}}\widetilde{\mathbf{v}}^\mathsf{T}$ for some unit vectors $\widetilde{\mathbf{u}}$ and $\widetilde{\mathbf{v}}$. For notational brevity, in what follows we write $\mathbf{u}$ for $\widetilde{\mathbf{u}}$ and $\mathbf{v}$ for $\widetilde{\mathbf{v}}$.

Note that both $\mathbf{C}$ and $\widetilde{\mathbf{C}}$ are full rank matrices because of the construction; therefore $\mathbf{CC^\mathsf{T}}$ (respectively, $\widetilde{\mathbf{C}}\widetilde{\mathbf{C}}^\mathsf{T}$) is a full dimensional $m \times m$ matrix. This implies that the affine transformation of the multi-variate Gaussian is well-defined (both the covariance $(\mathbf{CC^\mathsf{T}})^{-1}$ has full rank and $\det(\mathbf{CC^\mathsf{T}})$ is non-zero). That is, the PDF of the distributions of the rows, corresponding to $\mathbf{C}$ and $\widetilde{\mathbf{C}}$, is just a linear transformation of $\mathcal{N}(\mathbf{0}, \mathbb{I}_{m \times m})$. Let $\mathbf{y} \sim \mathcal{N}(0, 1)^t$.

$$\mathsf{PDF}_{\mathbf{C}Y}(\mathbf{x}) = \frac{1}{\sqrt{(2\pi)^t \det(\mathbf{CC^\mathsf{T}})}} e^{(-\frac{1}{2}\mathbf{x}(\mathbf{CC^\mathsf{T}})^{-1}\mathbf{x}^\mathsf{T})}$$

$$\mathsf{PDF}_{\widetilde{\mathbf{C}}Y}(\mathbf{x}) = \frac{1}{\sqrt{(2\pi)^t \det(\widetilde{\mathbf{C}}\widetilde{\mathbf{C}}^\mathsf{T})}} e^{(-\frac{1}{2}\mathbf{x}(\widetilde{\mathbf{C}}\widetilde{\mathbf{C}}^\mathsf{T})^{-1}\mathbf{x}^\mathsf{T})}$$

Let $\varepsilon_0 = \frac{\varepsilon}{\sqrt{4t\ln(1/\delta)\log(1/\delta)}}$ and $\delta_0 = \delta/2t$, We prove that every row of the published matrix is $(\varepsilon_0, \delta_0)$ differentially private. Let $\mathbf{x}$ be sampled either from $\mathcal{N}(\mathbf{0}, \mathbf{CC^\mathsf{T}})$ or $\mathcal{N}(\mathbf{0}, \widetilde{\mathbf{C}}\widetilde{\mathbf{C}}^\mathsf{T})$. It is

straightforward to see that the combination of Claim 5 and Claim 6 below proves differential privacy for a row of published matrix. The lemma then follows by an application of Theorem 4 and our choice of $\varepsilon_0$ and $\delta_0$.

**Claim 5.** *Let* $\mathbf{C}$ *and* $\varepsilon_0$ *be as defined above. Then*

$$e^{-\varepsilon_0} \leq \sqrt{\frac{\det(\mathbf{CC}^\mathsf{T})}{\det(\widetilde{\mathbf{C}}\widetilde{\mathbf{C}}^\mathsf{T})}} \leq e^{\varepsilon_0}.$$

**Claim 6.** *Let* $\mathbf{C}, \varepsilon_0,$ *and* $\delta_0$ *be as defined earlier. Let* $\mathbf{y} \sim \mathcal{N}(0,1)^m$. *If* $\mathbf{x}$ *is sampled either from* $\mathbf{Cy}$ *or* $\widetilde{\mathbf{C}}\mathbf{y}$, *then we have*

$$\Pr\left[\left|\mathbf{x}^\mathsf{T}(\mathbf{CC}^\mathsf{T})^{-1}\mathbf{x} - \mathbf{x}^\mathsf{T}(\widetilde{\mathbf{C}}\widetilde{\mathbf{C}}^\mathsf{T})^{-1}\mathbf{x}\right| \leq \varepsilon_0\right] \geq 1 - \delta_0.$$

*Proof of Claim 5.* The claim follows simply as in [13] after a slight modification. More concretely, we have $\det(\mathbf{CC}^\mathsf{T}) = \prod_i \sigma_i^2$, where $\sigma_1 \geq \cdots \geq \sigma_m \geq \sigma_{\mathsf{min}}(\mathbf{C})$ are the singular values of $\mathbf{C}$. Let $\widetilde{\sigma}_1 \geq \cdots \geq \widetilde{\sigma}_m \geq \sigma_{\mathsf{min}}(\widetilde{\mathbf{C}})$ be its singular value for $\widetilde{\mathbf{C}}$. The matrix $\mathbf{E}$ has only one singular value $\sqrt{1+\alpha}$. This is because $\mathbf{EE}^\mathsf{T} = (1+\alpha)\mathbf{vv}^\mathsf{T}$. To finish the proof of this claim, we use Theorem 8.

Since the singular values of $\mathbf{C} - \widetilde{\mathbf{C}}$ and $\widetilde{\mathbf{C}} - \mathbf{C}$ are the same, Lidskii's theorem (Theorem 8) gives $\sum_i(\sigma_i - \widetilde{\sigma}_i) \leq \sqrt{1+\alpha}$. Therefore, with probability $1 - \delta$,

$$\sqrt{\prod_{i:\widetilde{\sigma}_i \geq \sigma_i} \frac{\widetilde{\sigma}_i^2}{\sigma_i^2}} = \prod_{i:\widetilde{\sigma}_i \geq \sigma_i}\left(1 + \frac{\widetilde{\sigma}_i - \sigma_i}{\sigma_i}\right)$$

$$\leq \exp\left(\frac{\varepsilon}{32\sqrt{(1+\alpha)t\log(2/\delta)}\log(t/\delta)}\sum_i(\widetilde{\sigma}_i - \sigma_i)\right)$$

$$\leq e^{\varepsilon_0/2}.$$

The first inequality holds because $\mathbf{\Phi} \sim \mathcal{D}_R$ satisfies $(\alpha, \delta)$-JLP due to the choice of $t$ (second claim of Lemma 1). Since $\mathbf{C}$ and $\mathbf{A}$ have same rank, this implies that all the singular values of $\mathbf{C}$ are within a $(1 \pm \alpha)^{1/2}$ multiplicative factor of $\widehat{\mathbf{A}}$ due to a result by Sarlos [85]. In other words, $\sigma_i \geq \sigma_{\mathsf{min}}(\mathbf{C}) \geq (1-\alpha)^{1/2}\sigma_{\mathsf{min}}$. The case for all $i \in [m]$ when $\widetilde{\sigma}_i \leq \sigma_i$ follows similarly as the singular values of $\mathbf{E}$ and $-\mathbf{E}$ are the same. This completes the proof of Claim 5. $\qquad\square$

*Proof of Claim 6.* Without any loss of generality, we can assume $\mathbf{x} = \mathbf{Cy}$. The case for $\mathbf{x} = \widetilde{\mathbf{C}}\mathbf{y}$ is analogous. Let $\mathbf{C} - \widetilde{\mathbf{C}} = \mathbf{vu}^\mathsf{T}$. Note that $\mathbb{E}[(\mathbf{\Omega})_{i,j}] = 0$ for all $1 \leq i, j \leq m$ and $\mathsf{COV}((\mathbf{\Omega})_{i,j}) = 1$ if and only if $i = j$; and $0$ otherwise. First note that the following

$$\mathbf{x}^\mathsf{T}(\mathbf{CC}^\mathsf{T})^{-1}\mathbf{x} - \mathbf{x}^\mathsf{T}(\widetilde{\mathbf{C}}\widetilde{\mathbf{C}}^\mathsf{T})^{-1}\mathbf{x} = \mathbf{x}^\mathsf{T}(\mathbf{CC}^\mathsf{T})^{-1}(\widetilde{\mathbf{C}}\widetilde{\mathbf{C}}^\mathsf{T})(\widetilde{\mathbf{C}}\widetilde{\mathbf{C}}^\mathsf{T})^{-1}\mathbf{x} - \mathbf{x}^\mathsf{T}(\widetilde{\mathbf{C}}\widetilde{\mathbf{C}}^\mathsf{T})^{-1}\mathbf{x}$$

$$= \mathbf{x}^\mathsf{T}\left[(\mathbf{CC}^\mathsf{T})^{-1}(\mathbf{Cuv}^\mathsf{T} + \mathbf{vu}^\mathsf{T}\widetilde{\mathbf{C}}^\mathsf{T})(\widetilde{\mathbf{C}}\widetilde{\mathbf{C}}^\mathsf{T})^{-1}\right]\mathbf{x}.$$

Using the singular value decomposition of $\mathbf{C} = \mathbf{U_C}\mathbf{\Sigma_C}\mathbf{V_C}^\mathsf{T}$ and $\widetilde{\mathbf{C}} = \widetilde{\mathbf{U}}_\mathbf{C}\widetilde{\mathbf{\Sigma}}_\mathbf{C}\widetilde{\mathbf{V}}_\mathbf{C}^\mathsf{T}$, we have

$$\left(\mathbf{x}^\mathsf{T}(\mathbf{U_C}\mathbf{\Sigma_C}^{-1}\mathbf{V_C}^\mathsf{T})\mathbf{u}\right)\left(\mathbf{v}^\mathsf{T}(\widetilde{\mathbf{U}}_\mathbf{C}\widetilde{\mathbf{\Sigma}}_\mathbf{C}^{-2}\widetilde{\mathbf{U}}_\mathbf{C}^\mathsf{T})\mathbf{x}\right) + \left(\mathbf{x}^\mathsf{T}(\mathbf{U_C}\mathbf{\Sigma_C}^{-2}\mathbf{U_C}^\mathsf{T})\mathbf{v}\right)\left(\mathbf{u}^\mathsf{T}(\widetilde{\mathbf{V}}_\mathbf{C}\widetilde{\mathbf{\Sigma}}_\mathbf{C}^{-1}\widetilde{\mathbf{U}}_\mathbf{C}^\mathsf{T})\mathbf{x}\right)$$

Since $\mathbf{x} \sim \mathbf{Cy}$, where $\mathbf{y} \sim \mathcal{N}(0,1)^m$, we can write the above expression as $\tau_1\tau_2 + \tau_3\tau_4$, where

$$\tau_1 = \left(\mathbf{y}^\mathsf{T}\mathbf{C}^\mathsf{T}(\mathbf{U_C}\mathbf{\Sigma_C}^{-1}\mathbf{V_C}^\mathsf{T})\mathbf{u}\right) \qquad\qquad \tau_2 = \left(\mathbf{v}^\mathsf{T}(\widetilde{\mathbf{U}}_\mathbf{C}\widetilde{\mathbf{\Sigma}}_\mathbf{C}^{-2}\widetilde{\mathbf{U}}_\mathbf{C}^\mathsf{T})\mathbf{Cy}\right)$$

$$\tau_3 = \left(\mathbf{y}^\mathsf{T}\mathbf{C}^\mathsf{T}(\mathbf{U_C}\mathbf{\Sigma_C}^{-2}\mathbf{U_C}^\mathsf{T})\mathbf{v}\right) \qquad\qquad \tau_4 = \left(\mathbf{u}^\mathsf{T}(\widetilde{\mathbf{V}}_\mathbf{C}\widetilde{\mathbf{\Sigma}}_\mathbf{C}^{-1}\widetilde{\mathbf{U}}_\mathbf{C}^\mathsf{T})\mathbf{Cy}\right).$$

Now since $\|\widetilde{\mathbf{\Sigma}}_\mathbf{C}\|_2, \|\mathbf{\Sigma_C}\|_2 \geq \sigma_{\mathsf{min}}(\mathbf{C})$, plugging in the SVD of $\mathbf{C}$ and $\mathbf{C} - \widetilde{\mathbf{C}} = \mathbf{vu}^\mathsf{T}$, and that every term $\tau_i$ in the above expression is a linear combination of a Gaussian, i.e., each term is distributed as

> ### PRIVATE-FROBENIUS-LRF
>
> **Initialization.** Set $\eta = \max\left\{k, \alpha^{-1}\right\} t = O(\eta\alpha^{-1}\log(k/\delta)), v = O(\eta\alpha^{-2}\log(k/\delta))$, and $\rho = \sqrt{(1+\alpha)\ln(1/\delta)}/\varepsilon$. Sample $\mathbf{\Phi} \in \mathbb{R}^{m \times t}$ from $\mathcal{D}_R$ as in Lemma 1 and $\mathbf{S} \in \mathbb{R}^{v \times n}$ from $\mathcal{D}_R$ as in Lemma 2.
>
> **Computing the factorization.** On input the matrix $\mathbf{A}$,
> 1. Sample $\mathbf{N}_1 \sim \mathcal{N}(0, \rho^2)^{m \times t}, \mathbf{N}_2 \sim \mathcal{N}(0, \rho^2)^{v \times n}$.
> 2. Compute $\mathbf{Y} = \mathbf{A}\mathbf{\Phi} + \mathbf{N}_1$ and $\mathbf{Z} = \mathbf{S}\mathbf{A} + \mathbf{N}_2$.
> 3. Compute a matrix $\mathbf{U} \in \mathbb{R}^{m \times t}$ whose columns are an orthonormal basis for the column space of $\mathbf{Y}$.
> 4. Compute the singular value decomposition of $\mathbf{S}\mathbf{U} \in \mathbb{R}^{v \times t}$. Let it be $\widetilde{\mathbf{U}}\widetilde{\mathbf{\Sigma}}\widetilde{\mathbf{V}}^{\mathsf{T}}$.
> 5. Compute the singular value decomposition of $\widetilde{\mathbf{V}}\widetilde{\mathbf{\Sigma}}^{\dagger}[\widetilde{\mathbf{U}}^{\mathsf{T}}\mathbf{Z}]_k$. Let it be $\mathbf{U}'\mathbf{\Sigma}'\mathbf{V}'^{\mathsf{T}}$.
> 6. Output $\widetilde{\mathbf{U}} = \mathbf{U}\mathbf{U}', \widetilde{\mathbf{\Sigma}} = \mathbf{\Sigma}'$ and $\widetilde{\mathbf{V}} = \mathbf{V}'$.

Figure 3: Differentially private LRF Under $\mathsf{Priv}_2$

per $\mathcal{N}(0, \|\tau_i\|^2)$, we have the following:

$$\|\tau_1\|_2 = \|(\mathbf{V_C}\mathbf{\Sigma_C}\mathbf{U_C^{\mathsf{T}}})(\mathbf{U_C}\mathbf{\Sigma_C^{-1}}\mathbf{V_C^{\mathsf{T}}})\mathbf{u}\|_2 \le \|\mathbf{u}\|_2 \le \sqrt{1+\alpha},$$

$$\|\tau_2\|_2 = \|\mathbf{v}^{\mathsf{T}}(\widetilde{\mathbf{U}}_{\mathbf{C}}\widetilde{\mathbf{\Sigma}}_{\mathbf{C}}^{-2}\widetilde{\mathbf{U}}_{\mathbf{C}}^{\mathsf{T}})(\widetilde{\mathbf{U}}_{\mathbf{C}}\widetilde{\mathbf{\Sigma}}_{\mathbf{C}}\widetilde{\mathbf{V}}_{\mathbf{C}}^{\mathsf{T}} - \mathbf{v}\mathbf{u}^{\mathsf{T}})\|_2$$

$$\le \|\mathbf{v}^{\mathsf{T}}(\widetilde{\mathbf{U}}_{\mathbf{C}}\widetilde{\mathbf{\Sigma}}_{\mathbf{C}}^{-2}\widetilde{\mathbf{U}}_{\mathbf{C}}^{\mathsf{T}})\widetilde{\mathbf{U}}_{\mathbf{C}}\widetilde{\mathbf{\Sigma}}_{\mathbf{C}}\widetilde{\mathbf{U}}_{\mathbf{C}}^{\mathsf{T}}\|_2 + \|\mathbf{v}^{\mathsf{T}}(\widetilde{\mathbf{U}}_{\mathbf{C}}\widetilde{\mathbf{\Sigma}}_{\mathbf{C}}^{-2}\widetilde{\mathbf{U}}_{\mathbf{C}}^{\mathsf{T}})\mathbf{v}\mathbf{u}^{\mathsf{T}}\|_2 \le \frac{1}{\sigma_{\mathsf{min}}(\mathbf{C})} + \frac{\sqrt{1+\alpha}}{\sigma_{\mathsf{min}}^2(\mathbf{C})},$$

$$\|\tau_3\|_2 = \|(\mathbf{V_C}\mathbf{\Sigma_C}\mathbf{U_C^{\mathsf{T}}})(\mathbf{U_C}\mathbf{\Sigma_C^{-2}}\mathbf{U_C^{\mathsf{T}}})\mathbf{v}\|_2 \le \|\mathbf{\Sigma_C^{-1}}\|_2 \le \frac{1}{\sigma_{\mathsf{min}}(\mathbf{C})},$$

$$\|\tau_4\|_2 = \|\mathbf{u}^{\mathsf{T}}(\widetilde{\mathbf{V}}_{\mathbf{C}}\widetilde{\mathbf{\Sigma}}_{\mathbf{C}}^{-1}\widetilde{\mathbf{U}}_{\mathbf{C}}^{\mathsf{T}})(\widetilde{\mathbf{U}}_{\mathbf{C}}\widetilde{\mathbf{\Sigma}}_{\mathbf{C}}\widetilde{\mathbf{V}}_{\mathbf{C}}^{\mathsf{T}} - \mathbf{v}\mathbf{u}^{\mathsf{T}})\|_2$$

$$\le \|\mathbf{u}^{\mathsf{T}}(\widetilde{\mathbf{V}}_{\mathbf{C}}\widetilde{\mathbf{\Sigma}}_{\mathbf{C}}^{-1}\widetilde{\mathbf{U}}_{\mathbf{C}}^{\mathsf{T}})(\widetilde{\mathbf{U}}_{\mathbf{C}}\widetilde{\mathbf{\Sigma}}_{\mathbf{C}}\widetilde{\mathbf{V}}_{\mathbf{C}}^{\mathsf{T}}\|_2 + \|\mathbf{u}^{\mathsf{T}}(\widetilde{\mathbf{V}}_{\mathbf{C}}\widetilde{\mathbf{\Sigma}}_{\mathbf{C}}^{-1}\widetilde{\mathbf{U}}_{\mathbf{C}}^{\mathsf{T}})\mathbf{v}\|_2 \le \sqrt{1+\alpha} + \frac{\sqrt{1+\alpha}}{\sigma_{\mathsf{min}}(\mathbf{C})}.$$

Using the concentration bound on the Gaussian distribution, each term, $\tau_1, \tau_2, \tau_3$, and $\tau_4$, is less than $\|\tau_i\| \ln(4/\delta_0)$ with probability $1 - \delta_0/2$. The second claim now follows because with probability $1 - \delta_0$,

$$\left|\mathbf{x}^{\mathsf{T}}(\mathbf{C}\mathbf{C}^{\mathsf{T}})^{-1}\mathbf{x} - \mathbf{x}^{\mathsf{T}}(\widetilde{\mathbf{C}}^{\mathsf{T}}\widetilde{\mathbf{C}})^{-1}\mathbf{x}\right| \le 2\left(\frac{\sqrt{1+\alpha}}{\sigma_{\mathsf{min}}(\mathbf{C})} + \frac{1+\alpha}{\sigma_{\mathsf{min}}^2(\mathbf{C})}\right)\ln(4/\delta_0) \le \varepsilon_0,$$

where the second inequality follows from the choice of $\sigma_{\mathsf{min}}$ and the fact that $\sigma_{\mathsf{min}}(\mathbf{C}) \ge (1 - \alpha)^{1/2}\sigma_{\mathsf{min}}$. $\qquad\square$

Lemma 11 follows by combining Claim 5 and Claim 6.

### E.2 Low Space Differentially private Low-rank Factorization Under $\mathsf{Priv}_2$

In Appendix E.1, we gave an optimal space algorithm for computing LRF under $\mathsf{Priv}_1$. However, we cannot use the algorithm of Theorem 3 to simultaneously prove differential privacy under $\mathsf{Priv}_2$ and get optimal additive error. This is because we need to perturb the input matrix by a noise proportional to $\min\left\{\sqrt{km}, \sqrt{kn}\right\}$ to preserve differential privacy under $\mathsf{Priv}_2$. As a result, the additive error would depend linearly on $\min\{m, n\}$. We show that by maintaining noisy sketches $\mathbf{Y}$ and $\mathbf{Z}$ and some basic linear algebra, we can have a differentially private algorithm that outputs an optimal error LRF of an $m \times n$ matrix under $\mathsf{Priv}_2$. More concretely, we prove the following theorem.

**Theorem 11.** *Let $m, n \in \mathbb{N}$ and $\alpha, \varepsilon, \delta$ be the input parameters. Let $k$ be the desired rank of the factorization, $s = \max\{m, n\}$, $u = \min\{m, n\}$, and $\eta = \max\left\{k, \alpha^{-1}\right\}$. Given a private input matrix $\mathbf{A} \in \mathbb{R}^{m \times n}$, the factorization $\widetilde{\mathbf{U}}, \widetilde{\mathbf{\Sigma}}, \widetilde{\mathbf{V}}$ outputted by the algorithm,* PRIVATE-FROBENIUS-LRF, *presented in Figure 3, is a $k$-rank factorization and satisfies the following properties:*

1. PRIVATE-FROBENIUS-LRF *is $(\varepsilon, \delta)$-differentially private under* $\mathsf{Priv}_2$.

2. Let $\mathbf{M}_k := \widetilde{\mathbf{U}}\widetilde{\mathbf{\Sigma}}\widetilde{\mathbf{V}}^{\mathsf{T}}$. With probability $9/10$ over the coins of PRIVATE-FROBENIUS-LRF,

$$\|\mathbf{A} - \mathbf{M}_k\|_F \leq (1+\alpha)\|\mathbf{A} - [\mathbf{A}]_k\|_F + O\left(\left(\sqrt{ks} + \sqrt{\frac{u\eta}{\alpha^2}}\right)\frac{\sqrt{\log(1/\delta)}}{\varepsilon}\right).$$

3. The space used by PRIVATE-FROBENIUS-LRF is $O((m + n\alpha^{-1})\eta\alpha^{-1}\log(k/\delta))$.

4. The time required to compute the factorization is $O\left((\mathsf{nn}(\mathbf{A})\log(1/\delta) + \frac{(m+n\alpha^{-2})\eta^2\log^2(k/\delta)}{\alpha^2} + \frac{\eta^3\log^3(k/\delta)}{\alpha^s 3}\right)$.

We present the algorithm when $m \geq n$, i.e., $s = m$ and $u = n$. The case when $m < n$ follows by symmetry. The space required by the algorithm is the space required to store $\mathbf{Y}$ and $\mathbf{Z}$, which is $mt + nv = O((m + n\alpha^{-1})\eta\alpha^{-1}\log(k/\delta))$. This proves part 3 of Theorem 11. For the running time of part 4 of Theorem 11, we have the following.

1. Computing the sketch $\mathbf{Y}$ requires $O(\mathsf{nn}(\mathbf{A})\log(1/\delta)) + mt^2$ time and computing the sketch $\mathbf{Z}$ requires $O(\mathsf{nn}(\mathbf{A})\log(k/\delta)) + nv^2$.

2. Computing the orthonormal basis $\mathbf{U}$ requires $mt^2 = O(m\eta^2\alpha^{-2}\log^2(k/\delta))$ time.

3. Computing a SVD of the matrix $\mathbf{SU}$ requires $vt^2 = O(\eta^3\alpha^3\log^3(k/\delta))$.

4. Computation of $[\widetilde{\mathbf{U}}^{\mathsf{T}}\mathbf{Z}]_k$ requires $O(n\eta^2\alpha^{-2}\log^2(k/\delta))$.

5. Computing a SVD in Step 4 requires $nt^2 = O(n\eta^2\alpha^{-2}\log^2(k/\delta))$ time.

Combining all these terms, we have our claim on running time.

### E.2.1 Privacy Proof of PRIVATE-FROBENIUS-LRF

The following lemma proves Part 1 of Theorem 11.

**Lemma 12.** PRIVATE-FROBENIUS-LRF is $(\varepsilon, \delta)$ differentially private.

*Proof.* We use the second claim of both Lemma 1 and Lemma 2, i.e., for all $\mathbf{D}$, $\|\mathbf{D}\mathbf{T}\|_F^2 \leq (1 - \alpha)\|\mathbf{D}\|_F^2$ for $\mathbf{T} \sim \mathcal{D}_A$ and $\|\mathbf{D}\mathbf{\Phi}\|_F^2 \leq (1-\alpha)\|\mathbf{D}\|_F^2$ for $\mathbf{\Phi} \sim \mathcal{D}_R$. Let $\mathbf{A}$ and $\mathbf{A}'$ be two neighboring matrices such that $\mathbf{E} = \mathbf{A} - \mathbf{A}'$ has Frobenius norm 1. Then $\|\mathbf{S}\mathbf{E}\|_F^2 \leq (1+\alpha)\|\mathbf{E}\|_F^2 = 1 + \alpha$. Publishing $\mathbf{Z}$ preserves $(\varepsilon, \delta)$-differential privacy follows from considering the vector form of the matrix $\mathbf{SA}$ and $\mathbf{N}_2$ and Theorem 5. Similarly, we use the fact that, for any matrix $\mathbf{C}$ of appropriate dimension, $\|\mathbf{\Phi}\mathbf{C}\|^2 \leq (1 - \alpha)\|\mathbf{C}\|_F^2$, to prove that publishing $\mathbf{A}\mathbf{\Phi} + \mathbf{N}_1$ preserves differential privacy. The lemma follows by applying Lemma 3 and Theorem 4. $\qquad\square$

### E.2.2 Correctness Proof of PRIVATE-FROBENIUS-LRF

We now prove Part 2 of Theorem 11. We first show the following result.

**Theorem 12.** Let $\mathbf{M}_k = \mathbf{U}\widetilde{\mathbf{V}}\widetilde{\mathbf{\Sigma}}^{\dagger}[\widetilde{\mathbf{U}}^{\mathsf{T}}\mathbf{Z}]_k$ be the product of the factorization outputted by the algorithm in Figure 3. Then with probability $1 - O(\delta)$ over $\mathbf{\Phi} \sim \mathcal{D}_R$ and $\mathbf{S} \sim \mathcal{D}_A$,

$$\|\mathbf{M}_k - \mathbf{A}\|_F \leq (1+\alpha)\|\mathbf{A} - [\mathbf{A}]_k\|_F + 3\|\mathbf{S}^{\dagger}\mathbf{N}_2\|_F + 2\|\mathbf{N}_1([\mathbf{A}]_k\mathbf{\Phi})^{\dagger}[\mathbf{A}]_k\|_F.$$

*Proof.* We prove the result by proving a series of results. We provide an upper and a lower bound on $\min_{\mathbf{X}, r(\mathbf{X}) \leq k}\|\mathbf{Y}\mathbf{X} - \mathbf{B}\|_F$ in terms of $\|\mathbf{A} - [\mathbf{A}]_k\|_F$ and the output of the algorithm.

**Lemma 13.** Let $\mathbf{A}$ be the input matrix. Let $\mathbf{\Phi} \sim \mathcal{D}_R, \mathbf{S} \sim \mathcal{D}_A$ be as in Figure 3. Let $\mathbf{Y} = \mathbf{\Phi}\mathbf{A} + \mathbf{N}_1$ and $\mathbf{B} = \mathbf{A} + \mathbf{S}^{\dagger}\mathbf{N}_2$ for $\mathbf{N}_1, \mathbf{N}_2$ as defined in Figure 3. Then with probability $1 - \delta$ over $\mathbf{\Phi} \sim \mathcal{D}_R$,

$$\min_{\substack{\mathbf{X} \\ r(\mathbf{X}) \leq k}}\|\mathbf{Y}\mathbf{X} - \mathbf{B}\|_F \leq (1+\alpha)\|\mathbf{A} - [\mathbf{A}]_k\|_F + \|\mathbf{S}^{\dagger}\mathbf{N}_2\|_F + \|\mathbf{N}_1([\mathbf{A}]_k\mathbf{\Phi})^{\dagger}[\mathbf{A}]_k\|_F.$$

*Proof.* Set $r = k$, $\mathbf{P} = [\mathbf{A}]_k^{\mathsf{T}}$, and $\mathbf{Q} = \mathbf{A}^{\mathsf{T}}$ in Lemma 1. Then using Lemma 1, we have

$$\|[\mathbf{A}]_k^{\mathsf{T}}\mathbf{X}' - \mathbf{A}^{\mathsf{T}}\| \leq (1 + \alpha)\min_{\mathbf{X}}\|[\mathbf{A}]_k^{\mathsf{T}}\mathbf{X} - \mathbf{A}^{\mathsf{T}}\|_F,$$

where $\mathbf{X}' = \operatorname{argmin}_{\mathbf{X}} \|\mathbf{\Phi}^{\mathsf{T}}([\mathbf{A}]_k^{\mathsf{T}}\mathbf{X} - \mathbf{A}^{\mathsf{T}})\|_F$. Let $[\mathbf{A}]_k = \mathbf{U}_k\mathbf{\Sigma}_k\mathbf{V}_k^{\mathsf{T}}$. Taking the transpose and the fact that the Frobenius norm is preserved under transpose and $\mathbf{X}' = (([\mathbf{A}]_k\mathbf{\Phi})^{\mathsf{T}})^{\dagger}(\mathbf{A}\mathbf{\Phi})^{\mathsf{T}}$, we have with probability $1 - \delta$ over $\mathbf{\Phi} \sim \mathcal{D}_R$,

$$\|\mathbf{A}\mathbf{\Phi}([\mathbf{A}]_k\mathbf{\Phi})^{\dagger}[\mathbf{A}]_k - \mathbf{A}\|_F \leq (1 + \alpha)\min_{\mathbf{X}}\|[\mathbf{A}]_k^{\mathsf{T}}\mathbf{X} - \mathbf{A}^{\mathsf{T}}\|_F \leq (1 + \alpha)\|\mathbf{A} - [\mathbf{A}]_k\|_F, \quad (18)$$

where the inequality follows by setting $\mathbf{X} = \mathbf{U}_k\mathbf{U}_k^{\mathsf{T}}$.

Moreover, since $([\mathbf{A}]_k\mathbf{\Phi})^{\dagger}[\mathbf{A}]_k$ has rank at most $k$, $\mathbf{B} = \mathbf{A} + \mathbf{S}^{\dagger}\mathbf{N}_2$, and $\mathbf{Y} = \mathbf{A}\mathbf{\Phi} + \mathbf{N}_1$, with probability $1 - \delta$ over $\mathbf{\Phi} \sim \mathcal{D}_R$,

$$\min_{\substack{\mathbf{X} \\ r(\mathbf{X}) \leq k}}\|\mathbf{Y}\mathbf{X} - \mathbf{B}\|_F \leq \|\mathbf{A}\mathbf{\Phi}([\mathbf{A}]_k\mathbf{\Phi})^{\dagger}[\mathbf{A}]_k + \mathbf{N}_1([\mathbf{A}]_k\mathbf{\Phi})^{\dagger}[\mathbf{A}]_k - \mathbf{B}\|_F$$

$$= \|\mathbf{A}\mathbf{\Phi}([\mathbf{A}]_k\mathbf{\Phi})^{\dagger}[\mathbf{A}]_k + \mathbf{N}_1([\mathbf{A}]_k\mathbf{\Phi})^{\dagger}[\mathbf{A}]_k - \mathbf{A} - \mathbf{S}^{\dagger}\mathbf{N}_2\|_F$$

$$\leq \|\mathbf{A}\mathbf{\Phi}([\mathbf{A}]_k\mathbf{\Phi})^{\dagger}[\mathbf{A}]_k - \mathbf{A}\|_F + \|\mathbf{N}_1([\mathbf{A}]_k\mathbf{\Phi})^{\dagger}[\mathbf{A}]_k\|_F + \|\mathbf{S}^{\dagger}\mathbf{N}_2\|_F \quad (19)$$

Combining equation (18) and equation (19), we have with probability $1 - \delta$ over $\mathbf{\Phi} \sim \mathcal{D}_R$,

$$\min_{\substack{\mathbf{X} \\ r(\mathbf{X}) \leq k}}\|\mathbf{Y}\mathbf{X} - \mathbf{B}\|_F \leq (1 + \alpha)\|\mathbf{A} - [\mathbf{A}]_k\|_F + \|\mathbf{N}_1([\mathbf{A}]_k\mathbf{\Phi})^{\dagger}[\mathbf{A}]_k\|_F + \|\mathbf{S}^{\dagger}\mathbf{N}_2\|_F. \quad (20)$$

This completes the proof of Lemma 13. $\qquad\square$

Lemma 13 relates $\min_{\mathbf{X}, r(\mathbf{X}) \leq k}\|\mathbf{Y}\mathbf{X} - \mathbf{B}\|$ with $(1 + \alpha)\|\mathbf{A} - [\mathbf{A}]_k\|_F$, $\|\mathbf{N}_1([\mathbf{A}]_k\mathbf{\Phi})^{\dagger}[\mathbf{A}]_k\|_F$, and $\|\mathbf{S}^{\dagger}\mathbf{N}_2\|_F$. Since $\mathbf{U}$ is the orthonormal basis for the column space of $\mathbf{Y}$, we further have

$$\min_{\substack{\mathbf{X} \\ r(\mathbf{X}) \leq k}}\|\mathbf{U}\mathbf{X} - \mathbf{B}\|_F \leq \min_{\substack{\mathbf{X} \\ r(\mathbf{X}) \leq k}}\|\mathbf{Y}\mathbf{X} - \mathbf{B}\|_F. \quad (21)$$

Combining equation (20) and equation (21), we have with probability $1 - \delta$ over $\mathbf{\Phi} \sim \mathcal{D}_R$,

$$\min_{\substack{\mathbf{X} \\ r(\mathbf{X}) \leq k}}\|\mathbf{U}\mathbf{X} - \mathbf{B}\|_F \leq (1 + \alpha)\|\mathbf{A} - [\mathbf{A}]_k\|_F + (\|\mathbf{N}_1([\mathbf{A}]_k\mathbf{\Phi})^{\dagger}[\mathbf{A}]_k\|_F + \|\mathbf{S}^{\dagger}\mathbf{N}_2\|_F). \quad (22)$$

**Lemma 14.** *Let* $\mathbf{U}, \mathbf{B}, \mathbf{A}, \mathbf{S}, \mathbf{N}_1$, *and* $\mathbf{N}_2$ *be as above, and let* $\widetilde{\mathbf{X}} = \operatorname{argmin}_{\mathbf{X}, r(\mathbf{X})=k}\|\mathbf{S}(\mathbf{U}\mathbf{X} - \mathbf{B})\|_F$. *Let* $\mathcal{D}_A$ *be a distribution that satisfies* $(\alpha, \delta)$-*subspace embedding. Then with probability* $1 - 4\delta$ *over* $\mathbf{S} \sim \mathcal{D}_A$,

$$\|(\mathbf{U}\widetilde{\mathbf{X}} - \mathbf{B})\|_F \leq (1 + \alpha)^2\|\mathbf{A} - [\mathbf{A}]_k\|_F + (1 + \alpha)(\|\mathbf{S}^{\dagger}\mathbf{N}_2\|_F + \|\mathbf{N}_1([\mathbf{A}]_k\mathbf{\Phi})^{\dagger}[\mathbf{A}]_k\|_F).$$

*Proof.* Set $p = k/\alpha$, $\mathbf{D} = \mathbf{U}$ and $\mathbf{E} = \mathbf{B}$ in the statement of Lemma 2. Let us restrict our attention to rank $k$ matrices $\mathbf{X}$ and denote by $\widehat{\mathbf{X}} = \operatorname{argmin}_{\mathbf{X}, r(\mathbf{X})=k}\|\mathbf{U}\mathbf{X} - \mathbf{E}\|_F$ and $\widetilde{\mathbf{X}} = \operatorname{argmin}_{\mathbf{X}, r(\mathbf{X})=k}\|\mathbf{S}(\mathbf{U}\mathbf{X} - \mathbf{B})\|_F$. Then we have with probability $1 - \delta$ over $\mathbf{S} \sim \mathcal{D}_A$,

$$(1 + \alpha)\min_{\substack{\mathbf{X} \\ r(\mathbf{X})=k}}\|\mathbf{U}\mathbf{X} - \mathbf{B}\|_F = \|\mathbf{U}\widehat{\mathbf{X}} - \mathbf{B}\|_F \geq (1 + \alpha)^{1/2}\|\mathbf{S}(\mathbf{U}\widehat{\mathbf{X}} - \mathbf{B})\|_F$$

$$\geq (1 + \alpha)^{1/2}\min_{\substack{\mathbf{X} \\ r(\mathbf{X})}}\|\mathbf{S}(\mathbf{U}\mathbf{X} - \mathbf{B})\|_F$$

$$= (1 + \alpha)^{1/2}\|\mathbf{S}(\mathbf{U}\widetilde{\mathbf{X}} - \mathbf{B})\|_F \geq \|(\mathbf{U}\widetilde{\mathbf{X}} - \mathbf{B})\|_F. \quad (23)$$

Combining equation (23) with equation (22), we have with probability $1 - 2\delta$ over $\mathbf{\Phi} \sim \mathcal{D}_R$ and $\mathbf{S} \sim \mathcal{D}_A$,

$$\|(\mathbf{U}\widetilde{\mathbf{X}} - \mathbf{B})\|_F \leq (1 + \alpha)^2\|\mathbf{A} - [\mathbf{A}]_k\|_F + (1 + \alpha)(\|\mathbf{S}^{\dagger}\mathbf{N}_2\|_F + \|\mathbf{N}_1([\mathbf{A}]_k\mathbf{\Phi})^{\dagger}[\mathbf{A}]_k\|_F). \quad (24)$$

This completes the proof of Lemma 14. $\qquad\square$

To finalize the proof of Theorem 12, we need to compute $\widetilde{\mathbf{X}} = \operatorname{argmin}_{\mathbf{X}, r(\mathbf{X}) \leq k}\|\mathbf{S}(\mathbf{U}\mathbf{X} - \mathbf{B})\|_F$ and lower bound $\|(\mathbf{U}\widetilde{\mathbf{X}} - \mathbf{B})\|_F$. Invoking [21, Lem 4.2] with $\mathbf{O} = \widetilde{\mathbf{U}}$ and $\mathbf{Z} = \mathbf{S}\mathbf{B}$, we get

$$[\widetilde{\mathbf{U}}^{\mathsf{T}}\mathbf{Z}]_k = [\widetilde{\mathbf{U}}^{\mathsf{T}}\mathbf{S}\mathbf{B}]_k = \operatorname*{argmin}_{\substack{\mathbf{X} \\ r(\mathbf{X}) \leq k}}\|\widetilde{\mathbf{U}}\mathbf{X} - \mathbf{S}\mathbf{B}\|_F.$$

This in particular implies that

$$\widetilde{\mathbf{X}} = \widetilde{\mathbf{V}}\widetilde{\mathbf{\Sigma}}^\dagger[\widetilde{\mathbf{U}}^\mathsf{T}\mathbf{Z}]_k = \underset{\substack{\mathbf{X}\\ r(\mathbf{X})\le k}}{\operatorname{argmin}} \|\mathbf{S}(\mathbf{U}\mathbf{X}-\mathbf{B})\|_F. \qquad (25)$$

Using equation (25) in equation (24), and the fact that $\mathbf{B} = \mathbf{A} + \mathbf{S}^\dagger\mathbf{N}_2$, we have the final result.

$$\|\mathbf{U}\widetilde{\mathbf{V}}\widetilde{\mathbf{\Sigma}}^\dagger[\widetilde{\mathbf{U}}^\mathsf{T}\mathbf{Z}]_k - \mathbf{A}\|_F - \|\mathbf{S}^\dagger\mathbf{N}_2\|_F \le \|\mathbf{U}\widetilde{\mathbf{V}}\widetilde{\mathbf{\Sigma}}^\dagger[\widetilde{\mathbf{U}}^\mathsf{T}\mathbf{S}\mathbf{A}]_k - \mathbf{B}\|_F$$
$$\le (1+\alpha)^2\|\mathbf{A} - [\mathbf{A}]_k\|_F + (1+\alpha)(\|\mathbf{S}^\dagger\mathbf{N}_2\|_F + \|\mathbf{N}_1([\mathbf{A}]_k\mathbf{\Phi})^\dagger[\mathbf{A}]_k\|_F).$$

This implies that

$$\|\mathbf{U}\widetilde{\mathbf{V}}\widetilde{\mathbf{\Sigma}}^\dagger[\widetilde{\mathbf{U}}^\mathsf{T}\mathbf{Z}]_k - \mathbf{A}\|_F \le (1+3\alpha)\|\mathbf{A} - [\mathbf{A}]_k\|_F + 3\|\mathbf{S}^\dagger\mathbf{N}_2\|_F + 2\|\mathbf{N}_1([\mathbf{A}]_k\mathbf{\Phi})^\dagger[\mathbf{A}]_k\|_F.$$

This completes the proof of Theorem 12. $\qquad\square$

**Lemma 15.** *Let* $\mathbf{N}_1 \sim \mathcal{N}(0,\rho^2)^{m\times t}$. *Then*

$$\|\mathbf{N}_1([\mathbf{A}]_k\mathbf{\Phi})^\dagger[\mathbf{A}]_k\|_F = O(\rho\sqrt{km})$$

*with probability* $99/100$ *over* $\mathbf{\Phi} \sim \mathcal{D}_R$.

*Proof.* Let $\mathbf{C} = \mathbf{N}_1([\mathbf{A}]_k\mathbf{\Phi})^\dagger[\mathbf{A}]_k$. Then $\mathbf{C}\mathbf{\Phi} = \mathbf{N}_1([\mathbf{A}]_k\mathbf{\Phi})^\dagger[\mathbf{A}]_k\mathbf{\Phi}$. Now $([\mathbf{A}]_k\mathbf{\Phi})^\dagger[\mathbf{A}]_k\mathbf{\Phi}$ is a projection unto a random subspace of dimension $k$. Since every entries of $\mathbf{N}_1$ is picked i.i.d. from $\mathcal{N}(0,\rho^2)$, $\mathbf{C}\mathbf{\Phi} = \mathbf{N}_1([\mathbf{A}]_k\mathbf{\Phi})^\dagger[\mathbf{A}]_k\mathbf{\Phi} = \widetilde{\mathbf{N}}_1$, where $\widetilde{\mathbf{N}}_1$ is an $m \times k$ matrix with every entries picked i.i.d. from $\mathcal{N}(0,\rho^2)$. This is because we can write $([\mathbf{A}]_k\mathbf{\Phi})^\dagger[\mathbf{A}]_k\mathbf{\Phi}$ is a projection unto a random subspace of dimension $k$. Using Lemma 4, this implies that

$$\mathbb{E}[\|\mathbf{C}\mathbf{\Phi}\|_F^2] = \mathbb{E}\left[\|\widetilde{\mathbf{N}}_1\|_F^2\right] = \sum_{i,j}\mathbb{E}[(\widetilde{\mathbf{N}}_1)_{ij}^2] = km\rho^2.$$

The result follows using Markov's inequality, the fact that $\|\mathbf{C}\mathbf{\Phi}\|_F^2 = (1+\alpha)\|\mathbf{C}\|_F^2$, and $\alpha < 1$. $\quad\square$

**Lemma 16.** *Let* $\mathbf{N}_2 \sim \mathcal{N}(0,\rho^2)^{v\times n}$. *Then* $\|\mathbf{S}^\dagger\mathbf{N}_2\|_F = O(\rho\sqrt{vn})$ *with probability* $99/100$ *over* $\mathbf{S} \sim \mathcal{D}_A$.

*Proof.* Let $\mathbf{C} = \mathbf{S}^\dagger\mathbf{N}_2$. Then $\mathbf{S}\mathbf{C} = \mathbf{S}\mathbf{S}^\dagger\mathbf{N}_2$. Now $\mathbf{S}\mathbf{S}^\dagger$ is a projection unto a random subspace of dimension $k$. Since every entries of $\mathbf{N}_2$ is picked i.i.d. from $\mathcal{N}(0,\rho^2)$, $\mathbf{S}\mathbf{C} = \widetilde{\mathbf{N}}_2$, where $\widetilde{\mathbf{N}}_1$ is an $v \times n$ matrix with every entries picked i.i.d. from $\mathcal{N}(0,\rho^2)$. Using Lemma 4, this implies that

$$\mathbb{E}[\|\mathbf{S}\mathbf{C}\|_F^2] = \mathbb{E}\left[\|\widetilde{\mathbf{N}}_2\|_F^2\right] = \sum_{i,j}\mathbb{E}[(\widetilde{\mathbf{N}}_2)_{ij}^2] = vn\rho^2.$$

The result follows using Markov's inequality and the fact that $\|\mathbf{S}\mathbf{C}\|_F^2 = (1+\alpha)\|\mathbf{C}\|_F^2$ and $\alpha \le 1$. $\quad\square$

Theorem 11 now follows from Theorem 12, Lemma 15, Lemma 16, and the choice of $\rho$ in Lemma 12.

# F  Differentially private LRF Under General Turnstile Model

In this section, we are interested in computing a low-rank factorization of a private matrix in the general turnstile update model while preserving differential privacy. In this setting, we are allowed only one pass over the private matrix, and by the end of the stream, we are required to output a low-rank factorization. In Appendix F.1, we give a differentially private low rank factorization under $\mathsf{Priv}_1$. We give a differentially private low rank factorization under a much stronger privacy guarantee, $\mathsf{Priv}_2$, in Appendix F.2.

<div style="border: 1px solid black; padding: 10px;">

<div align="center">PRIVATE-STREAMING-SPACE-OPTIMAL-LRF</div>

**Initialization.** Set $\eta = \max\{k, \alpha^{-1}\}$, and the dimension of random projections to be $t = O(\eta\alpha^{-1}\log(k/\delta)), v = O(\eta\alpha^{-2}\log(k/\delta))$. Let $\rho_1 = \sqrt{(1+\alpha)\ln(1/\delta)}/\varepsilon$ and $\rho_2 = (1+\alpha)\sqrt{\ln(1/\delta)}/\varepsilon$. Set $\sigma_{\mathsf{min}} = 16\log(1/\delta)\sqrt{t(1+\alpha)(1-\alpha)^{-1}\ln(1/\delta)}/\varepsilon$.

1. Sample $\boldsymbol{\Omega} \sim \mathcal{N}(0,1)^{m\times t}$. Let $\boldsymbol{\Phi} \in \mathbb{R}^{(m+n)\times m}, \boldsymbol{\Psi} \in \mathbb{R}^{t\times m}$ such that $\boldsymbol{\Phi}^{\mathsf{T}} \sim \mathcal{D}_R, \boldsymbol{\Psi} \sim \mathcal{D}_R$ satisfies Lemma 1.
2. Let $\mathbf{S} \in \mathbb{R}^{v\times m}, \mathbf{T} \in \mathbb{R}^{v\times(m+n)}$ such that $\mathbf{S} \sim \mathcal{D}_A, \mathbf{T}^{\mathsf{T}} \sim \mathcal{D}_A$ satisfies Lemma 2.
3. Sample $\mathbf{N}_1 \sim \mathcal{N}(0,\rho_1^2)^{t\times(m+n)}$ and $\mathbf{N}_2 \sim \mathcal{N}(0,\rho_2^2)^{v\times v}$. Define $\widehat{\boldsymbol{\Phi}} = t^{-1}\boldsymbol{\Phi}\boldsymbol{\Omega} \in \mathbb{R}^{(m+n)\times t}$.

**Update Stage.** Set $\widehat{\mathbf{A}} = \begin{pmatrix} \mathbf{0}^{m\times n} & \sigma_{\mathsf{min}}\mathbb{I}_m \end{pmatrix}$. Compute $\mathbf{Y}_c = \widehat{\mathbf{A}}\widehat{\boldsymbol{\Phi}}, \mathbf{Y}_r' = \boldsymbol{\Psi}\widehat{\mathbf{A}}$, and $\mathbf{Z}' = \mathbf{S}\widehat{\mathbf{A}}\mathbf{T}^{\mathsf{T}}$.

**Update rule.** When $(i_\tau, j_\tau, \Delta_\tau)$, where $(i_\tau, j_\tau) \in [m] \times [n]$ and $\Delta_\tau \in \mathbb{R}$, is streamed, update the matrices by the following rule: (i) $\mathbf{Y}_c = \mathbf{Y}_c + \mathbf{A}_\tau\widehat{\boldsymbol{\Phi}}$, (ii) $\mathbf{Y}_r' = \mathbf{Y}_r' + \boldsymbol{\Psi}\mathbf{A}_\tau$, and (iii) $\mathbf{Z}' = \mathbf{Z}' + \mathbf{S}\mathbf{A}_\tau\mathbf{T}^{\mathsf{T}}$, where $\mathbf{A}_\tau$ is an $m \times (n+m)$ matrix with only non-zero entry $\Delta_\tau$ in position $(i_\tau, j_\tau)$..

**Computing the factorization.** Once the matrix is streamed, we follow the following steps.

1. Compute $\mathbf{Y}_r = \mathbf{Y}_r' + \mathbf{N}_1 = \boldsymbol{\Psi}\widehat{\mathbf{A}} + \mathbf{N}_1$, and $\mathbf{Z} = \mathbf{Z}' + \mathbf{N}_2 = \mathbf{S}\widehat{\mathbf{A}}\mathbf{T}^{\mathsf{T}} + \mathbf{N}_2$.
2. Output FACTOR$(\mathbf{Y}_c, \mathbf{Y}_r, \mathbf{Z}, \mathbf{S}, \mathbf{T}, m, m+n, k, t, v)$.

**FACTOR**$(\mathbf{Y}_c, \mathbf{Y}_r, \mathbf{Z}, \mathbf{S}, \mathbf{T}, m, n, k, t, v)$

1. Compute a matrix $\mathbf{U} \in \mathbb{R}^{m\times t}$ whose columns are orthonormal basis for the column space of $\mathbf{Y}_c$ and matrix $\mathbf{V} \in \mathbb{R}^{t\times n}$ whose rows are the orthonormal basis for the row space of $\mathbf{Y}_r$.
2. Compute a SVD of $\mathbf{S}\mathbf{U} := \widetilde{\mathbf{U}}_s\widetilde{\boldsymbol{\Sigma}}_s\widetilde{\mathbf{V}}_s^{\mathsf{T}} \in \mathbb{R}^{v\times t}$ and a SVD of $\mathbf{V}\mathbf{T}^{\mathsf{T}} := \widetilde{\mathbf{U}}_t\widetilde{\boldsymbol{\Sigma}}_t\widetilde{\mathbf{V}}_t^{\mathsf{T}} \in \mathbb{R}^{t\times v}$.
3. Compute a SVD of $\widetilde{\mathbf{V}}_s\widetilde{\boldsymbol{\Sigma}}_s^{\dagger}[\widetilde{\mathbf{U}}_s^{\mathsf{T}}\mathbf{Z}\widetilde{\mathbf{V}}_t]_k\widetilde{\boldsymbol{\Sigma}}_t^{\dagger}\widetilde{\mathbf{U}}_t^{\mathsf{T}}$. Let it is be $\mathbf{U}'\boldsymbol{\Sigma}'\mathbf{V}'^{\mathsf{T}}$.
4. Output the matrix $\widetilde{\mathbf{U}} = \mathbf{U}\mathbf{U}'$ compromising of left singular vectors, diagonal matrix $\widetilde{\boldsymbol{\Sigma}} = \boldsymbol{\Sigma}'$, and the matrix $\widetilde{\mathbf{V}} = \mathbf{V}^{\mathsf{T}}\mathbf{V}'$ with right-singular vectors.

</div>

<div align="center">Figure 4: Differentially private LRF Under Priv$_1$ in Turnstile Update Model</div>

### F.1 Space Optimal Differentially private Low-rank Factorization Under Priv$_1$

The main idea behind the differentially private algorithm for low-rank factorization under Priv$_1$ in turnstile update model is that the corresponding algorithm (PRIVATE-SPACE-OPTIMAL-LRF) maintains linear sketches. It has been shown by Li *et al.* [63] that in general turnstile update model, it is better off to just use linear sketches. Together with our low space algorithm, this gives us the insight to develop a private algorithm in the general turnstile update model. Figure 4 gives the detail description of our algorithm. We show the following:

**Theorem 13.** *Let $m, n, k \in \mathbb{N}$ and $\alpha, \varepsilon, \delta$ be the input parameters. Let $s = \max\{m, n\}, u = \min\{m, n\}, \kappa = (1+\alpha)/(1-\alpha), \eta = \max\{k, \alpha^{-1}\}$, and $\sigma_{\mathsf{min}} = 16\log(1/\delta)\sqrt{t\kappa\ln(1/\delta)}/\varepsilon$. Given an $m \times n$ matrix $\mathbf{A}$ in a turnstile update model, PRIVATE-STREAMING-SPACE-OPTIMAL-LRF, described in Figure 4, outputs a factorization $\widetilde{\mathbf{U}}, \widetilde{\boldsymbol{\Sigma}}, \widetilde{\mathbf{V}}$ such that*

1. *PRIVATE-STREAMING-SPACE-OPTIMAL-LRF is $(3\varepsilon, 3\delta)$ differentially private under Priv$_1$.*

2. *Let $\mathbf{M}_k = \widetilde{\mathbf{U}}\widetilde{\boldsymbol{\Sigma}}\widetilde{\mathbf{V}}^{\mathsf{T}}$. With probability $9/10$ over the random coins of PRIVATE-STREAMING-SPACE-OPTIMAL-LRF,*

$$\|(\mathbf{A} \quad \mathbf{0}) - \mathbf{M}_k\|_F \le (1+\alpha)\|\mathbf{A} - [\mathbf{A}]_k\|_F + O(\sigma_{\mathsf{min}}\sqrt{u} + \varepsilon^{-1}\sqrt{ks\ln(1/\delta)}),$$

3. *The space used by PRIVATE-STREAMING-SPACE-OPTIMAL-LRF is $O((m+n)\eta\alpha^{-1}\log(k/\delta))$.*

---

<div style="border:1px solid black; padding:10px">

<p align="center">PRIVATE-STREAMING-FROBENIUS-LRF</p>

**Initialization.** Set $\eta = \max\left\{k, \alpha^{-1}\right\}$ $t = O(\eta\alpha^{-1}\log(k/\delta)), v = O(\eta\alpha^{-2}\log(k/\delta))$, and $\rho = \sqrt{(1+\alpha)\ln(1/\delta)}/\varepsilon$. Sample $\mathbf{N}_1 \sim \mathcal{N}(0,\rho^2)^{m\times t}, \mathbf{N}_2 \sim \mathcal{N}(0,\rho^2)^{v\times n}$. Sample $\mathbf{\Phi} \in \mathbb{R}^{m\times t}$ from $\mathcal{D}_R$ as in Lemma 1 and $\mathbf{S} \in \mathbb{R}^{v\times n}$ from $\mathcal{D}_R$ as in Lemma 2. Initialize an all zero $m \times t$ matrix $\mathbf{Y}'$ and an all zero $v \times n$ matrix $\mathbf{Z}'$.

**Update rule.** Suppose at time $\tau$, the stream is $(i_\tau, j_\tau, \Delta_\tau)$, where $(i_\tau, j_\tau) \in [m] \times [n]$. Let $\mathbf{A}_\tau$ be a matrix with only non-zero entry $\Delta_\tau$ in position $(i_\tau, j_\tau)$. Update the matrices by the following rule: $\mathbf{Y}' = \mathbf{Y}' + \mathbf{A}_\tau\mathbf{\Phi}$ and $\mathbf{Z}' = \mathbf{Z}' + \mathbf{S}\mathbf{A}_\tau$.

**Computing the factorization.** Once the matrix is streamed, we follow the following steps.

1. Compute $\mathbf{Y} = \mathbf{Y}' + \mathbf{N}_1$ and $\mathbf{Z} = \mathbf{Z}' + \mathbf{N}_2$.
2. Compute a matrix $\mathbf{U} \in \mathbb{R}^{m\times t}$ whose columns are an orthonormal basis for the column space of $\mathbf{Y}$.
3. Compute the singular value decomposition of $\mathbf{SU} \in \mathbb{R}^{v\times t}$. Let it be $\widetilde{\mathbf{U}}\widetilde{\mathbf{\Sigma}}\widetilde{\mathbf{V}}^\mathsf{T}$.
4. Compute the singular value decomposition of $\widetilde{\mathbf{V}}\widetilde{\mathbf{\Sigma}}^\dagger[\widetilde{\mathbf{U}}^\mathsf{T}\mathbf{Z}]_k$. Let it be $\mathbf{U}'\mathbf{\Sigma}'\mathbf{V}'^\mathsf{T}$.
5. Output $\widetilde{\mathbf{U}} = \mathbf{U}\mathbf{U}'$, $\widetilde{\mathbf{\Sigma}} = \mathbf{\Sigma}'$ and $\widetilde{\mathbf{V}} = \mathbf{V}'$.

</div>

<p align="center">Figure 5: Differentially private LRF Under $\mathsf{Priv}_2$ in Turnstile Update Model</p>

*4. The initialization time is $O((m+n)u\log(k/\delta))$ and the total computational time is*

$$O\left(\mathsf{nn}(\mathbf{A})\log(1/\delta) + \frac{(m+n)\eta^2\log^2(k/\delta)}{\alpha^2} + \frac{\eta^3\log^3(k/\delta)}{\alpha^5}\right).$$

*Proof.* Part 3 follows immediately by setting the values of $t$ and $v$. Part 4 of Theorem 13 requires some computation. More precisely, we have the following. Computing $\mathbf{Y}_c$ requires $O(\mathsf{nn}(\mathbf{A})\log(1/\delta)) + mt^2$ time and computing $\mathbf{Y}_r$ requires $O(\mathsf{nn}(\mathbf{A})\log(1/\delta)) + (m+n)t^2$ time. Computing $\mathbf{U}$ and $\mathbf{V}$ requires $O((m+n)\eta^2\alpha^{-2}\log^2(k/\delta))$ time. Computing a SVD of the matrix $\mathbf{SU}$ and $\mathbf{TV}^\mathsf{T}$ requires $vt^2 + tv^2 = O(k^3\alpha^{-5}\log^2(k/\delta))$. Computing $\mathbf{Z}$ requires $O(\mathsf{nn}(\mathbf{A})\log(k/\delta) + \eta^2/\alpha^2\log^2(k/\delta) + n\eta^2\alpha^{-2}\log^2(k/\delta))$ Computation of $[\widetilde{\mathbf{U}}_s^\mathsf{T}\mathbf{Z}\widetilde{\mathbf{V}}]_k$ requires $O_\delta(\mathsf{nn}(\mathbf{A})) + tv^2 = O(\mathsf{nn}(\mathbf{A}) + k^3\alpha^{-5}\log^3(k/\delta))$ time. Computation of the last SVD requires $O((m+n)\eta^2\alpha^{-2}\log^2(k/\delta))$ time. Combining all these terms, we have our claim on the running time.

Furthermore, combining Lemma 4, Lemma 7, Lemma 8, Claim 2, Claim 3, and Lemma 9, we have part 2 while part 1 follows from Lemma 11. This completes the proof of Theorem 13. □

## F.2 Differentially private Low-rank Factorization Under $\mathsf{Priv}_2$ in Turnstile Update Model

We describe and analyze the algorithm when $m \geq n$, i.e., $s = m$ and $u = n$ in the theorem that follows. The case when $m < n$ follows by symmetry. We prove the following.

**Theorem 14.** *Let $m, n \in \mathbb{N}$ and $\alpha, \varepsilon, \delta$ be the input parameters. Let $k$ be the desired rank of the factorization and $\eta = \max\left\{k, \alpha^{-1}\right\}$. Let $s = \max\{m, n\}$ and $u = \min\{m, n\}$. Given a private input matrix $\mathbf{A} \in \mathbb{R}^{m\times n}$ recieved in a turnstile model, the factorization $\widetilde{\mathbf{U}}, \widetilde{\mathbf{\Sigma}}, \widetilde{\mathbf{V}}$ outputted by the algorithm, PRIVATE-STREAMING-FROBENIUS-LRF, presented in Figure 5, is a $k$-rank factorization and satisfies the following properties:*

1. PRIVATE-STREAMING-FROBENIUS-LRF *is $(\varepsilon, \delta)$-differentially private under $\mathsf{Priv}_2$.*

2. *Let $\mathbf{M}_k := \widetilde{\mathbf{U}}\widetilde{\mathbf{\Sigma}}\widetilde{\mathbf{V}}^\mathsf{T}$. With probability $9/10$ over the coins of PRIVATE-STREAMING-FROBENIUS-LRF,*

$$\|\mathbf{A} - \mathbf{M}_k\|_F \leq (1+\alpha)\|\mathbf{A} - [\mathbf{A}]_k\|_F + O\left(\left(\sqrt{ks} + \sqrt{\frac{u\eta}{\alpha^2}}\right)\frac{\sqrt{\log(1/\delta)}}{\varepsilon}\right).$$

---

PRIVATE-COVARIANCE-LRF

**Initialization.** Set $\eta = \max\left\{k, \alpha^{-1}\right\}$ $t = O(\eta\alpha^{-1}\log(k/\delta)), v = O(\eta\alpha^{-2}\log(k/\delta))$, and $\rho = \sqrt{(1+\alpha)\ln(1/\delta)}/\varepsilon$. Sample $\mathbf{N}_1 \sim \mathcal{N}(0, \rho^2)^{n \times t}, \mathbf{N}_2 \sim \mathcal{N}(0, \rho^2)^{v \times n}$. Sample $\mathbf{\Phi} \in \mathbb{R}^{n \times t}$ from $\mathcal{D}_R$ as in Lemma 1 and $\mathbf{S} \in \mathbb{R}^{v \times n}$ from $\mathcal{D}_R$ as in Lemma 2. Initialize an all zero $n \times t$ matrix $\mathbf{Y}'$ and an all zero $v \times n$ matrix $\mathbf{Z}'$.

**Update rule.** Suppose at time $\tau$, the stream is an index-row tuple $(i_\tau, \mathbf{A}_{i_\tau})$, where $i_\tau \in [m]$ and $\mathbf{A}_{i_\tau} \in \mathbb{R}^n$. Let $\mathbf{A}_\tau$ be a matrix with only non-zero row $\mathbf{A}_{i_\tau}$ in the row $i_\tau$. Update the matrices by the following rule: $\mathbf{Y}' = \mathbf{Y}' + \mathbf{A}_\tau^\mathsf{T}\mathbf{A}_\tau\mathbf{\Phi}$ and $\mathbf{Z}' = \mathbf{Z}' + \mathbf{S}\mathbf{A}_\tau^\mathsf{T}\mathbf{A}_\tau$.

**Computing the factorization.** Once the matrix is streamed, we follow the following steps.

1. Compute $\mathbf{Y} = \mathbf{Y}' + \mathbf{N}_1$ and $\mathbf{Z} = \mathbf{Z}' + \mathbf{N}_2$.
2. Compute a matrix $\mathbf{U} \in \mathbb{R}^{m \times t}$ whose columns are an orthonormal basis for the column space of $\mathbf{Y}$.
3. Compute the singular value decomposition of $\mathbf{S}\mathbf{U} \in \mathbb{R}^{v \times t}$. Let it be $\widetilde{\mathbf{U}}\widetilde{\mathbf{\Sigma}}\widetilde{\mathbf{V}}^\mathsf{T}$.
4. Compute the singular value decomposition of $\widetilde{\mathbf{V}}\widetilde{\mathbf{\Sigma}}^\dagger[\widetilde{\mathbf{U}}^\mathsf{T}\mathbf{Z}]_k$. Let it be $\mathbf{U}'\mathbf{\Sigma}'\mathbf{V}'^\mathsf{T}$.
5. Output $\widetilde{\mathbf{U}} = \mathbf{U}\mathbf{U}', \widetilde{\mathbf{\Sigma}} = \mathbf{\Sigma}'$ and $\widetilde{\mathbf{V}} = \mathbf{V}'$.

---

Figure 6: Differentially private Covariance Approximation Under $\mathsf{Priv}_2$ in Row-wise Update Model

3. *The space used by* PRIVATE-STREAMING-FROBENIUS-LRF *is* $O((m + n\alpha^{-1})\eta\alpha^{-1}\log(k/\delta))$.

4. *The time required to compute the factorization is*

$$O\left((\mathsf{nn}(\mathbf{A})\log(1/\delta) + \frac{(m + n\alpha^{-2})\eta^2\log^2(k/\delta)}{\alpha^2} + \frac{\eta^3\log^3(k/\delta)}{\alpha^{s3}}\right).$$

*Proof.* The space required by the algorithm is the space required to store $\mathbf{Y}$ and $\mathbf{Z}$, which is $mt + nv = O((m + n\alpha^{-1})k\alpha^{-1}\log k\log(1/\delta))$. This proves part 3 of Theorem 14. For the running time of part 4 of Theorem 14, we have the following. Computing the sketch $\mathbf{Y}$ requires $O(\mathsf{nn}(\mathbf{A})\log(k/\delta)) + mt^2$ time and computing the sketch $\mathbf{Z}$ requires $O(\mathsf{nn}(\mathbf{A})\log(k/\delta)) + nv^2$. Computing the orthonormal basis $\mathbf{U}$ requires $mt^2 = O(m\eta^2\alpha^{-2}\log^2(k/\delta))$ time. Computing a SVD of the matrix $\mathbf{S}\mathbf{U}$ requires $vt^2 = O(\eta^3\alpha^3\log^3(k/\delta))$. Computation of $[\widetilde{\mathbf{U}}^\mathsf{T}\mathbf{Z}]_k$ requires $O(n\eta^2\alpha^{-2}\log^2(k/\delta))$. Computing a SVD in Step 4 requires $nt^2 = O(n\eta^2\alpha^{-2}\log^2(k/\delta))$ time. Combining all these terms, we have our claim on running time.

Part 2 follows from Lemma 16, Lemma 15, and Theorem 12. Part 1 follows from Lemma 12. This completes the proof of Theorem 14. $\square$

# G   Case Study: Normalized Row Matrices

An important class of matrices is matrices with normalized rows. In this section, we give a bound on such matrices. Figure 6 is the detailed description of the algorithm. It receives the matrix row-wise and computes the low-rank factorization.

**Theorem 15.** *Let* $m, n \in \mathbb{N}$ *and* $\alpha, \varepsilon, \delta$ *be the input parameters (with* $m > n$*). Let* $k$ *be the desired rank of the factorization and* $\eta = \max\left\{k, \alpha^{-1}\right\}$*. Given a private input matrix* $\mathbf{A} \in \mathbb{R}^{m \times n}$ *recieved in a row wise update model, the factorization* $\widetilde{\mathbf{U}}, \widetilde{\mathbf{\Sigma}}, \widetilde{\mathbf{V}}$ *outputted by the algorithm,* PRIVATE-COVARIANCE-LRF, *presented in Figure 5, is a* $k$*-rank factorization and satisfies the following properties:*

1. PRIVATE-COVARIANCE-LRF *is* $(\varepsilon, \delta)$*-differentially private under* $\mathsf{Priv}_2$*.*

2. *Let* $\mathbf{M}_k := \widetilde{\mathbf{U}}\widetilde{\mathbf{\Sigma}}\widetilde{\mathbf{V}}^{\mathsf{T}}$. *With probability* $9/10$ *over the coins of* PRIVATE-COVARIANCE-LRF,

$$\|\mathbf{A}^{\mathsf{T}}\mathbf{A} - \mathbf{M}_k\|_F \leq (1+\alpha)\|\mathbf{A}^{\mathsf{T}}\mathbf{A} - [\mathbf{A}^{\mathsf{T}}\mathbf{A}]_k\|_F + O\left(\frac{\sqrt{n\eta \log(1/\delta)}}{\varepsilon\alpha}\right).$$

3. *The space used by* PRIVATE-COVARIANCE-LRF *is* $O(n\alpha^{-2}\eta \log(k/\delta))$.

*Proof.* The space required by the algorithm is the space required to store $\mathbf{Y}$ and $\mathbf{Z}$, which is $mt + nv = O((m + n\alpha^{-1})k\alpha^{-1}\log k \log(1/\delta))$. This proves part 3 of Theorem 15.

Part 2 follows from Lemma 16, Lemma 15, and Theorem 12. For part 1, first notice that since every row has bounded norm 1, the sensitivity of the function $\mathbf{A}^{\mathsf{T}}\mathbf{A}$ is at most 1; i.e., the sensitivity of the vector form of $\mathbf{A}^{\mathsf{T}}\mathbf{A}$ is at most 1. Part 1 then follows from Lemma 12. This completes the proof of Theorem 15. □

## H   Case Study 2: Low-rank factorization Under Continual Release Model

In this section, we are interested in computing a low-rank factorization of a private matrix in the continual release model while preserving differential privacy. In this setting, we are allowed only one pass over the private matrix, and at every time epoch, we are required to output a low-rank factorization (see Definition 3 for a formal definition). In Appendix H.1, we give a differentially private low rank factorization under $\mathsf{Priv}_1$. We give a differentially private low rank factorization under a much stronger privacy guarantee, $\mathsf{Priv}_2$, in Appendix H.2.

In past, there are known algorithms for converting any "one-shot" algorithm for any monotonic function to an algorithm that continually release the output [37]. Since optimization function like low-rank factorization are not monotonic, it is not clear whether we can use the generic transformation. Our algorithm generates and maintains linear sketches during the updates and later compute low-rank factorization using these sketches. This allows us to use the generic transformation to maintain the updates. For computing the factorization, we collect all the sketches for any range using range queries.

### H.1   Differentially Private Continual Release Low Rank Factorization Under $\mathsf{Priv}_2$

We start by giving a differentially private algorithm under $\mathsf{Priv}_2$ that continually release a low rank factorization. We first give an overview of our algorithm with the details of the algorithm appearing in Figure 7.

The idea behind our algorithm for continual release is the fact that the factorization stage only uses a small space sketches of the matrix and the sketches are linear sketches. Since the sketches are linear, we can use the binary tree mechanism [19, 37] to get low-rank factorization under continual release model. The algorithm stores the sketches of matrix generated at various time epochs in the form of a binary tree. Every leaf node $\tau$ stores the sketches of $\mathbf{A}_\tau$, where $\mathbf{A}_\tau$ is the stream at time $\tau$. The root node stores the sketch of the entire matrix streamed in $[0, T]$, and every other node $\mathsf{n}$ stores the sketch corresponding to the updates in a time range represented by the leaves of the subtree rooted at $\mathsf{n}$, i.e., $\widehat{\mathbf{Y}}_i$ and $\widehat{\mathbf{Z}}_i$ stores sketches involving $2^i$ updates to $\mathbf{A}$. If a query is to compute the low-rank factorization of the matrix from a particular time range $[1, \tau]$, we find the nodes that uniquely cover the time range $[1, \tau]$. We then use the value of $\mathbf{Y}(\tau)$ and $\mathbf{Z}(\tau)$ formed using those nodes to compute the low-rank factorization. From the binary tree construction, every time epoch appears in exactly $O(\log T)$ nodes (from the leaf to the root node). Moreover, every range $[1, \tau]$ appears in at most $O(\log T)$ nodes of the tree (including leaves and root node). A straightforward application of the analysis of Chan *et al.* [19] to Theorem 14 gives us the following

**Theorem 16.** *Let* $\mathbf{A}$ *be an* $m \times n$ *matrix with* $\mathsf{nn}(\mathbf{A})$ *non-zero entries with* $m \leq n$. *Let* $\eta = \max\{k, \alpha^{-1}\}$. *Then there is an* $(\varepsilon, \delta)$-*differentially private algorithm*, PRIVATE-FROBENIUS-CONTINUAL-LRF *defined in Figure 7, under* $\mathsf{Priv}_2$ *that receives* $\mathbf{A}$ *as a stream and outputs a rank-$k$ factorization* $\widetilde{\mathbf{U}} := \widehat{\mathbf{U}}_k(\tau), \widetilde{\mathbf{\Sigma}} := \widehat{\mathbf{\Sigma}}_k(\tau), \widetilde{\mathbf{V}} := \widehat{\mathbf{V}}_k(\tau)$ *under the continual release for $T$ time epochs*

<div style="border:1px solid black; padding:10px">

<div align="center">PRIVATE-FROBENIUS-CONTINUAL-LRF</div>

**Input:** A time upper bound T , privacy parameters $\varepsilon, \delta$, and a stream $\mathbf{s} \in \mathbb{R}^T$.

**Output:** At each time step $\tau$, output a factorization $\widehat{\mathbf{U}}_k(\tau), \widehat{\mathbf{\Sigma}}_k(\tau)$, and $\widehat{\mathbf{V}}_k(\tau)$.

**Initialization:** Set $t, v, \mathbf{\Phi}, \mathbf{S}$ as in Figure 3. Every $\widehat{\mathbf{Y}}_i$ and $\widehat{\mathbf{Z}}_i$ are initialized to an all zero matrices for $i \in [\log T]$. Set $\varepsilon' = \varepsilon/\sqrt{\log T}, \delta' = \delta/2\log T$ and $\rho = \sqrt{(1+\alpha)\ln(1/\delta)}/\varepsilon'$.

**Estimating the** LRF **at time** $t$. On receiving an input $(\mathsf{r}, \mathsf{c}, \mathbf{s}_\tau)$ where $\mathbf{s}_\tau \in \mathbb{R}$ at $1 \leq \tau \leq T$, form a matrix $\mathbf{A}_\tau \in \mathbb{R}^{m \times n}$ which is an all zero matrix except with only non-zero entry $\mathbf{s}_\tau$ at location $(\mathsf{r}, \mathsf{c}) \in [m] \times [n]$.

1. Compute $i := \min\{j : \tau_j \neq 0\}$, where $\tau = \sum_j \tau_j \cdot 2^j$ is the binary expansion of $\tau$.
2. Compute $\widehat{\mathbf{Y}}_i := \mathbf{A}_\tau \mathbf{\Phi} + \sum_{j<i} \widehat{\mathbf{Y}}_j$ and $\widehat{\mathbf{Z}}_i := \mathbf{S}\mathbf{A}_\tau + \sum_{j<i} \widehat{\mathbf{Z}}_j$.
3. For $j := 0, \cdots, i-1$, set $\mathbf{Y}_j = \widehat{\mathbf{Y}}_j = \mathbf{0}$ and $\mathbf{Z}_j = \widehat{\mathbf{Z}}_j = \mathbf{0}$. Compute $\mathbf{Y}_i = \widehat{\mathbf{Y}}_i + \mathcal{N}(0, \rho^2)^{\times n}$ and $\mathbf{Z}_i = \widehat{\mathbf{Z}}_i + \mathcal{N}(0, \rho^2)^{m \times v}$. Compute $\mathbf{Y}(\tau) = \sum_{j:\tau_j=1} \mathbf{Y}_j$ and $\mathbf{Z}(\tau) = \sum_{j:\tau_j=1} \mathbf{Z}_j$.
4. Compute a matrix $\mathbf{U} \in \mathbb{R}^{m \times t}$ whose columns are an orthonormal basis for the column space of $\mathbf{Y}(\tau)$.
5. Compute the singular value decomposition of $\mathbf{S}\mathbf{U} \in \mathbb{R}^{v \times t}$. Let it be $\widetilde{\mathbf{U}}\widetilde{\mathbf{\Sigma}}\widetilde{\mathbf{V}}^\mathsf{T}$.
6. Compute the singular value decomposition of $\widetilde{\mathbf{V}}\widetilde{\mathbf{\Sigma}}^\dagger\widetilde{\mathbf{U}}^\mathsf{T}[\widetilde{\mathbf{U}}\widetilde{\mathbf{U}}^\mathsf{T}\mathbf{Z}(\tau)]_k$. Let it be $\mathbf{U}'\mathbf{\Sigma}'\mathbf{V}'^\mathsf{T}$.
7. Output $\widehat{\mathbf{U}}_k(\tau) := \mathbf{U}\mathbf{U}', \widehat{\mathbf{\Sigma}}_k(\tau) := \mathbf{\Sigma}'$ and $\widehat{\mathbf{V}}_k(\tau) := \mathbf{V}'$.
8. Let $\mathbf{M}_k(\tau) := \mathbf{U}_k(\tau)\mathbf{\Sigma}_k(\tau)\mathbf{V}_k(\tau)^\mathsf{T}$.

</div>

Figure 7: Differentially private Low-rank Factorization Under Continual Release

*such that, with probability* $9/10$,

$$\|\mathbf{A} - \mathbf{M}_k\|_F \leq (1+\alpha)\|\mathbf{A} - [\mathbf{A}]_k\|_F + O\left(\left(\sqrt{km} + \sqrt{\frac{n\eta}{\alpha^2}}\right)\frac{\sqrt{\log(1/\delta)}\log T}{\varepsilon}\right)$$

*where* $\mathbf{M}_k = \widetilde{\mathbf{U}}\widetilde{\mathbf{\Sigma}}\widetilde{\mathbf{V}}^\mathsf{T}$ *and* $\mathbf{A}(\tau)$ *is the matrix received till time* $\tau$.

### H.2 Differentially Private Continual Release Low Rank Factorization Under $\mathsf{Priv}_1$

We can also convert the algorithm PRIVATE-SPACE-OPTIMAL-LRF to one that outputs a low-rank factorization under continual release by using less space than PRIVATE-CONTINUAL-FROBENIUS-LRF and secure under $\mathsf{Priv}_2$. We make the following changes to PRIVATE-CONTINUAL-FROBENIUS-LRF: (i) Initialize $(\widehat{\mathbf{Y}}_c)_i, (\widehat{\mathbf{Y}}_r)_i$, and $(\widehat{\mathbf{Z}})_i$ as we initialize $\mathbf{Y}_c, \widehat{\mathbf{Y}}_r$ and $\widehat{\mathbf{Z}}$ in Figure 4 for all $i \in [\log T]$, (ii) we maintain $(\mathbf{Y}_c)_j, (\widehat{\mathbf{Y}}_c)_j, (\mathbf{Y}_r)_j, (\widehat{\mathbf{Y}}_r)_j, \mathbf{Z}_j$, and $\widehat{\mathbf{Z}}_j$. A straightforward application of the analysis of Chan *et al.* [19] to Theorem 13 gives us the following theorem.

**Theorem 17.** *Let* $\mathbf{A}$ *be an* $m \times n$ *matrix with* $\mathsf{nn}(\mathbf{A})$ *non-zero entries with* $m \leq n$. *Let* $\eta = \max\{k, \alpha^{-1}\}$. *Let* $s = \max\{m, n\}, u = \min\{m, n\}, \kappa = (1+\alpha)/(1-\alpha)$, *and* $\sigma_{\mathsf{min}} = 16\log(1/\delta)\sqrt{t\kappa\ln(1/\delta)}/\varepsilon$. *Then there is an* $(\varepsilon, \delta)$-*differentially private algorithm under* $\mathsf{Priv}_1$ *that receives* $\mathbf{A}$ *as a stream and outputs a rank-k factorization* $\widetilde{\mathbf{U}} := \widehat{\mathbf{U}}_k(\tau), \widetilde{\mathbf{\Sigma}} := \widehat{\mathbf{\Sigma}}_k(\tau), \widetilde{\mathbf{V}} := \widehat{\mathbf{U}}_k(\tau)$ *under the continual release for T time epochs such that, with probability* $9/10$,

$$\|(\mathbf{A}(\tau) \quad \mathbf{0}) - \mathbf{M}_k\|_F \leq (1+\alpha)\|\mathbf{A}(\tau) - [\mathbf{A}(\tau)]_k\|_F + O\left(\left(\sigma_{\mathsf{min}}\sqrt{u} + \varepsilon^{-1}\sqrt{ks\ln(1/\delta)}\right)\log T\right),$$

*where* $\mathbf{M}_k = \widetilde{\mathbf{U}}\widetilde{\mathbf{\Sigma}}\widetilde{\mathbf{V}}^\mathsf{T}$ *and* $\mathbf{A}(\tau)$ *is the matrix received till time* $\tau$.

## I Space Lower Bound for Low-rank Factorization When $\gamma \neq 0$

This section is devoted to proving a lower bound on the space requirement for low-rank factorization with non-trivial additive error. It is well known that any private algorithm (not necessarily differentially

private) incurs an additive error $o(\sqrt{k(m+n)})$ [47] due to linear reconstruction attack. On the other hand, the only known space lower bound of Clarkson and Woodruff [21] holds when $\gamma = 0$; therefore, one might hope to construct an improve space algorithm when we allow $\gamma \neq 0$. In this section, we show that for any non-trivial values of $\gamma$, this is not the case. This directly implies that our algorithm uses optimal space for a large range of parameters.

**Theorem 18.** *Let $m, n, k \in \mathbb{N}$ and $\alpha > 0$. Then the space used by any randomized single-pass algorithm for $(\alpha, 5/6, O(m+n), k)$-LRA in the general turnstile model is at least $\Omega((m+n)k/\alpha)$.*

Theorem 18 shows that PRIVATE-SPACE-OPTIMAL-LRF uses optimal space when $\gamma = O(m+n)$ and $k \geq 1/\alpha$. If we set $\alpha = \sqrt{2} - 1$ (as in Hardt and Roth [47]) and note any non-trivial result implies that $\gamma = o(m+n)$, we have a matching lower bound for all $k \geq 3$.

The space lower bound in the turnstile update model is shown by showing that any algorithm Alg in the turnstile model yields a single round communication protocol for some function $f$. The idea is as follows. On input $\mathbf{x}$, Alice invokes Alg on its input to compute $\mathrm{Alg}(x)$. She then sends the state st to Bob, who computes $\mathrm{Alg}(\mathbf{x}\|\mathbf{y})$ using his input $\mathbf{y}$ and st, and uses this to compute the function $f$. The communication is therefore the same as the space required by the algorithm. In what follows, we use the notation $\mathbf{C}_{:i}$ to denote the $i$-th column of the matrix $\mathbf{C}$.

We give a reduction to the augmented indexing problem, AIND. It is defined as follows.

**Definition 8.** *(AIND problem). Alice is given an $N$-bit string $\boldsymbol{x}$ and Bob is given an index $\mathsf{ind} \in [N]$ together with $\boldsymbol{x}_{\mathsf{ind}+1}, \cdots, \boldsymbol{x}_N$. The goal of Bob is to output $\boldsymbol{x}_{\mathsf{ind}}$.*

The communication complexity for solving AIND is well known due to the result of Miltersen *et al.* [72].

**Theorem 19.** *The minimum bits of communication required to solve AIND with probability $2/3$, when the message is sent only in one direction, i.e., either from Alice to Bob or from Bob to Alice, is $\Omega(n)$. This lower bound holds even if the index, $\mathsf{ind}$, and the string, $\boldsymbol{x}$, is chosen uniformly at random.*

Before we state our result and its proof, we fix a notation. For a matrix $\mathbf{A}$ and set of indices $C$, we use the notation $\mathbf{A}(C)$ to denote the submatrix formed by the columns indexed by $C$.

*Proof of Theorem 18.* We adapt the proof of Clarkson and Woodruff [21] for the case when $\gamma \neq 0$. Suppose $m \geq n$ and let $a = k/20\alpha$. Without loss of generality, we can assume that $a$ is at most $n/2$. Let $\ell$ be the word size. We assume Alice has a string $\mathbf{x} \in \{-1, +1\}^{(m-a)a}$ and Bob has an index $\mathsf{ind} \in [(m-a)a]$. The idea is to define the matrix $\mathbf{A}$ with high Frobenius norm. The matrix $\mathbf{A}$ is the summation of the matrix $\widetilde{\mathbf{A}}$ constructed by Alice and $\bar{\mathbf{A}}$ constructed by Bob. We first define how Alice and Bob construct the instant $\mathbf{A} = \widetilde{\mathbf{A}} + \bar{\mathbf{A}}$.

Alice constructs its matrix $\widetilde{\mathbf{A}}$ as follows. Alice partitions the set $\{1, \cdots, a\}$ in to $\ell$ disjoint sets $I_1, \cdots, I_\ell$ such that $I_i := \{(i-1)a/\ell + 1, \cdots ia/\ell\}$. Let $\mathbf{M}(I_i)$ be an $(m-a) \times a/\ell$ matrix for all $1 \leq i \leq \ell$. We form a bijection between entries of $\mathbf{x}$ and the entries of $\mathbf{M}$ in the following manner. Every entry of $\mathbf{M}(I_i)$ is defined by a unique bit of $\mathbf{x}$, i.e., $\mathbf{M}(I_i)_{j,k} = (-1)^{\mathbf{x}_d}(10)^i$ for $d = (i-1)(m-a)a/\ell + (k-1)(m-a) + j$. The matrix $\widetilde{\mathbf{A}}$ is now defined as follows.

$$\widetilde{\mathbf{A}} = \begin{pmatrix} \mathbf{0}^{a \times a} & \mathbf{0}^{a \times (n-a)} \\ \mathbf{M} & \mathbf{0}^{(m-a) \times (n-a)} \end{pmatrix},$$

where $\mathbf{M} = (\mathbf{M}_{I_1} \quad \cdots \quad \mathbf{M}_{I_\ell})$.

Suppose Bob is given an index $\mathsf{ind} \in [(m-a)a]$ such that $\mathbf{x}_{\mathsf{ind}}$ corresponds to the sub-matrix $\mathbf{M}(I_\theta)$ for some $1 \leq \theta \leq \ell$. Then we can assume that Bob also knows every entry in the sub-matrix $\mathbf{M}(I_{\theta'})$ for $\theta' > \theta$. Bob forms a second level partition of the columns of $\mathbf{M}(I_\theta)$ in to equal size groups $G_1, \cdots, G_{a/k\ell}$. Due to our construction, there exists a unique $r$ such that $\mathbf{x}_{\mathsf{ind}}$ maps to an entry in the sub-matrix formed by columns indexed by one of the second level partition $G_r$. Let $C = \{c, c+1, \cdots, c+k-1\}$ be the columns corresponding to the $k$-size group of $I_\theta$ in which $\mathsf{ind}$ is present. As its input, Bob streams a matrix $\bar{\mathbf{A}}$ which is an all-zero matrix, except for entries $\bar{\mathbf{A}}_{c+i,c+i} = \zeta$ for $0 \leq i \leq k-1$ and $\zeta$ to be chosen later. In other words, Bob inserts a scaled identity matrix in the stream, where the scaling parameter $\zeta$ is large enough to make sure that most of the

error of any randomized algorithm is due to other columns of $\mathbf{A}$. As we shall see later, we set the value of $\zeta$ as a large polynomial in the approximation error of the algorithm.

Let $\mathcal{A}$ be the algorithm that computes LRA under the turnstile model. Alice feeds its matrix $\widetilde{\mathbf{A}}$ to $\mathcal{A}$ in the turnstile manner and send the state of the algorithm by the end of her feed to Bob. Bob uses the state received by Alice and feed the algorithm $\mathcal{A}$ with its own matrix $\bar{\mathbf{A}}$ in a turnstile manner. Therefore, the algorithm $\mathcal{A}$ gets as input a matrix $\mathbf{A} = \widetilde{\mathbf{A}} + \bar{\mathbf{A}}$ and it is required to output a rank-$k$ matrix $\mathbf{B}$ with additive error $\gamma = O(m + n)$. We will show that any such output allows us to solve AIND. Denote by $\mathbf{A}(C)$ the sub-matrix formed by the columns $C := \{c, c + 1, \cdots, c + k - 1\}$.

Let us first understand the properties of the constructed matrix $\mathbf{A}$. To compute the Frobenius norm of this matrix, we need to consider two cases: the case for sub-matrices in which ind belongs, i.e, $\mathbf{M}(I_r)$, and the rest of the matrix. For the sub-matrix corresponding to the columns indexed by $C$, the columns of $\mathbf{A}(I_\theta)$ have Euclidean length $(\zeta^2 + (m - a)100^\theta)^{1/2}$. For $\theta' < \theta$, every columns have Euclidean norm $(a(m - a))^{1/2}10^{\theta'}$. Therefore, we have the following:

$$\|\mathbf{A} - [\mathbf{A}]_k\|_F^2 \leq \frac{((a - k)(m - a)100^\theta}{\ell} + \sum_{\theta' < \theta} \frac{a(m - a)100^{\theta'}}{\ell}$$

$$\leq \frac{((a - k)(m - a)100^\theta}{\ell} + \frac{a(m - a)100^\theta}{99\ell} \leq 2 \cdot (100)^\theta m^2/\ell = \Gamma$$

In order to solve $(\alpha, \beta, \gamma, k)$-LRF, the algorithm needs to output a matrix $\mathbf{B}$ of rank at most $k$ such that, with probability $5/6$ over its random coins,

$$\|\mathbf{A} - \mathbf{B}\|_F^2 \leq \left[(1 + \alpha)\sqrt{\Gamma} + \gamma\right]^2 \leq 2(1 + \alpha)\Gamma + 2\gamma^2$$

$$\leq 2\Gamma + 100^\theta k(m - a)\left(\frac{1}{10} + \frac{1}{99}\right) + 2\gamma^2$$

$$\leq 4 \cdot (100)^\theta m^2/\ell + \frac{100^\theta k(m - a)}{5} + 2\gamma^2$$

Let us denote by $\Upsilon := 4 \cdot (100)^\theta m^2/\ell + 100^\theta k(m - a)\left(\frac{1}{10} + \frac{1}{99}\right) + 2\gamma^2$. The proof idea is now to show the following:

**(i)** Columns of $\mathbf{B}$ corresponding to index set in $C$ are linearly independent.

**(ii)** Bound the error incurred by $\|\mathbf{A} - \mathbf{B}\|_F$ in terms of the columns indexed by $G_r$.

The idea is to show that most of the error is due to the other columns in $\mathbf{B}$; and therefore, sign in the submatrix $\mathbf{A}(C)$ agrees with that of the signs of those in the submatrix $\mathbf{B}(C)$. This allows Bob to solve the AIND problem as Bob can just output the sign of the corresponding position.

Let

$$R := \{ra/k + 1, \cdots, (r + 1)a/k\}$$

and

$$C := \{c, \cdots, c + k - 1\}.$$

Let $\mathbf{Y}$ be the submatrix of $\mathbf{B}$ formed by the rows indexed by $R$ and columns indexed by $C$.

The following lemma proves that when $\zeta$ is large enough, then the columns of $\mathbf{B}$ corresponding to index set $C$ are linearly independent. This proves part (i) of our proof idea.

**Lemma 17.** *Let* $\mathbf{B}(C) := [\mathbf{B}_{:c} \quad \cdots \mathbf{B}_{:c+k-1}]$ *be the columns corresponding to the sub-matrix formed by columns* $c, \cdots, c + k - 1$ *of* $\mathbf{B}$. *If* $\zeta \geq 2\Upsilon^2$, *then the columns of* $\mathbf{B}(C)$ *spans the column space of* $[\mathbf{A}]_k$.

*Proof.* We will prove the lemma by considering the $k \times k$ sub-matrix, say $\mathbf{Y}$. Recall that $\mathbf{Y}$ is a submatrix of $\mathbf{B}$ formed by the rows indexed by $R$ and the columns indexed by $C$. For the sake of brevity and abuse of notation, let us denote the restriction of $\mathbf{B}$ to this sub-matrix $\mathbf{Y} := [\mathbf{Y}_{:1}, \cdots, \mathbf{Y}_{:k}]$. In what follows, we prove a stronger claim that the submatrix $\mathbf{Y}$ is a rank-$k$ matrix.

Suppose, for the sake of contradiction that the vectors $\{\mathbf{Y}_{:1}, \cdots, \mathbf{Y}_{:k}\}$ are linearly dependent. In other words, there exists a vector $\mathbf{Y}_{:i}$ and real numbers $a_1, \cdots, a_k$, not all of which are identically zero, such that

$$\mathbf{Y}_{:i} = \sum_{j=1, j \neq i}^{k} a_j \mathbf{Y}_{:j}.$$

From the construction, since Bob inserts a sub-matrix $\zeta \mathbb{I}_k$, we know that

$$\sum_{j=1}^{k} (\mathbf{Y}_{j,j} - \zeta)^2 \leq \|\mathbf{A} - \mathbf{B}\|_F^2 \leq \Upsilon. \tag{26}$$

$$\sum_{j=1}^{k} \sum_{p \neq j} \mathbf{Y}_{p,j}^2 \leq \|\mathbf{A} - \mathbf{B}\|_F^2 \leq \Upsilon. \tag{27}$$

From equation (26) and choice of $\zeta$, for all $j$, we have $\mathbf{Y}_{j,j} \geq \Upsilon^2$. Further, equation (27) implies that $\mathbf{Y}_{p,j} \leq \sqrt{\Upsilon}$. We have

$$\mathbf{Y}_{i,i} = \sum_{j=1, j \neq i}^{k} a_j \mathbf{Y}_{i,j} \geq \Upsilon^2$$

imply that there is an $p \in \{1, \cdots, k\} \setminus \{i\}$ such that $|a_p| \geq \frac{\Upsilon^2}{k\sqrt{\Upsilon}}$.

Let $\tilde{i}$ be the index in $\{1, \cdots, k\} \setminus \{i\}$ for which $|a_{\tilde{i}}|$ attains the maximum value. We have $|a_{\tilde{i}} \mathbf{Y}_{\tilde{i},\tilde{i}}| \geq |a_{\tilde{i}}| \Upsilon^2$ and $|a_j \mathbf{Y}_{\tilde{i},j}| \leq |a_{\tilde{i}}| \sqrt{\Upsilon}$. Now consider the $\tilde{i}$-entry of $\mathbf{Y}_{:i}$. Note that $\tilde{i} \neq i$. Since $\Upsilon$ depends quadratically on $m$ and $\gamma$, we have

$$\left| \sum_{j=1, j \neq i}^{k} a_j \mathbf{Y}_{\tilde{i},j} \right| \geq |a|(\Upsilon^2 - k\sqrt{\Upsilon}) \geq (\Upsilon^2 - k\sqrt{\Upsilon}) \frac{\Upsilon^2}{k\sqrt{\Upsilon}} > \sqrt{\Upsilon}.$$

This is a contradiction because $\mathbf{Y}_{p,j} \leq \sqrt{\Upsilon}$ due to equation (27) for $p \neq j$. This completes the proof. $\square$

For the sake of brevity, let $\mathbf{V}_{:1}, \cdots, \mathbf{V}_{:k}$ be the columns of $\mathbf{B}(C)$ and $\widetilde{\mathbf{V}}_{:1}, \cdots, \widetilde{\mathbf{V}}_{:k}$ be the restriction of these column vectors to the rows $a + 1, \cdots, m$. In other words, vectors $\widetilde{\mathbf{V}}_{:1}, \cdots, \widetilde{\mathbf{V}}_{:k}$ are the column vectors corresponding to the columns in $\mathbf{M}$. We showed in Lemma 17 that the columns $\mathbf{B}(C)$ spans the column space of $\mathbf{B}$. We can assume that the last $n - a$ columns of $\mathbf{B}$ are all zero vectors because $\mathbf{B}$ is a rank-$k$ matrix. We can also assume without any loss of generality that, except for the entries in the row indexed by $R$, all the other entries of $\mathbf{B}(C)$ are zero. This is because we have shown in Lemma 17, we showed that the submatrix of $\mathbf{B}(C)$ formed by rows indexed by $R$ and columns indexed by $C$ have rank $k$.

Now any row $i$ of $\mathbf{B}$ can be therefore represented as $\sum \eta_{i,j} \mathbf{V}_{:j}$, for real numbers $\eta_{i,j}$, not all of which are identically zero. The following lemma proves part (ii) of our proof idea. For

**Lemma 18.** *Let $\mathbf{V}_{:1}, \cdots, \mathbf{V}_{:k}$ be as defined above. Then column $i$ of $\mathbf{B}$ can be written as linear combination of real numbers $\eta_{i,1}, \cdots \eta_{i,k}$ of the vectors $\mathbf{V}_{:1}, \cdots, \mathbf{V}_{:k}$ such that, for all $j$ and $i \in R$, $\eta_{i,j}^2 \leq 4/\Upsilon^3$.*

*Proof.* Let $\mathbf{M}_{:1}, \cdots \mathbf{M}_{:a}$ be the columns of $\mathbf{M}$, where $\mathbf{M}$ is the $(m - a) \times a$ submatrix of the matrix $\widetilde{\mathbf{A}}$ corresponding to the input of Alice. We have

$$\Upsilon \geq \|\mathbf{A} - \mathbf{B}\|_F^2 \sum_{i=1}^{k} (\zeta - \mathbf{V}_{r(a/k)+i,i})^2 + \sum_{i=1}^{k} \sum_{j \neq i} \mathbf{V}_{r(a/k)+i,j}^2 + \sum_{i=1}^{k} \|\mathbf{M}_{:r(a/k)+i} - \widetilde{\mathbf{V}}_{:i}\|^2$$

$$+ \sum_{i \notin R} \sum_{j=1}^{k} \left( \eta_{i,j} \mathbf{V}_{ra/k+j,j} + \sum_{j' \neq j} \eta_{i,j'} \mathbf{V}_{ra/k+j,j'} \right)^2 + \sum_{i \notin R} \left\| \mathbf{M}_{:i} - \sum_{j=1}^{k} \eta_{i,j} \widetilde{\mathbf{V}}_{:j} \right\|^2.$$

As in the proof of Lemma 17, we have $|\mathbf{V}^2_{r(a/k)+i,j}| \leq \sqrt{\Upsilon}$ and $|\mathbf{V}_{r(a/k)+i,i}| \geq \Upsilon^2$. Let $j_i$ be the index such that $|\eta_{i,j_i}|$ is the maximum. Then the above expression is at least $|\eta_{i,j_i}|^2(\Upsilon^2 - k\sqrt{\Upsilon})^2 \geq |\eta_{i,j_i}|^2\Upsilon^4/4$. Since this is less than $\Upsilon$, the result follows from the definition of $j_i$. $\qquad\square$

We can now complete the proof. First note that since $\mathbf{M}$ is a signed matrix, each $\widetilde{\mathbf{V}}_i$ in the third term of the above expression is at least $\sqrt{\Upsilon}$. Therefore, for all $i \notin S$ and all $j$

$$\left| \sum_{j=1}^{k} \eta_{i,j} \widetilde{\mathbf{V}}_{:j} \right| \leq \frac{4k\Upsilon^{1/2}}{\Upsilon^{3/2}} = \frac{4k}{\Upsilon}.$$

As $\mathbf{M}_{:i}$ is a sign vector and if $\gamma = O(m+n) = O(m)$, this implies that

$$\sum_{i \notin R} \left\| \mathbf{M}_{:i} - \sum_{j=1}^{k} \eta_{i,j} \widetilde{\mathbf{V}}_{:j} \right\|^2 \geq \sum_{i \notin R} \|\mathbf{M}_{:i}\|^2 \left(1 - \frac{4k}{\Upsilon}\right) \geq O((100)^\theta m^2/\ell) - O(100^\theta a)$$

$$\sum_{i=1}^{k} \left\| \mathbf{M}_{:r(a/k)+i} - \widetilde{\mathbf{V}}_{:i} \right\|^2 = \sum_{i=1}^{k} \sum_{j=1}^{m-a} (\mathbf{M}_{j,r(a/k)+i} - (\widetilde{\mathbf{V}}_i)_j)^2 \leq \frac{100^\theta k(m-a)}{5} + O(100^\theta a)$$

Now, since there are in total $k(m-a)$ entries in the submatrix formed by the columns indexed by $C$, at least $1 - \left(\frac{1}{10} + \frac{1}{99} + o(1)\right)$ fraction of the entries have the property that the sign of $\mathbf{M}_{j,ra/k+i}$ matches the sign of $\widetilde{\mathbf{V}}_{j,i}$. Since ind is in one of the columns of $\mathbf{M}_{:ra/k+1}, \cdots \mathbf{M}_{:ra/k+k}$, with probability at least $1 - \left(\frac{1}{10} + \frac{1}{99} + o(1)\right)$, if Bob outputs the sign of the corresponding entry in $\mathbf{B}$, then Bob succeeds in solving AIND. This gives a lower bound of $\Omega((m-a)a) = \Omega(mk\ell/\alpha)$ space. The case when $m \leq n$ is analogous and gives a lower bound of $\Omega(nk\ell/\alpha)$. Thus, there is a lower bound of $\Omega((m+n)k\ell/\alpha)$. $\qquad\square$

## J  Noninteractive Local Differentially Private PCA

Till now, we have considered a single server that receives the private matrix in a streamed manner. We next consider a stronger variant of differential privacy known as *local differential privacy* (LDP) [34, 36, 42, 97]. In the *local* model, each individual applies a differentially private algorithm locally to their data and shares only the output of the algorithm—called a report—with a server that aggregates users' reports. A multi-player protocol is $\varepsilon$-LDP if for all possible inputs and runs of the protocol, the transcript of player $i$'s interactions with the server is $\varepsilon$-LDP.

One can study two variants of local differential privacy depending on whether the server and the users interact more than once or not (see Figure 8). In the interactive variant, the server sends several messages, each to a subset of users. In the noninteractive variant, the server sends a single message to all the users at the start of the protocol and sends no message after that. Smith, Thakurta, and Upadhyay [89] argued that noninteractive locally private algorithms are ideal for implementation.

Figure 8: LDP with (left) and without (right) interaction.

The natural extension of Problem 1 in the local model is when the matrix is distributed among the users such that every user has one row of the matrix and users are responsible for the privacy of their row vector. Unfortunately, known private algorithms (including the results presented till now) do not yield non trivial additive error in the local model. For example, if we convert Theorem 1 to

the local model, we end up with an additive error $\widetilde{O}(\sqrt{kmn})$. This is worse than the trivial bound of $O(\sqrt{mn})$, for example, when $\mathbf{A} \in \{0,1\}^{m \times n}$, a trivial output of all zero matrix incurs an error at most $O(\sqrt{mn})$. In fact, existing lower bounds in the local model suggests that one is likely to incur an error which is $O(\sqrt{m})$ factor worse than in the central model, where $m$ is the number of users. However, owing to the result of Dwork *et al.* [40], we can hope to achieve non-trivial result for differentially private principal component analysis (see, Definition 4) leading us to ask

**Question 1.** *Is there a locally private algorithm for low rank principal component analysis?*

This problem has been studied without privacy under the *row-partition model* [8, 64, 61, 17, 44, 83, 84, 91]). Even though there is a rich literature on local differentially private algorithms [5, 11, 34, 41, 42, 49, 58, 73, 97], the known approaches to convert existing (private or distributed non-private) algorithms to locally private algorithms either leads to a large additive error or require interaction. We exploit the fact that our meta algorithm only stores differentially private sketches of the input matrix to give a noninteractive algorithm for low-rank principal component analysis (PCA) under local differential privacy.

Tight upper and lower bounds known on the achievable accuracy for many problems; however, low-rank factorization (and even low-rank approximation) has not been studied in this model. The naive approach to convert existing algorithms to locally private algorithms leads to a large additive error and are interactive. On the other hand, low-rank factorization is a special optimization problem and the role of interaction in local differentially private optimization was recently investigated by Smith *et al.* [89].

**Theorem 20.** *Let $m, n \in \mathbb{N}$ and $\alpha, \varepsilon, \delta$ be the input parameters. Let $k$ be the desired rank of the factorization and $\eta = \max\left\{k, \alpha^{-1}\right\}$. Let $t = O(\eta\alpha^{-1}\log(k/\delta))$ and $v = O(\eta\alpha^{-2}\log(k/\delta))$. Given a private input matrix $\mathbf{A} \in \mathbb{R}^{m \times n}$ distributed in a row-wise manner amongst $m$ users, the output $\mathbf{U}$ of the algorithm, $\textsc{Private-Local-LRF}$, presented in Figure 9 is a $k$-rank orthonormal matrix such that*

1. $\textsc{Private-Local-LRF}$ *is a non-interactive $(\varepsilon, \delta)$-local differentially private under $\mathsf{Priv}_2$.*

2. *With probability $9/10$ over the coins of $\textsc{Private-Local-LRF}$,*

$$\|\mathbf{A} - \mathbf{U}\mathbf{U}^{\mathsf{T}}\mathbf{A}\|_F \leq (1 + O(\alpha))\|\mathbf{A} - [\mathbf{A}]_k\|_F + O\left(v\sqrt{m\log(1/\delta)}/\epsilon\right).$$

3. *The words of communication used by every users in $\textsc{Private-Local-LRF}$ is $O(v^2)$ words.*

Our algorithm only produces an $(\epsilon, \delta)$-locally differentially private algorithm; however, it is non-interactive. This allows us to use the generic transformation of Bun *et al.* [18] to get the following result. The above theorem gives the first instance of non-interactive algorithm that computes low-rank principal component analysis in the model of local differential privacy. The best known lower bound on additive error for $(\epsilon, \delta)$-differentially private PCA is by Dwork *et al.* [40] in the central model for static data matrix. Their lower bound on the additive error is $\widetilde{\Omega}(k\sqrt{n})$ for squared Frobenius norm and has no multiplicative error. An interesting question from our result is to investigate how close or far we are from optimal error.

The proof for local private algorithm is little different from previous proofs. Therefore, we first give a proof sketch without privacy. The proof becomes more involved with privacy.

$$\begin{aligned}
\mathbf{P}_1 &:= \operatorname{argmin} \|\mathbf{\Psi}(\mathbf{A}_k\mathbf{X} - \mathbf{A})\|_F, & \mathbf{P}_2 &: \operatorname{argmin} \|(\mathbf{A}_k\mathbf{X} - \mathbf{A})\|_F \\
\mathbf{P}_3 &: \operatorname{argmin} \|(\mathbf{X}\mathbf{P}_1 - \mathbf{A})\mathbf{\Phi}\|_F, & \mathbf{P}_4 &:= \operatorname{argmin} \|\mathbf{X}\mathbf{P}_1 - \mathbf{A}\|_F
\end{aligned}$$

Using the normal form of Frobenius regression problem, $\mathbf{P}_1 = (\mathbf{\Psi}\mathbf{A}_k)^\dagger \mathbf{\Psi}\mathbf{A}$ and $\mathbf{P}_3 = \mathbf{A}\mathbf{\Phi}(\mathbf{P}_1\mathbf{\Phi})^\dagger$. Then we can show that

$$\|\mathbf{A}\mathbf{\Phi}(\mathbf{P}_1\mathbf{\Phi})^\dagger\mathbf{P}_1 - \mathbf{A}\|_F \leq (1 + \alpha)\min_{\mathbf{X}} \|\mathbf{X}\mathbf{P}_1 - \mathbf{A}\|_F \leq (1 + \alpha)^2\|\mathbf{A}_k - \mathbf{A}\|_F$$

Since $(\mathbf{P}_1\mathbf{\Phi})^\dagger(\mathbf{\Psi}\widehat{\mathbf{A}}_k)^\dagger$ is a rank-k matrix, we have

$$\min_{\mathsf{r}(\mathbf{X}) \leq k} \|\mathbf{A}\mathbf{\Phi}\mathbf{X}\mathbf{\Psi}\mathbf{A} - \mathbf{A}\|_F \leq \|\mathbf{A}\mathbf{\Phi}(\mathbf{P}_1\mathbf{\Phi})^\dagger\mathbf{P}_1 - \mathbf{A}\|_F$$

$$\boxed{\begin{aligned}
&\text{\textsc{Private-Local-LRF}}\\
&\textbf{Initialization.} \text{ Let } \eta = \max\{k, \alpha^{-1}\}, t = O(\eta\alpha^{-1}\log(k/\delta)), v = O(\eta\alpha^{-2}\log(k/\delta)). \text{ Let } \rho_1 = \\
&\sqrt{(1+\alpha)\ln(1/\delta)}/\varepsilon, \; \rho_2 = (1+\alpha)\sqrt{\ln(1/\delta)}/\varepsilon. \text{ Sample } \mathbf{\Phi} \sim \mathcal{N}(0,1)^{n \times t}, \mathbf{\Psi} \sim \mathcal{N}(0,1)^{t \times m}, \\
&\mathbf{S} \sim \mathcal{N}(0,1)^{v \times m}, \text{ and } \mathbf{T} \sim \mathcal{N}(0,1)^{n \times v}. \text{ Make them public.}
\end{aligned}}$$

**Initialization.** Let $\eta = \max\{k, \alpha^{-1}\}$, $t = O(\eta\alpha^{-1}\log(k/\delta))$, $v = O(\eta\alpha^{-2}\log(k/\delta))$. Let $\rho_1 = \sqrt{(1+\alpha)\ln(1/\delta)}/\varepsilon$, $\rho_2 = (1+\alpha)\sqrt{\ln(1/\delta)}/\varepsilon$. Sample $\mathbf{\Phi} \sim \mathcal{N}(0,1)^{n \times t}$, $\mathbf{\Psi} \sim \mathcal{N}(0,1)^{t \times m}$, $\mathbf{S} \sim \mathcal{N}(0,1)^{v \times m}$, and $\mathbf{T} \sim \mathcal{N}(0,1)^{n \times v}$. Make them public.

**User-$i$ computation.** On input the row $\mathbf{A}_{i:}$, user-$i$ does the following:

1. Sample $\mathbf{N}_{1,i} \sim \mathcal{N}(0,\rho_1^2)^{1 \times t}$, $\mathbf{N}_{2,i} \sim \mathcal{N}(0,\rho_2^2)^{t \times v}$ and $\mathbf{N}_{3,i} \sim \mathcal{N}(0,\rho_2^2)^{v \times v}$.

2. Set $\widehat{\mathbf{A}}_{i:} \in \mathbb{R}^{m \times n}$ such that every row other than row-$i$ is an all zero vector. Compute $\mathbf{Y}_{i:} = \mathbf{A}_{i:}\mathbf{\Phi} + \mathbf{N}_{1,i}$, $\widetilde{\mathbf{Y}}_{i:} = \mathbf{\Psi}\widehat{\mathbf{A}}_{i:}\mathbf{T} + \mathbf{N}_{2,i}$, and $\mathbf{Z}_{i:} = \mathbf{S}\widehat{\mathbf{A}}_{i:}\mathbf{T} + \mathbf{N}_{3,i}$.

**Server side computation.** Once the server receives the reports from all the users, it follows the following steps.

1. Form $\mathbf{Y}$ whose row-$i$ is $\mathbf{Y}_{i:}$. Compute $\mathbf{Z} = \sum \mathbf{Z}_{i:}$ and $\widetilde{\mathbf{Y}} = \sum \widetilde{\mathbf{Y}}_{i:}$. Compute $\widehat{\mathbf{Y}} = \mathbf{S}\mathbf{Y}$.

2. Compute $\widetilde{\mathbf{X}} := \operatorname{argmin}_{\mathsf{rk}(\mathbf{X}) \leq k} \|\widehat{\mathbf{Y}}\mathbf{X}\widetilde{\mathbf{Y}} - \mathbf{Z}\|_F$. Compute a SVD of $\widetilde{\mathbf{X}}$. Let it is be $\mathbf{U}'\mathbf{\Sigma}'\mathbf{V}'^{\mathsf{T}}$.

3. Output the orthonormal basis $\mathbf{U}$ for the span of $\mathbf{Y}\mathbf{U}'$.

Figure 9: Non-interactive Local Differentially private LRF Under Priv$_1$

Now we are done because of the following. Suppose $\mathbf{U}$ is the orthonormal basis in the span of $\mathbf{A}\mathbf{\Phi}\widetilde{\mathbf{X}}$, where

$$\widetilde{\mathbf{X}} := \operatorname*{argmin}_{\mathsf{rank}(X) \leq k} \|\mathbf{S}\mathbf{A}\mathbf{\Phi}\mathbf{X}\mathbf{\Psi}\mathbf{A}\mathbf{T} - \mathbf{S}\mathbf{A}\mathbf{T}\|_F,$$

then by picking $\mathbf{P} = \mathbf{\Psi}\mathbf{A}$,

$$\begin{aligned}
\|\mathbf{U}\mathbf{U}^T\mathbf{A} - \mathbf{A}\|_F = \|\mathbf{A}\mathbf{\Phi}\widetilde{\mathbf{X}}(\mathbf{A}\mathbf{\Phi}\widetilde{\mathbf{X}})^\dagger\mathbf{A} - \mathbf{A}\|_F &\leq \|\mathbf{A}\mathbf{\Phi}\widetilde{\mathbf{X}}\mathbf{P} - \mathbf{A}\|_F \\
&\leq (1+\alpha)\|\mathbf{S}(\mathbf{A}\mathbf{\Phi}\widetilde{\mathbf{X}}\mathbf{\Psi}\mathbf{A} - \mathbf{A})\mathbf{T}\|_F = \min_{r(X)\leq k}\|\mathbf{S}(\mathbf{A}\mathbf{\Phi}\mathbf{X}\mathbf{\Psi}\mathbf{A} - \mathbf{A})\mathbf{T}\|_F \\
&\leq (1+\alpha)\min_{r(\mathbf{X})\leq k}\|\mathbf{A}\mathbf{\Phi}\mathbf{X}\mathbf{\Psi}\mathbf{A} - \mathbf{A}\|_F,
\end{aligned}$$

where the last inequality follows as in the case of Appendix E.1.

*Proof.* The local privacy is easy to follow from the Gaussian mechanism and as in Lemma 11 with the choice of $\rho_1$ and $\rho_2$. For the communication cost, note that every user $i$ has to send a sketch $\mathbf{Y}_{i:}$, $\widehat{\mathbf{Y}}_{i:}$, and $\mathbf{Z}_{i:}$. The sketch $\mathbf{Y}_{i:}$ is a real $1 \times t$ matrix, $\widehat{\mathbf{Y}}_{i:}$ is an $t \times v$ real matrix, and $\mathbf{Z}_{i:}$ is a $v \times v$ real matrix. The total communication cost is $O((tv + v^2)\log(nm))$ words. Since $t \leq v$, the result on the communication cost follows.

We now prove Part 2 of Theorem 20. Let $\mathbf{N}_1$ be a random Gaussian matrices whose row-$i$ is $\mathbf{N}_{1,i}$. Let $\mathbf{N}_2 = \sum \mathbf{N}_{2,i}$, $\mathbf{N}_3 = \sum \mathbf{N}_{3,i}$. Note that $\mathbf{N}_1 \sim \mathcal{N}(0,\rho_1^2)^{m \times t}$, $\mathbf{N}_2 \sim \mathcal{N}(0,m\rho_2^2)^{t \times v}$, and $\mathbf{N}_2 \sim \mathcal{N}(0,m\rho_2^2)^{v \times v}$. Let $\mathbf{Y}$ be the matrix whose row-$i$ is $\mathbf{Y}_{i:}$. Further, $\mathbf{Z} = \sum \mathbf{Z}_{i:}$ and $\widetilde{\mathbf{Y}} = \sum \widetilde{\mathbf{Y}}_{i:}$. If the matrix distributed among the users is $\mathbf{A}$, then it means that $\mathbf{Y} = \mathbf{\Phi}\mathbf{A} + \mathbf{N}_1$, $\widetilde{\mathbf{Y}} = \mathbf{\Psi}\mathbf{A}\mathbf{T} + \mathbf{N}_3$ and $\mathbf{Z} = \mathbf{S}\mathbf{A}\mathbf{T} + \mathbf{N}_2$.

Let the singular value decomposition of $[\mathbf{A}]_k$ be $[\mathbf{A}]_k = \mathbf{U}_k\mathbf{\Sigma}_k\mathbf{V}_k^{\mathsf{T}}$. Let $\mathbf{C} = \mathbf{\Psi}(\mathbf{A} + \mathbf{\Psi}^\dagger\mathbf{N}_2\mathbf{T}^\dagger)$. We will use Lemma 1 to relate $\min_{\mathsf{rk}(\mathbf{X})\leq k} \|\mathbf{Y}\mathbf{X}\mathbf{C} - (\mathbf{A} + \mathbf{S}^\dagger\mathbf{N}_3\mathbf{T}^\dagger)\|_F$ with $\|\mathbf{A} - [\mathbf{A}]_k\|_F$. Set $\mathbf{\Phi} = \mathbf{\Psi}, \mathbf{P} = [\mathbf{A}]_k, \mathbf{Q} = \mathbf{A} + \mathbf{\Psi}^\dagger\mathbf{N}_2\mathbf{T}^\dagger$ in Lemma 1. For

$$\widetilde{\mathbf{X}} := (\mathbf{\Psi}[\mathbf{A}]_k)^\dagger(\mathbf{\Psi}\mathbf{A} + \mathbf{N}_2\mathbf{T}^\dagger) = (\mathbf{\Psi}[\mathbf{A}]_k)^\dagger\mathbf{C} = \operatorname*{argmin}_{\mathbf{X}} \|\mathbf{\Psi}([\mathbf{A}]_k\mathbf{X} - \mathbf{A} + (\mathbf{\Psi}^\dagger\mathbf{N}_2\mathbf{T}^\dagger))\|,$$

we have with probability $1 - \delta$ over $\mathbf{\Psi} \sim \mathcal{D}_R$,

$$\begin{aligned}
\|[\mathbf{A}]_k\widetilde{\mathbf{X}} - (\mathbf{A} + \mathbf{\Psi}^\dagger\mathbf{N}_2\mathbf{T}^\dagger)\|_F &\leq (1+\alpha)\min_{\mathbf{X}}\|[\mathbf{A}]_k\mathbf{X} - (\mathbf{A} + \mathbf{\Psi}^\dagger\mathbf{N}_2\mathbf{T}^\dagger)\|_F \\
&\leq (1+\alpha)\|[\mathbf{A}]_k - \mathbf{A}\|_F + (1+\alpha)\|\mathbf{\Psi}^\dagger\mathbf{N}_2\mathbf{T}^\dagger\|_F. \quad (28)
\end{aligned}$$

In the above, the second inequality follows by setting $\mathbf{X} = \mathbf{V}_k \mathbf{V}_k^\mathsf{T}$.

Let $\mathbf{W}^\mathsf{T} := \widetilde{\mathbf{X}} = (\mathbf{\Psi}[\mathbf{A}]_k)^\dagger \mathbf{C}$. We now use Lemma 1 on the following regression problem:

$$\min_{\mathbf{X}} \|\mathbf{\Phi}^\mathsf{T}(\mathbf{WX} - \mathbf{B})\| \quad \text{and} \quad \min_{\mathbf{X}} \|\mathbf{WX} - \mathbf{B}\|_F, \quad \text{where } \mathbf{B} = (\mathbf{A} + \mathbf{N}_1 \mathbf{\Phi}^\dagger)^\mathsf{T}$$

with the candidate solutions

$$\widehat{\mathbf{X}} = \operatorname*{argmin}_{\mathbf{X}} \|\mathbf{\Phi}^\mathsf{T}(\mathbf{WX} - \mathbf{A})\|_F \quad \text{and} \quad \widetilde{\mathbf{X}} = \operatorname*{argmin}_{\mathbf{X}} \|(\mathbf{WX} - \mathbf{A})\|_F$$

One of the candidate solutions to $\operatorname{argmin}_{\mathbf{X}} \|\mathbf{\Phi}^\mathsf{T}(\mathbf{WX} - \mathbf{A})\|_F$ is $\widehat{\mathbf{X}} := (\mathbf{\Phi}^\mathsf{T} \mathbf{W})^\dagger (\mathbf{\Phi}^\mathsf{T} \mathbf{B})$. Since $[\mathbf{A}]_k$ has rank $k$, Lemma 1 and equation (28) gives with probability $1 - \delta$ over $\mathbf{\Psi} \sim \mathcal{D}_R$

$$\|\widehat{\mathbf{X}}^\mathsf{T} \mathbf{W}^\mathsf{T} - \mathbf{B}^\mathsf{T}\|_F \leq (1+\alpha) \min_{\mathbf{X}} \|\mathbf{X}^\mathsf{T} \mathbf{W}^\mathsf{T} - \mathbf{B}^\mathsf{T}\|_F \leq (1+\alpha)\|[\mathbf{A}]_k (\mathbf{\Psi}[\mathbf{A}]_k)^\dagger \mathbf{C} - \mathbf{B}^\mathsf{T}\|_F$$

$$\leq (1+\alpha)\|[\mathbf{A}]_k (\mathbf{\Psi}[\mathbf{A}]_k)^\dagger \mathbf{C} - \mathbf{A}\|_F + (1+\alpha)\|\mathbf{N}_1 \mathbf{\Phi}^\dagger\|_F$$

$$\leq (1+\alpha)^2 \|\mathbf{A} - [\mathbf{A}]_k\|_F + (1+\alpha)(2+\alpha)\|\mathbf{\Psi}^\dagger \mathbf{N}_2 \mathbf{T}^\dagger\|_F + (1+\alpha)\|\mathbf{N}_1 \mathbf{\Phi}^\dagger\|_F$$

This in particular implies that

$$\|\widehat{\mathbf{X}}^\mathsf{T} \mathbf{W}^\mathsf{T} - \mathbf{A}\|_F \leq (1+\alpha)^2 \|\mathbf{A} - [\mathbf{A}]_k\|_F + (1+\alpha)(2+\alpha)\|\mathbf{\Psi}^\dagger \mathbf{N}_2 \mathbf{T}^\dagger\|_F + (2+\alpha)\|\mathbf{N}_1 \mathbf{\Phi}^\dagger\|_F.$$

Let

$$\tau_1 = (1+\alpha)(2+\alpha)\|\mathbf{\Psi}^\dagger \mathbf{N}_2 \mathbf{T}^\dagger\|_F + (2+\alpha)\|\mathbf{N}_1 \mathbf{\Phi}^\dagger\|_F$$

be the additive error due to the effect of noise $\mathbf{N}_1$ and $\mathbf{N}_2$.

Substituting the value of $\widehat{\mathbf{X}}^\mathsf{T} := (\mathbf{B}^\mathsf{T} \mathbf{\Phi})(\mathbf{W}^\mathsf{T} \mathbf{\Phi})^\dagger = (\mathbf{A}\mathbf{\Phi} + \mathbf{N}_1)(\mathbf{W}^\mathsf{T} \mathbf{\Phi})^\dagger = \mathbf{Y}(\mathbf{W}^\mathsf{T} \mathbf{\Phi})^\dagger$, with probability $1 - 2\delta$ over $\mathbf{\Phi}^\mathsf{T}, \mathbf{\Psi} \sim \mathcal{D}_R$, we have

$$\|\mathbf{Y}(\mathbf{W}^\mathsf{T} \mathbf{\Phi})^\dagger (\mathbf{\Psi}[\mathbf{A}]_k)^\dagger \mathbf{C} - \mathbf{A}\|_F \leq (1+\alpha)^2 \|\mathbf{A} - [\mathbf{A}]_k\|_F + \tau_1.$$

Let $\mathbf{X}_* := (\mathbf{W}^\mathsf{T} \mathbf{\Phi})^\dagger (\mathbf{\Psi}[\mathbf{A}]_k)^\dagger$, i.e.,

$$\|\mathbf{Y}\mathbf{X}_* \mathbf{C} - \mathbf{A}\|_F \leq (1+\alpha)^2 \|\mathbf{A} - [\mathbf{A}]_k\|_F + \tau_1.$$

Let $\mathbf{E} = \mathbf{A} + \mathbf{S}^\dagger \mathbf{N}_3 \mathbf{T}^\dagger$. Since $\mathbf{X}_*$ has rank at most $k$, this implies that

$$\min_{\substack{\mathbf{X} \\ \mathrm{rk}(\mathbf{X}) \leq k}} \|(\mathbf{Y}\mathbf{X}\mathbf{C} - \mathbf{E})\|_F \leq \|(\mathbf{Y}\mathbf{X}_* \mathbf{C} - \mathbf{E})\|_F$$

$$\leq \|(\mathbf{Y}\mathbf{X}_* \mathbf{C} - \mathbf{A})\|_F + \|\mathbf{S}^\dagger \mathbf{N}_3 \mathbf{T}^\dagger\|_F$$

$$\leq (1+\alpha)^2 \|\mathbf{A} - [\mathbf{A}]_k\|_F + \tau_1 + \|\mathbf{S}^\dagger \mathbf{N}_3 \mathbf{T}^\dagger\|_F.$$

Since $\alpha \in (0,1)$ and substituting the value of $\tau_1$, we can get an upper bound on the additive terms.

$$\min_{\substack{\mathbf{X} \\ \mathrm{rk}(\mathbf{X}) \leq k}} \|(\mathbf{Y}\mathbf{X}\mathbf{C} - \mathbf{E})\|_F \leq (1+\alpha)^2 \|\mathbf{A} - [\mathbf{A}]_k\|_F + O\left(\tau_1 + \|\mathbf{S}^\dagger \mathbf{N}_3 \mathbf{T}^\dagger\|_F\right). \tag{29}$$

Now consider the following two regression problems:

$$\min_{\substack{\mathbf{X} \\ \mathrm{rk}(\mathbf{X}) \leq k}} \|\mathbf{S}\mathbf{Y}\mathbf{X}\mathbf{C}\mathbf{T} - \mathbf{S}\mathbf{E}\mathbf{T}\|_F \quad \text{and} \quad \min_{\substack{\mathbf{X} \\ \mathrm{rk}(\mathbf{X}) \leq k}} \|\mathbf{Y}\mathbf{X}\mathbf{C} - \mathbf{E}\|_F \tag{30}$$

with candidate solutions $\widetilde{\mathbf{X}} := \operatorname*{argmin}_{\mathrm{rk}(\mathbf{X}) \leq k} \|\mathbf{S}\mathbf{Y}\mathbf{X}\mathbf{C}\mathbf{T} - \mathbf{S}\mathbf{E}\mathbf{T}\|_F$ and $\widehat{\mathbf{X}} := \operatorname*{argmin}_{\mathrm{rk}(\mathbf{X}) \leq k} \|\mathbf{Y}\mathbf{X}\mathbf{C} - \mathbf{E}\|_F$, respectively. Set $p = k/\alpha$, $\mathbf{D} = \mathbf{Y}\mathbf{X}\mathbf{C}$ and $\mathbf{E} = \mathbf{A} + \mathbf{S}^\dagger \mathbf{N}_3 \mathbf{T}^\dagger$ in the statement of Lemma 2. Then we have with probability $1 - 2\delta$ over $\mathbf{S}.\mathbf{T} \sim \mathcal{D}_A$,

$$\min_{\substack{\mathbf{X} \\ r(\mathbf{X}) = k}} \|\mathbf{Y}\mathbf{X}\mathbf{C} - \mathbf{E}\|_F^2 = \|\mathbf{Y}\widehat{\mathbf{X}}\mathbf{C} - \mathbf{E}\|_F^2 = \|\mathbf{Y}\widehat{\mathbf{X}}\mathbf{C} - (\mathbf{A} + \mathbf{S}^\dagger \mathbf{N}_3 \mathbf{T}^\dagger)\|_F^2$$

$$\geq (1+\alpha)\|\mathbf{S}(\mathbf{Y}\widehat{\mathbf{X}}\mathbf{C} - (\mathbf{A} + \mathbf{S}^\dagger \mathbf{N}_3 \mathbf{T}^\dagger))\mathbf{T}\|_F^2$$

$$\geq (1+\alpha) \min_{\substack{\mathbf{X} \\ r(\mathbf{X})}} \|\mathbf{S}\mathbf{Y}\mathbf{X}\mathbf{C}\mathbf{T} - \mathbf{S}(\mathbf{A} + \mathbf{S}^\dagger \mathbf{N}_3 \mathbf{T}^\dagger))\mathbf{T}\|_F$$

$$= (1+\alpha)\|\widehat{\mathbf{Y}}\widetilde{\mathbf{X}}\widetilde{\mathbf{Y}} - \mathbf{Z}\|_F. \tag{31}$$

The second and last equality follows from the definition, the first inequality follows from Lemma 2 and the second inequality follows from the fact that minimum is smaller than any other choice of $\mathbf{X}$, more specifically $\mathbf{X} = \widehat{\mathbf{X}}$. Since $\mathbf{U}$ is in the span of $\mathbf{YU}'$, where $\mathbf{U}'$ is the left singular vectors of $\widetilde{\mathbf{X}}$, we have the following:

$$
\begin{aligned}
\|\mathbf{UU}^\mathsf{T}\mathbf{E} - \mathbf{E}\|_F &= \|(\mathbf{YU})(\mathbf{YU})^\dagger \mathbf{E} - \mathbf{E}\|_F \\
&= \min_{\mathbf{X}} \|(\mathbf{YU}')\mathbf{X} - \mathbf{E}\|_F \leq \|\mathbf{Y}\widetilde{\mathbf{X}}\mathbf{\Psi}\mathbf{Q} - \mathbf{E}\|_F \\
&\leq (1+\alpha)\|\mathbf{SY}\widetilde{\mathbf{X}}\mathbf{\Psi}\mathbf{QT} - \mathbf{SET}\|_F = (1+\alpha)\|\widehat{\mathbf{Y}}\widetilde{\mathbf{X}}\widetilde{\mathbf{Y}} - \mathbf{Z}\|_F .
\end{aligned}
$$

The second equality comes from the normal form of Frobenius norm. Combining equation (31) and equation (29), and subadditivity of norm after putting in the value of $\mathbf{E}$, this implies that

$$
\|(\mathbb{I} - \mathbf{UU}^\mathsf{T})\mathbf{A}\|_F \leq (1 + O(\alpha))\|\mathbf{A} - [\mathbf{A}]_k\|_F + O\left(\|\mathbf{\Psi}^\dagger \mathbf{N}_2 \mathbf{T}^\dagger\|_F + \|\mathbf{N}_1 \mathbf{\Phi}^\dagger\|_F + \|\mathbf{S}^\dagger \mathbf{N}_3 \mathbf{T}^\dagger\|_F\right).
$$

As in the proof of Theorem 3, for the choice of $v$ and $t$, for every matrices $\mathbf{D}$, $\|\mathbf{SD}\|_F \leq (1+\alpha)\|\mathbf{D}\|_F$, $\|\mathbf{DT}\|_F \leq (1+\alpha)\|\mathbf{D}\|_F$, $\|\mathbf{\Psi D}\|_F \leq (1+\alpha)\|\mathbf{D}\|_F$, and $\|\mathbf{D\Phi}\|_F \leq (1+\alpha)\|\mathbf{D}\|_F$. In other words, $\|\mathbf{\Psi}^\dagger \mathbf{N}_2 \mathbf{T}^\dagger\|_F \leq (1+\alpha)\|\mathbf{N}_2\|_F$, $\|\mathbf{N}_1 \mathbf{\Phi}^\dagger\|_F \leq \sqrt{1+\alpha}\|\mathbf{N}_1\|_F$, $\|\mathbf{S}^\dagger \mathbf{N}_3 \mathbf{T}^\dagger\|_F \leq (1+\alpha)\|\mathbf{N}_3\|_F$. Since $\alpha \in (0,1)$,

$$
\|(\mathbb{I} - \mathbf{UU}^\mathsf{T})\mathbf{A}\|_F \leq (1 + O(\alpha))\|\mathbf{A} - [\mathbf{A}]_k\|_F + O\left(\|\mathbf{N}_2\|_F + \|\mathbf{N}_1\|_F + \|\mathbf{N}_3\|_F\right).
$$

The result follows using Lemma 4, Markov's inequality, and values of $\rho_1$ and $\rho_2$. $\qquad\square$

## K    Empirical Evaluation of Our Algorithms

Any algorithm to compute the low-rank factorization In this section, we give the experimental evaluation of our algorithms and compare it with the best known results. We ran our algorithms on a 2.7 GHz Intel Core i5 processor with 16 GB 1867 MHz DDR3 RAM. Our algorithms keep on sampling a random matrix randomly until we sample a matrix with number of columns more than 200.

### K.1    Empirical Evaluation of PRIVATE-OPTIMAL-SPACE-LRF

We first start with the discussion on the empirical evaluation of PRIVATE-OPTIMAL-SPACE-LRF (see Figure 1 for the detail description and the supplementary materials for the source code). Since the error incurred by PRIVATE-FROBENIUS-LRF is strictly less than that by PRIVATE-OPTIMAL-SPACE-LRF, we only concern ourselves with PRIVATE-OPTIMAL-SPACE-LRF. For the private setting, we sampled matrices from the following distributions:

1. All the entries are sampled uniformly from the interval $[1, 5000]$
2. All the entries are integers sampled uniformly from the interval $[1, 5000]$.

In our experimental set-up, we keep the value of $\alpha = 0.25$ and $k = 10$ fixed to get a better understanding of how the approximation changes with the changing values of dimensions.

We start by explaining what every columns in Table 3 means. The first two columns are the dimension of the private matrix, the third column is the desired rank of the output matrix, and the fourth column is the value of multiplicative approximation. For the ease of comparison, we have set $k$ and $\alpha$ to be a constant parameter in this experiment and let the dimension to be the free parameters.

Recall that the problem of the low-rank factorization is to output a singular value decomposition $\widetilde{\mathbf{U}}, \widetilde{\mathbf{\Sigma}}, \widetilde{\mathbf{V}}$ such that $\mathbf{M}_k = \widetilde{\mathbf{U}}\widetilde{\mathbf{\Sigma}}\widetilde{\mathbf{V}}^\mathsf{T}$ is a rank-$k$ matrix and

$$
\|\mathbf{A} - \mathbf{M}_k\|_F \leq (1+\alpha)\|\mathbf{A} - [\mathbf{A}]_k\|_F + \gamma,
$$

where $\gamma$ is the additive error. The fifth and the sixth columns enumerate the value of the expression resulting from running our algorithm PRIVATE-OPTIMAL-SPACE-LRF and that by Hardt and Roth [47], respectively. The last column represents the optimal low-rank factorization, $\|\mathbf{A} - [\mathbf{A}]_k\|_F$.

There is no way to compute the actual additive error, $\gamma$, empirically. This is because there is a factor of multiplicative error and it is tough to argue what part of error is due to the multiplicative factor

alone. In other words, the best we can compute is $\alpha\Delta_k + \gamma$ or the total approximation error incurred by our algorithm. From the practitioner point of view, the total error is a much useful parameter than the $\alpha\Delta_k + \gamma$. Therefore, in this section, we use the total approximation error ($\|\mathbf{A} - \mathbf{M}_k\|_F$) as the measure for our evaluation.

| Distribution of $\mathbf{A}$ | Rows | Columns | $k$ | $\alpha$ | Our error $\|\mathbf{A} - \mathbf{M}_k\|_F$ | Hardt-Roth [47] $\|\mathbf{A} - \mathbf{M}_k\|_F$ | Optimal Error $\|\mathbf{A} - [\mathbf{A}]_k\|_F$ |
|---|---|---|---|---|---|---|---|
| | 535 | 50 | 10 | 0.25 | 223649.755822 | 552969.553361 | 190493.286508 |
| | 581 | 57 | 10 | 0.25 | 254568.54093 | 491061.894752 | 213747.405532 |
| | 671 | 65 | 10 | 0.25 | 295444.274372 | 470153.533646 | 250629.568178 |
| | 705 | 70 | 10 | 0.25 | 317280.295345 | 546149.007321 | 269647.886009 |
| | 709 | 68 | 10 | 0.25 | 309627.397618 | 664748.40864 | 265799.14431 |
| | 764 | 74 | 10 | 0.25 | 344154.666385 | 529618.155224 | 291053.598305 |
| | 777 | 50 | 10 | 0.25 | 270458.465497 | 436864.395454 | 235057.184632 |
| Uniform real | 861 | 57 | 10 | 0.25 | 311968.552859 | 494331.526734 | 269761.822539 |
| | 1020 | 65 | 10 | 0.25 | 367175.274642 | 562322.74973 | 317998.616149 |
| | 1054 | 70 | 10 | 0.25 | 389357.219211 | 490379.45171 | 338605.316751 |
| | 1061 | 68 | 10 | 0.25 | 386772.176623 | 497648.401337 | 334581.574424 |
| | 1137 | 74 | 10 | 0.25 | 413134.221292 | 528692.808214 | 364187.907915 |
| | 1606 | 158 | 10 | 0.25 | 736233.063187 | 848827.366953 | 654600.528481 |
| | 1733 | 169 | 10 | 0.25 | 786963.961154 | 932695.219591 | 706550.246496 |
| | 522 | 50 | 10 | 0.25 | 217497.498819 | 496080.416815 | 185817.742179 |
| | 555 | 51 | 10 | 0.25 | 229555.549295 | 463022.451669 | 195569.953293 |
| | 605 | 60 | 10 | 0.25 | 267625.256679 | 525350.686285 | 225614.671569 |
| | 714 | 70 | 10 | 0.25 | 316232.378407 | 477066.707503 | 270968.006565 |
| | 804 | 51 | 10 | 0.25 | 284102.975661 | 548426.535153 | 241720.509615 |
| | 899 | 86 | 10 | 0.25 | 402886.168791 | 554702.285328 | 346840.731082 |
| | 906 | 60 | 10 | 0.25 | 328747.816311 | 455091.762984 | 284433.77154 |
| | 913 | 90 | 10 | 0.25 | 412114.948358 | 634520.151202 | 358345.162361 |
| Uniform integers | 1061 | 106 | 10 | 0.25 | 486139.117249 | 618819.626784 | 423775.149619 |
| | 1063 | 70 | 10 | 0.25 | 395772.128472 | 485655.074685 | 339950.706212 |
| | 1305 | 86 | 10 | 0.25 | 488729.886028 | 551863.893152 | 427234.256941 |
| | 1383 | 90 | 10 | 0.25 | 513573.18853 | 595195.801858 | 451019.165808 |
| | 1486 | 145 | 10 | 0.25 | 677118.945777 | 776008.62945 | 600584.597101 |
| | 1481 | 146 | 10 | 0.25 | 670290.341074 | 733574.295922 | 600877.636254 |
| | 1635 | 106 | 10 | 0.25 | 616323.217861 | 652624.510827 | 541305.826364 |
| | 1848 | 180 | 10 | 0.25 | 836139.102987 | 884143.446663 | 755160.753156 |
| | 1983 | 194 | 10 | 0.25 | 896926.450848 | 1005652.63777 | 814717.343468 |

Table 3: Empirical Comparison Between PRIVATE-OPTIMAL-SPACE-LRF and Hardt and Roth [47].

The empirical evaluations, listed in Table 3, reflect that our algorithm perform consistently better than that of Hardt and Roth [47] for all the dimension range. This agrees with our analysis and our discussion in the main text. In particular, we showed that theoretically we perform better than Hardt and Roth [47] by a factor of $O(c\sqrt{k})$, where $c$ is the largest entry in their projection matrix.

Another observation that one can make from our evaluation is that the error of our algorithm is quiet close to the actual error of approximation for all the dimension range. On the other hand, the error of Hardt and Roth [47] is close to optimal error in the large dimensional matrices. For small dimensional matrices, the error incurred by Hardt and Roth [47] is lot more than the actual error. The error of Hardt and Roth [47] starts getting better as the dimension increases. We believe that the fact that our algorithm performs well over all the range of dimensions makes it more stable with respect to different datasets. In practice, this is highly desirable as one would like the algorithm to perform well on both large and small datasets.

The last key observation one can gather from the empirical evaluation is that though the total error depends on the Frobenius norm, the additive error is independent of the Frobenius norm of the original matrix. It is expected that the total error depends on the Frobenius norm because of the multiplicative factor, but if we see the difference between the errors (of both our and Hardt and Roth's algorithm) and the optimal error, the difference scales proportional to the dimensions of the matrices.

| Rows | Columns | $k$ | $\alpha$ | Our additive error | Hardt-Roth Additive Error | Expected Additive Error |
|------|---------|-----|----------|-------------------|---------------------------|-------------------------|
| 546 | 50 | 10 | 0.25 | 665.797531323 | 13971.0499468 | 818.4149601308452 |
| 780 | 50 | 10 | 0.25 | 777.586619111 | 16772.4716145 | 915.2974642186384 |
| 532 | 51 | 10 | 0.25 | 719.492368601 | 23512.4449181 | 817.9937262895351 |
| 808 | 51 | 10 | 0.25 | 796.220653146 | 14613.575971 | 932.3276903711178 |
| 655 | 62 | 10 | 0.25 | 845.550304391 | 34161.5584705 | 941.6210056415899 |
| 951 | 62 | 10 | 0.25 | 903.21367849 | 14101.7225434 | 1055.244933017033 |
| 891 | 89 | 10 | 0.25 | 1040.90463257 | 17863.8728746 | 1190.7199415503355 |
| 1344 | 89 | 10 | 0.25 | 1273.05275389 | 18646.717977 | 1342.904603982863 |
| 522 | 50 | 10 | 0.25 | 691.996294265 | 23193.0915951 | 806.8851715645378 |
| 791 | 50 | 10 | 0.25 | 764.535817382 | 16245.8487264 | 919.3095213779045 |
| 1449 | 140 | 10 | 0.25 | 1392.08606822 | 20835.1618122 | 1639.1144500265145 |
| 2143 | 140 | 10 | 0.25 | 1518.90720786 | 13521.4077062 | 1827.1906485042728 |
| 834 | 80 | 10 | 0.25 | 969.883327308 | 22051.2853027 | 1119.4759720856953 |
| 1234 | 80 | 10 | 0.25 | 1005.64736332 | 13617.4286089 | 1257.2282266407094 |
| 682 | 64 | 10 | 0.25 | 833.555604169 | 19875.8713339 | 965.2425226826088 |
| 967 | 64 | 10 | 0.25 | 872.993347497 | 15774.1091244 | 1073.4637306190796 |
| 924 | 90 | 10 | 0.25 | 1024.67412984 | 20018.1421648 | 1208.8365389072335 |
| 1374 | 90 | 10 | 0.25 | 1168.33857697 | 15267.6596425 | 1357.3931639793143 |
| 1981 | 194 | 10 | 0.25 | 2845.12484193 | 19457.1056713 | 2035.7863227397222 |
| 2945 | 194 | 10 | 0.25 | 1938.83169063 | 14210.6828353 | 2263.796573983761 |
| 1022 | 100 | 10 | 0.25 | 1130.29734323 | 14839.3883841 | 1298.0001587502882 |
| 1530 | 100 | 10 | 0.25 | 1289.31236852 | 14886.7349415 | 1458.2654318931454 |
| 1867 | 182 | 10 | 0.25 | 1806.04962492 | 13443.8218792 | 1952.5511080639412 |
| 2757 | 182 | 10 | 0.25 | 1983.37270829 | 13509.2925192 | 2168.928742648721 |

Table 4: Empirical Comparison of Additive Error of PRIVATE-OPTIMAL-SPACE-LRF and Hardt and Roth [47].

### K.1.1 Empirical Evaluation of Additive Error for various dimension

As we mentioned earlier, if the matrix has rank greater than $k$, it is not possible to empirically evaluate the additive error. However, we believe it is still imperative to analyze the effect of differential privacy on the low-rank approximation of matrices. This in turn implies that one should also define experiments to empirically evaluate the additive error. One easy way to do this is to take as input a matrix with rank exactly $k$ and compare the error incurred with that of expected error promised by our theoretical results. In the next experiment we do the same and prune the last $n - k$ columns of the matrix and make it identically zero (see Figure 1 for the detail description). The result of our experiment is presented in Table 4. Since every entries of the matrix is identically zero, we notice that the same trend as in Table 3:

1. The additive error incurred by our algorithm is way less than the additive error incurred by Hardt and Roth [47] for all ranges of the dimension. We note that the matrices are highly incoherent as all the entries are sampled i.i.d. We believe the reason for this behavior is the fact that the theoretical result provided by Hardt and Roth [47] for incoherent matrices depended on the Frobenius norm of the input matrix.

2. Our algorithm consistently perform better than the additive error guaranteed by the theoretical results, but the difference becomes smaller as the dimension increases. This trend can be seen as due to the fact that our results are asymptotic and we believe as $m$ and $n$ are sufficiently large, our theoretical result would match the empirical results.

### K.1.2 Empirical Evaluation of Additive Error for various values of $\alpha$.

An important parameter that comes in our bounds and is absent in the bounds of Hardt and Roth [47] is the factor of $\alpha$. This is because Hardt and Roth [47] consider a constant $\alpha$. Therefore, we feel it is important to analyze the additive error with respect to the change in $\alpha$ in order to better understand the effect of differential privacy on the low-rank approximation of matrices. Again, we take as input a matrix with rank exactly $k$ and compare the error incurred with that of expected error promised by our theoretical results while keeping the dimensions and the value of $k$ constant (see Figure 1 for the detail description). The result of our experiment is presented in Table 5. Since every entries of the matrix is identically zero, we notice that the same trend as in Table 3:

1. The additive error incurred by our algorithm is way less than the additive error incurred by Hardt and Roth [47] for all ranges of the dimension. We note that the matrices are highly incoherent as all the entries are sampled i.i.d. We believe the reason for this behavior is the fact that the theoretical result provided by Hardt and Roth [47] for incoherent matrices depended on the Frobenius norm of the input matrix.

2. Our algorithm consistently perform better than the additive error guaranteed by the theoretical results, except for certain values of the dimensions and multiplicative error ($m = 2800, n = 200, \alpha = 0.24$). Even in these cases, the error is not that far from what is predicted from the theoretical analysis.

| Rows | Columns | $k$ | $\alpha$ | Expected Additive Error | Our additive error | Hardt-Roth Additive Error |
|------|---------|-----|----------|-------------------------|--------------------|---------------------------|
| 1800 | 200 | 10 | 0.1  | 7889.477972559828  | 8056.91337611 | 18155.7964938 |
| 1800 | 200 | 10 | 0.12 | 7114.371042156239  | 6986.62461896 | 16748.9933963 |
| 1800 | 200 | 10 | 0.14 | 6234.5851506132285 | 6171.60624904 | 26904.2257957 |
| 1800 | 200 | 10 | 0.16 | 6234.5851506132285 | 5982.88510433 | 16983.3495414 |
| 1800 | 200 | 10 | 0.18 | 5190.996544903864  | 4956.753081   | 15746.884528  |
| 1800 | 200 | 10 | 0.2  | 5190.996544903864  | 5044.67253402 | 14124.190425  |
| 1800 | 200 | 10 | 0.22 | 5190.996544903864  | 5012.95041479 | 18030.3508985 |
| 1800 | 200 | 10 | 0.24 | 5190.996544903864  | 4951.97119364 | 18013.7573095 |
| 2800 | 200 | 10 | 0.1  | 8690.661799154163  | 8550.34968943 | 12281.3954039 |
| 2800 | 200 | 10 | 0.12 | 7846.020358487806  | 7607.18833498 | 14296.1174909 |
| 2800 | 200 | 10 | 0.14 | 6887.309251977769  | 6494.49329799 | 11674.4990144 |
| 2800 | 200 | 10 | 0.16 | 6887.309251977769  | 6603.16942717 | 13860.6899516 |
| 2800 | 200 | 10 | 0.18 | 5750.100760072804  | 5417.53433303 | 13425.7590356 |
| 2800 | 200 | 10 | 0.2  | 5750.100760072804  | 5612.34884207 | 12731.6645942 |
| 2800 | 200 | 10 | 0.22 | 5750.100760072804  | 5524.92292528 | 10703.6065701 |
| 2800 | 200 | 10 | 0.24 | 5750.100760072804  | 6450.77223767 | 12610.5718019 |

Table 5: Empirical Comparison of Additive Error of PRIVATE-OPTIMAL-SPACE-LRF for various values of $\alpha$ with Hardt and Roth [47].

### K.1.3 Empirical Evaluation of Additive Error for various values of $k$.

The last parameter that comes in our bounds and in the bounds of Hardt and Roth [47] is the factor of $k$. Therefore, we feel it is important to analyze the additive error with respect to the change in $k$ in order to better understand the effect of differential privacy on the low-rank approximation of matrices. Again, we take as input a matrix with rank exactly $k$ and compare the error incurred with that of expected error promised by our theoretical results while keeping the dimensions and the value of $k$ constant (see Figure 1 for the detail description and). The result of our experiment is presented in Table 6. Since every entries of the matrix is identically zero, we notice that the same trend as in Table 3:

1. The additive error incurred by our algorithm is way less than the additive error incurred by Hardt and Roth [47] for all ranges of the dimension. We note that the matrices are highly incoherent as all the entries are sampled i.i.d. We believe the reason for this behavior is the fact that the theoretical result provided by Hardt and Roth [47] for incoherent matrices depended on the Frobenius norm of the input matrix.

2. Our algorithm is almost the same as the additive error guaranteed by the theoretical results.

### K.2 Empirical Evaluation of PRIVATE-LOCAL-LRF

In this section, we understand the result of our empirical evaluation of PRIVATE-LOCAL-LRF (see Figure 1 for the detail description). Recall that in this case, we output a rank-$k$ orthonormal matrix $\mathbf{U}$ such that $\mathbf{UU}^\mathsf{T}\mathbf{A}$ well approximates the matrix $\mathbf{A}$ with high probabiliy, i.e.,

$$\|\mathbf{A} - \mathbf{UU}^\mathsf{T}\mathbf{A}\|_F \leq (1+\alpha)\|\mathbf{A} - [\mathbf{A}]_k\|_F + \gamma$$

with probability at least $1 - \beta$.

| Rows | Columns | $k$ | $\alpha$ | Expected Additive Error | Our additive error | Hardt-Roth Additive Error |
|------|---------|-----|----------|-------------------------|--------------------|---------------------------|
| 450 | 50 | 10 | 0.25 | 1955.5571620375354 | 1898.46860098 | 20016.9761114 |
| 450 | 50 | 11 | 0.25 | 1967.6211924065349 | 2067.40361185 | 24063.2388836 |
| 450 | 50 | 12 | 0.25 | 1979.148228191017 | 1915.44710901 | 23731.1221157 |
| 450 | 50 | 13 | 0.25 | 1990.2041707005583 | 1975.07412336 | 28148.8596261 |
| 450 | 50 | 14 | 0.25 | 2000.8424524636212 | 1937.32846282 | 44746.5335036 |
| 450 | 50 | 15 | 0.25 | 2395.058604918562 | 2656.30764859 | 38466.5764635 |
| 450 | 50 | 16 | 0.25 | 2404.9864394488554 | 2396.43074838 | 51068.6496986 |
| 450 | 50 | 17 | 0.25 | 2414.6085814624234 | 2518.70267919 | 59951.048053 |
| 450 | 50 | 18 | 0.25 | 2423.9516422006986 | 2412.15253004 | 49652.4140161 |
| 450 | 50 | 19 | 0.25 | 2433.0385823622364 | 2421.29320351 | 60475.2111995 |
| 700 | 50 | 10 | 0.25 | 2210.505167206932 | 2137.57732175 | 18322.4177099 |
| 700 | 50 | 11 | 0.25 | 2225.805492035416 | 2244.60952111 | 15915.3100262 |
| 700 | 50 | 12 | 0.25 | 2240.424768215734 | 2341.16599056 | 18904.9300369 |
| 700 | 50 | 13 | 0.25 | 2254.4465757300545 | 2356.95623935 | 23088.1856217 |
| 700 | 50 | 14 | 0.25 | 2267.9386809066186 | 2251.25863623 | 29899.7509467 |
| 700 | 50 | 15 | 0.25 | 2707.304866721297 | 2948.78705396 | 33654.4497548 |
| 700 | 50 | 16 | 0.25 | 2719.895940227483 | 2906.80323902 | 36885.4983615 |
| 700 | 50 | 17 | 0.25 | 2732.0993162013783 | 2677.54649011 | 36275.4890195 |
| 700 | 50 | 18 | 0.25 | 2743.9487446109097 | 2849.04550002 | 37581.6388795 |
| 700 | 50 | 19 | 0.25 | 2755.473345587197 | 2658.52670606 | 47314.5277845 |

Table 6: Empirical Comparison of Additive Error of PRIVATE-OPTIMAL-SPACE-LRF and Hardt and Roth [47] for various values of $k$.

| Rows | Columns | $k$ | $\alpha$ | Our error ($\|\mathbf{A} - \mathbf{U}\mathbf{U}^{\mathsf{T}}\mathbf{A}\|_F$) | Optimal Error ($\|\mathbf{A} - [\mathbf{A}]_k\|_F$) |
|------|---------|-----|----------|------------------------------------------------------------------------------|-----------------------------------------------------|
| 460 | 50 | 10 | 0.25 | 26730.7062683 | 18376.5128345 |
| 486 | 50 | 10 | 0.25 | 28080.8322915 | 18964.3784188 |
| 516 | 55 | 10 | 0.25 | 30526.641347 | 20870.0427927 |
| 553 | 56 | 10 | 0.25 | 29787.7727748 | 21862.0631087 |
| 568 | 59 | 10 | 0.25 | 31632.1354094 | 22760.6941581 |
| 616 | 64 | 10 | 0.25 | 33981.2524409 | 25159.7572035 |
| 709 | 50 | 10 | 0.25 | 36449.213243 | 23110.1192303 |
| 730 | 50 | 10 | 0.25 | 35232.7419048 | 23414.630713 |
| 742 | 50 | 10 | 0.25 | 36167.200779 | 23613.9643873 |
| 805 | 56 | 10 | 0.25 | 38175.3014108 | 26613.9920423 |
| 817 | 56 | 10 | 0.25 | 38604.715515 | 26721.6604408 |
| 846 | 59 | 10 | 0.25 | 39702.5550961 | 28179.965914 |
| 907 | 64 | 10 | 0.25 | 41889.0912424 | 30717.3035814 |
| 924 | 101 | 10 | 0.25 | 50376.7374653 | 40511.7157875 |
| 1195 | 130 | 10 | 0.25 | 66049.3867783 | 53122.2489582 |
| 1262 | 138 | 10 | 0.25 | 70596.6678822 | 56526.8303644 |
| 1433 | 101 | 10 | 0.25 | 63976.8587336 | 50810.8395315 |
| 1698 | 186 | 10 | 0.25 | 91159.7299285 | 77358.5067276 |
| 1857 | 130 | 10 | 0.25 | 85812.7647106 | 66588.7353796 |
| 1956 | 138 | 10 | 0.25 | 82720.7186764 | 70710.4258919 |

Table 7: Empirical Evaluation of PRIVATE-LOCAL-LRF.

For empirical evaluation of our locally-private algorithm, we sampled matrices such that every entries are uniform real number between $[0, 500]$. In our experimental set-up, we keep the value of $\alpha = 0.25$ and $k = 10$ fixed to get a better understanding of how the approximation changes with the changing values of dimensions. Our empirical results are listed in Table 7. From the table, we can immediately see the effect of the role of the dimension $m$ and can see the difference between the optimal approximation and our approximation scale faster than in the case of PRIVATE-OPTIMAL-SPACE-LRF. However, even in the case of very large matrices, our approximation error is very small compared to the actual approximation error. In other words, this evaluation gives us a hint that our algorithm does not pay a lot for local model of computation. We make this more explicit in our next set of experiments given in the following sections when we empirically evaluate the additive error to better understand the effect of local differential privacy on the accuracy of the algorithm.

| Rows | Columns | $k$ | $\alpha$ | Our Additive Error | Expected Additive Error |
|------|---------|-----|----------|--------------------|--------------------------|
| 454  | 50  | 10 | 0.25 | 1591.70826988 | 2236.0029137367846 |
| 494  | 53  | 10 | 0.25 | 1659.12206578 | 2348.466431926056 |
| 511  | 52  | 10 | 0.25 | 1737.01336788 | 2395.0391957271527 |
| 562  | 60  | 10 | 0.25 | 2000.5063552  | 2530.799729601783 |
| 622  | 66  | 10 | 0.25 | 1580.25894963 | 2683.7137939487484 |
| 643  | 70  | 10 | 0.25 | 1962.46122303 | 2735.6746706859394 |
| 645  | 70  | 10 | 0.25 | 2172.80349046 | 2740.583809665359 |
| 702  | 50  | 10 | 0.25 | 1989.93430285 | 2877.768275821911 |
| 728  | 52  | 10 | 0.25 | 2059.26581077 | 2938.6953727048567 |
| 743  | 53  | 10 | 0.25 | 1993.78092528 | 2973.406277353555 |
| 844  | 60  | 10 | 0.25 | 2328.32881246 | 3199.4689043324324 |
| 853  | 94  | 10 | 0.25 | 2557.29517064 | 3219.0131110666084 |
| 925  | 66  | 10 | 0.25 | 2484.75888129 | 3372.181278187486 |
| 981  | 70  | 10 | 0.25 | 2198.25471982 | 3487.6698882062387 |
| 983  | 70  | 10 | 0.25 | 2597.33562269 | 3491.7393627228817 |
| 1055 | 113 | 10 | 0.25 | 3217.07445615 | 3635.863197534171 |
| 1190 | 131 | 10 | 0.25 | 4220.63858322 | 3894.742583462755 |
| 1320 | 94  | 10 | 0.25 | 2808.85540377 | 4131.88571345083 |
| 1583 | 113 | 10 | 0.25 | 3870.14181092 | 4581.675389464531 |
| 1713 | 186 | 10 | 0.25 | 4445.32063486 | 4791.555177594492 |
| 1835 | 131 | 10 | 0.25 | 5447.86868861 | 4982.106342591863 |
| 1900 | 206 | 10 | 0.25 | 4732.78855445 | 5081.305279633226 |
| 2213 | 158 | 10 | 0.25 | 4145.11357428 | 5539.0037396773205 |
| 2888 | 206 | 10 | 0.25 | 6234.47976474 | 6436.032325032722 |

Table 8: Empirical Evaluation of Additive Error of PRIVATE-LOCAL-LRF for varying values of dimensions.

### K.2.1 Empirical Evaluation of Additive Error for Various Dimension

As we argued earlier, it is still imperative to analyze the effect of differential privacy on the principal component analysis of matrices. This in turn implies that one should also define experiments to empirically evaluate the additive error. As earlier, we take as input a matrix with rank exactly $k$ and compare the error incurred with that of expected error promised by our theoretical results. In the next experiment we do the same and prune the last $n - k$ columns of the matrix and make it identically zero (see Figure 1 for the detail description). The result of our experiment is presented in Table 8. Recall that the additive error incurred by our algorithm is

$$\gamma := O\left(\eta\alpha^{-2}\log(k/\delta)\sqrt{m\log(1/\delta)}/\epsilon\right)$$

for $\eta = \max\{k, 1/\alpha\}$.

Since every entries of the matrix $\mathbf{A} - [\mathbf{A}]_k$ is identically zero, the error listed in the Table 8 is the additive error. We note that the trend of Table 8 shows the same trend as in Table 7.

Our algorithm consistently perform better than the additive error guaranteed by the theoretical results (except for $m = 1835$ and $n = 131$), but the difference becomes smaller as the dimension increases. This trend can be seen as due to the fact that our results are asymptotic and we believe when $m$ is sufficiently large, our theoretical result would match the empirical results.

### K.2.2 Empirical Evaluation of Additive Error for various values of $\alpha$.

An important parameter that comes in our bounds is the factor of $\alpha$. Our theoretical result shows a tradeoff between the additive error and the multiplicative approximation factor. This makes it important to analyze the additive error with respect to the change in $\alpha$ in order to better understand the effect of differential privacy on the low-rank approximation of matrices. Again, we take as input a matrix with rank exactly $k$ and compare the error incurred with that of expected error promised by our theoretical results while keeping the dimensions and the value of $k$ constant. The result of our experiment is presented in Table 9.

Recall that the additive error incurred by our algorithm is

$$\gamma := O\left(\eta\alpha^{-2}\log(k/\delta)\sqrt{m\log(1/\delta)}/\epsilon\right)$$

for $\eta = \max\{k, 1/\alpha\}$.

Since every entries of the matrix $\mathbf{A} - [\mathbf{A}]_k$ is identically zero, the error listed is the additive error incurred by our algorithm. We notice that the same trend as in Table 7.

We ran our algorithm for $m = \{450, 700\}$ and $n = 50$ with $k = 10$ and $\alpha$ ranging from 0.10 to 0.24 in the step-size of 0.02. Our algorithm consistently perform better than the additive error guaranteed by the theoretical results. The empirical error is way better than the error predicted by the theoretical result for small values of $\alpha$ and it starts getting closer as the values of $\alpha$ increases.

| Rows | Columns | $k$ | $\alpha$ | Our Additive Error | Expected Additive Error |
|---|---|---|---|---|---|
| 450 | 50 | 10 | 0.10 | 750.710811389 | 14088.628533821675 |
| 450 | 50 | 10 | 0.12 | 748.91024846 | 9639.587944193776 |
| 450 | 50 | 10 | 0.14 | 764.598714683 | 7415.067649379828 |
| 450 | 50 | 10 | 0.16 | 727.388618355 | 5190.54735456588 |
| 450 | 50 | 10 | 0.18 | 748.286674358 | 4449.040589627896 |
| 450 | 50 | 10 | 0.20 | 722.269126382 | 3707.533824689914 |
| 450 | 50 | 10 | 0.22 | 695.354504806 | 2966.027059751931 |
| 450 | 50 | 10 | 0.24 | 687.993333996 | 2224.520294813948 |
| 700 | 50 | 10 | 0.1 | 932.294964215 | 18195.922623848826 |
| 700 | 50 | 10 | 0.12 | 937.676945315 | 12449.841795264987 |
| 700 | 50 | 10 | 0.14 | 939.872486613 | 9576.801380973067 |
| 700 | 50 | 10 | 0.16 | 942.039074746 | 6703.760966681148 |
| 700 | 50 | 10 | 0.18 | 909.247538784 | 5746.08082858384 |
| 700 | 50 | 10 | 0.20 | 948.147219877 | 4788.400690486534 |
| 700 | 50 | 10 | 0.22 | 860.725294579 | 3830.7205523892267 |
| 700 | 50 | 10 | 0.24 | 882.673634462 | 2873.04041429192 |

Table 9: Empirical Evaluation of Additive Error of PRIVATE-LOCAL-LRF for various values of $\alpha$.

| Rows | Columns | $k$ | $\alpha$ | Our Additive Error | Expected Additive Error |
|---|---|---|---|---|---|
| 450 | 50 | 10 | 0.2 | 739.297184071 | 2966.027059751931 |
| 450 | 50 | 11 | 0.2 | 792.139775851 | 3707.533824689914 |
| 450 | 50 | 12 | 0.2 | 827.193502967 | 3707.533824689914 |
| 450 | 50 | 13 | 0.2 | 857.238846842 | 4449.040589627896 |
| 450 | 50 | 14 | 0.2 | 881.890196768 | 4449.040589627896 |
| 450 | 50 | 15 | 0.2 | 877.407208023 | 5190.54735456588 |
| 450 | 50 | 16 | 0.2 | 955.688935848 | 5190.54735456588 |
| 450 | 50 | 17 | 0.2 | 942.773082147 | 5932.054119503862 |
| 450 | 50 | 18 | 0.2 | 1019.96432587 | 5932.054119503862 |
| 450 | 50 | 19 | 0.2 | 1033.82124639 | 6673.560884441846 |
| 700 | 50 | 10 | 0.2 | 921.415775041 | 3830.7205523892267 |
| 700 | 50 | 11 | 0.2 | 903.693226831 | 4788.400690486534 |
| 700 | 50 | 12 | 0.2 | 961.550364155 | 4788.400690486534 |
| 700 | 50 | 13 | 0.2 | 1055.58902486 | 5746.08082858384 |
| 700 | 50 | 14 | 0.2 | 1102.54543656 | 5746.08082858384 |
| 700 | 50 | 15 | 0.2 | 1139.854348 | 6703.760966681148 |
| 700 | 50 | 16 | 0.2 | 1188.983938 | 6703.760966681148 |
| 700 | 50 | 17 | 0.2 | 1216.1836631 | 7661.441104778453 |
| 700 | 50 | 18 | 0.2 | 1207.76296999 | 7661.441104778453 |
| 700 | 50 | 19 | 0.2 | 1303.58983727 | 8619.121242875759 |

Table 10: Empirical Evaluation of Additive Error of PRIVATE-LOCAL-LRF for various values of $k$.

### K.2.3 Empirical Evaluation of Additive Error for various values of $k$.

The last parameter that comes in our bounds for local differentially private algorithm is the factor of $k$. Therefore, we feel it is important to analyze the additive error with respect to the change in $k$ in order to better understand the effect of differential privacy on the low-rank approximation of matrices. Again, we take as input a matrix with rank exactly $k$ for varying values of $k$ and compare

| Rows | Columns | $k$ | $\alpha$ | Total error ($\|\mathbf{A} - \mathbf{M}_k\|_F$) | Optimal Error ($\|\mathbf{A} - [\mathbf{A}]_k\|_F$) |
|---|---|---|---|---|---|
| 498 | 52 | 10 | 0.25 | 203636.487171 | 197577.81058 |
| 565 | 62 | 10 | 0.25 | 241322.216245 | 234585.873351 |
| 600 | 66 | 10 | 0.25 | 258293.615653 | 251581.697045 |
| 634 | 68 | 10 | 0.25 | 272545.864945 | 265331.39875 |
| 701 | 50 | 10 | 0.25 | 236960.63674 | 230358.636959 |
| 719 | 76 | 10 | 0.25 | 311278.453125 | 302812.926139 |
| 736 | 50 | 10 | 0.25 | 243505.180195 | 236083.969219 |
| 775 | 52 | 10 | 0.25 | 255720.933052 | 248391.690752 |
| 780 | 86 | 10 | 0.25 | 348405.00368 | 340148.071775 |
| 818 | 90 | 10 | 0.25 | 364630.812101 | 355410.694701 |
| 888 | 62 | 10 | 0.25 | 306043.651247 | 297740.00906 |
| 954 | 66 | 10 | 0.25 | 330351.384387 | 321254.463405 |
| 966 | 68 | 10 | 0.25 | 337962.399198 | 329606.663985 |
| 1102 | 76 | 10 | 0.25 | 385372.431742 | 376530.056249 |
| 1149 | 127 | 10 | 0.25 | 529627.844251 | 516435.343938 |
| 1184 | 130 | 10 | 0.25 | 545340.762064 | 531377.485355 |
| 1206 | 86 | 10 | 0.25 | 434705.450777 | 424823.785653 |
| 1288 | 90 | 10 | 0.25 | 461257.277745 | 451382.89648 |
| 1549 | 169 | 10 | 0.25 | 729124.244066 | 701894.583657 |
| 1612 | 113 | 10 | 0.25 | 587309.340404 | 575305.302626 |
| 1802 | 127 | 10 | 0.25 | 666645.03627 | 650960.768662 |
| 1866 | 130 | 10 | 0.25 | 686638.036543 | 669968.400955 |
| 2367 | 169 | 10 | 0.25 | 904435.683545 | 870912.869511 |

Table 11: Empirical Evaluation of OPTIMAL-SPACE-LRF.

the error incurred with that of expected error promised by our theoretical results while keeping the dimensions and the value of $k$ constant. The result of our experiment is presented in Table 10.

Recall that the additive error incurred by our algorithm is

$$\gamma := O\left(\eta \alpha^{-2} \log(k/\delta) \sqrt{m \log(1/\delta)}/\epsilon\right)$$

for $\eta = \max\{k, 1/\alpha\}$.

Since every entries of the matrix $\mathbf{A} - [\mathbf{A}]_k$ is identically zero, the error of our algorithm is due to the additive error. Again as predicted by our results, we notice that the same trend as in Table 7:

We run our algorithm for $m = \{450, 700\}$ and $n = 50$ with $\alpha = 0.25$ and $k$ ranging from 10 to 19. Our algorithm consistently perform better than the additive error guaranteed by the theoretical results. The empirical error is way better than the error predicted by the theoretical result for small values of $\alpha$ and it starts getting closer as the values of $\alpha$ increases.

### K.3 Empirical Evaluation of OPTIMAL-SPACE-LRF

In this section, we understand the result of our empirical evaluation of PRIVATE-LOCAL-LRF for the detail description). Recall that in the non-private setting, we want to output a low-rank factorization such that its product $\mathbf{M}_k$ satisfies the following inequality with high probability:

$$\|\mathbf{M}_k - \mathbf{A}\|_F \leq (1 + \alpha)\|\mathbf{A} - [\mathbf{A}]_k\|_F.$$

For the empirical evaluation of our non-private algorithm, we sampled matrices such that every entries are uniform real number between $[0, 5000]$. In our experimental set-up, we keep the value of $\alpha = 0.25$ and $k = 10$ fixed to get a better understanding of how the approximation changes with the changing values of dimensions. Our empirical results are listed in Table 11. We see that for all the ranges of dimensions we evaluated on, the value in the column marked as $\|\mathbf{A} - \mathbf{M}_k\|_F$ is well within a $(1 + \alpha)$ factor of the corresponding entries marked under the column $\|\mathbf{A} - [\mathbf{A}]_k\|_F$. In fact, empirical evidence suggests that our algorithms gives a much better approximation than $\alpha = 0.25$ (in fact, it is closer to $\alpha = 0.05$). This gives a clear indication that our algorithm performs as the theoretical bound suggests for a large range of dimensions.

## L Future Work

In this paper, we initialize the study of low-rank factorization when the data matrix is streamed or is distributed over multiple servers giving various upper and lower bounds. Our work raises few questions which would shed more light on differentially private algorithms in various setting.

1. In the streaming setting, our results suggest interesting research directions by making the relationship between low-space algorithms and low error differential private algorithms more explicit In this context, some previous works used it implicitly to improve run-time efficiency [15, 11, 59, 92] and additive error [13, 92] — Blocki *et al.* [13] also incur a multiplicative error that can lead to large error for dense graphs. Some direct questions are as follows:
   1. propose streaming algorithms that are robust against sketches being noisy (and not leaky storage for which error-correcting or leakage resilient techniques suffice [78]), and
   2. characterize the relationship between low-space algorithms and privacy.

2. In the local model, an interesting question from our result is to investigate how close or far we are from optimal error. Towards this goal, an interesting question is whether we can use the minimax framework [4, 34, 96] to prove a lower bound on the accuracy. Our empirical evaluation shows that we outperform Hardt and Roth [47] even when the matrices are incoherent. This raises the question whether our algorithms achieve better results when matrices are incoherent?

## M Analysis of Boutsidis *et al.* Under Noisy Storage

The algorithm of Boutsidis *et al.* [17] maintains fives sketches, $\mathbf{M} = \mathbf{T}_l \mathbf{A} \mathbf{T}_r$, $\mathbf{L} = \mathbf{S} \mathbf{A} \mathbf{T}_r$, $\mathbf{N} = \mathbf{T}_l \mathbf{A} \mathbf{R}$, $\mathbf{D} = \mathbf{A} \mathbf{R}$ and $\mathbf{C} = \mathbf{S} \mathbf{A}$, where $\mathbf{S}$ and $\mathbf{R}$ are the embedding for the generalized linear regression and $\mathbf{T}_l$ and $\mathbf{T}_r$ are affine embedding matrices. It then computes

$$\mathbf{X}_* = \underset{\mathsf{r}(\mathbf{X}) \leq k}{\operatorname{argmin}} \|\mathbf{N}\mathbf{X}\mathbf{L} - \mathbf{M}\|_F$$

and its SVD as $\mathbf{X}_* = \mathbf{U}_* \boldsymbol{\Sigma}_* \mathbf{V}_*^\mathsf{T}$. It then outputs $\mathbf{D}\mathbf{U}_*$, $\boldsymbol{\Sigma}_*$, and $\mathbf{V}_*^\mathsf{T}\mathbf{C}$. Note that it does not compute a singular value decomposition of the low-rank approximation, but a different form of factorization.

Boutsidis *et al,* [17] prove the following three lemmas, combining which they get their result.

**Lemma 19.** *For all matrices $\mathbf{X} \in \mathbb{R}^{t \times t}$, with probability at least $98/100$, we have*

$$(1-\alpha)^2 \|\mathbf{A}\mathbf{R}\mathbf{X}\mathbf{S}\mathbf{A} - \mathbf{A}\|_F^2 \leq \|\mathbf{T}_l(\mathbf{A}\mathbf{R}\mathbf{X}\mathbf{S}\mathbf{A} - \mathbf{A})\mathbf{T}_r\|_F^2 \leq (1+\alpha)^2 \|\mathbf{A}\mathbf{R}\mathbf{X}\mathbf{S}\mathbf{A} - \mathbf{A}\|_F^2.$$

**Lemma 20.** *Let $\widetilde{\mathbf{X}} = \operatorname{argmin}_{\mathsf{r}(\mathbf{X}) \leq k} \|\mathbf{A}\mathbf{R}\mathbf{X}\mathbf{S}\mathbf{A} - \mathbf{A}\|_F^2$. Then with probability at least $98/100$, we have*

$$\min_{\mathsf{r}(\mathbf{X}) \leq k} \|\mathbf{T}_l(\mathbf{A}\mathbf{R}\mathbf{X}\mathbf{S}\mathbf{A} - \mathbf{A})\mathbf{T}_r\|_F^2 \leq \|\mathbf{A}\mathbf{R}\mathbf{X}_*\mathbf{S}\mathbf{A} - \mathbf{A}\|_F^2 \leq \|\mathbf{A}\mathbf{R}\widetilde{\mathbf{X}}\mathbf{S}\mathbf{A} - \mathbf{A}\|_F^2.$$

**Lemma 21.** *Let $\widetilde{\mathbf{X}} = \operatorname{argmin}_{\mathsf{r}(\mathbf{X}) \leq k} \|\mathbf{A}\mathbf{R}\mathbf{X}\mathbf{S}\mathbf{A} - \mathbf{A}\|_F^2$. Then with probability at least $98/100$, we have*

$$\|\mathbf{A}\mathbf{R}\widetilde{\mathbf{X}}\mathbf{S}\mathbf{A} - \mathbf{A}\|_F^2 \leq (1+\alpha)\Delta_k(\mathbf{A})^2.$$

To compute the value of $\widetilde{\mathbf{X}}$, they use a result in generalized rank-constrained matrix approximations, which says that

$$\mathbf{N}^\dagger \left[\mathbf{U_N}\mathbf{U_N^\mathsf{T}}\mathbf{M}\mathbf{V_L}\mathbf{V_L^\mathsf{T}}\right]_k \mathbf{L}^\dagger = \underset{\mathsf{r}(\mathbf{X}) \leq k}{\operatorname{argmin}} \|\mathbf{N}\mathbf{X}\mathbf{L} - \mathbf{M}\|_F, \tag{32}$$

where $\mathbf{U_N}$ is the matrix whose columns are the left singular vectors of $\mathbf{N}$ and $\mathbf{V_L}$ is the matrix whose columns are the right singular vectors of $\mathbf{L}$.

Now, in order to make the above algorithm differentially private, we need to use the same trick as we used in the algorithm for PRIVATE-OPTIMAL-SPACE-LRF; otherwise, we would suffer from the problems mentioned earlier in Section **??**. More precisely, we compute $\widehat{\mathbf{A}} = (\mathbf{A} \quad \sigma_{\mathsf{min}}\mathbb{I})$ and then store the following sketches: $\mathbf{M} = \mathbf{T}_l\widehat{\mathbf{A}}\mathbf{T}_r + \mathbf{N}_1$, $\mathbf{L} = \mathbf{S}\widehat{\mathbf{A}}\mathbf{T}_r + \mathbf{N}_2$, $\mathbf{N} = \mathbf{T}_l\widehat{\mathbf{A}}\mathbf{R} + \mathbf{N}_3$, $\mathbf{D} = \widehat{\mathbf{A}}\mathbf{R}$ and $\mathbf{C} = \mathbf{S}\widehat{\mathbf{A}} + \mathbf{N}_4$, where $\mathbf{R} = t^{-1}\boldsymbol{\Omega}\boldsymbol{\Phi}$ with $\boldsymbol{\Omega}$ being a Gaussian matrix.

| | Neighbouring Data Assumptions | Multiplicative Error Additive Error |
|---|---|---|
| Theorem [3] | $\mathbf{A} - \mathbf{A}' = \mathbf{u}\mathbf{v}^\mathsf{T}$ | $(1+\alpha)$ |
| $(\varepsilon, \delta)$-differential privacy | No assumption | $\widetilde{O}\left(\left(\sqrt{m\eta/\alpha} + \sqrt{kn}\right)\varepsilon^{-1}\right)$ |
| Theorem [11] | $\|\mathbf{A} - \mathbf{A}'\|_F = 1$ | $(1+\alpha)$ |
| $(\varepsilon, \delta)$-differential privacy | No assumption | $\widetilde{O}\left(\left(\sqrt{m\eta\alpha^{-2}} + \sqrt{kn}\right)\varepsilon^{-1}\right)$ |
| Hardt-Roth [47] | $\mathbf{A} - \mathbf{A}' = \mathbf{e}_s\mathbf{v}^\mathsf{T}$ | $\sqrt{2}$ |
| $(\varepsilon, \delta)$-differential privacy | $\mu_0$-coherent matrix | $\widetilde{O}\left(\frac{1}{\varepsilon}\left(\sqrt{km} + k\left(\frac{n}{m}\right)^{1/4}\sqrt{\mu_0\|\mathbf{A}\|_F}\right)\right)$ |
| Kapralov and Talwar [57] | $\|\mathbf{A}\|_2 - \|\mathbf{A}'\|_2 = 1$ | $1$ |
| $(\varepsilon, 0)$-differential privacy | Singular-value separation (SVS) | $c\lambda_1, \text{ where } \lambda_1 = \Omega\left(\frac{nk^3}{\varepsilon c^6}\right)$ |
| Upadhyay [93] | $\mathbf{A} - \mathbf{A}' = \mathbf{e}_s\mathbf{v}^\mathsf{T}$ | $\mathsf{poly}(k)$ |
| $(\varepsilon, \delta)$-differential privacy | No assumption | $\widetilde{O}\left(k^2\varepsilon^{-1}\sqrt{(m+n)}\right)$ |
| Hardt and Price [46] | $\mathbf{A} - \mathbf{A}' = \mathbf{e}_s\mathbf{e}_t^\mathsf{T}$ | $1$ |
| $(\varepsilon, \delta)$-differential privacy | Incoherence and SVS | $\widetilde{O}\left(\frac{\sigma_1\sqrt{k\mu\log d\log(\log d\sigma_k/(\sigma_k-\sigma_{k+1}))}}{\varepsilon(\sigma_k-\sigma_{k+1})}\right)$ |
| Hardt and Price [46] | $\mathbf{A} - \mathbf{A}' = \mathbf{e}_s\mathbf{e}_t^\mathsf{T}$ | $1$ |
| $(\varepsilon, \delta)$-differential privacy | SVS | $\widetilde{O}\left(\frac{\sigma_1\sqrt{kd\log d\log(\log(d)\sigma_k/(\sigma_k-\sigma_{k+1}))}}{\varepsilon(\sigma_k-\sigma_{k+1})}\right)$ |
| Dwork *et al.* [40] | $\mathbf{A} - \mathbf{A}' = \mathbf{e}_s\mathbf{v}^\mathsf{T}$ | $1$ |
| $(\varepsilon, \delta)$-differential privacy | No assumptions | $\widetilde{O}\left(k\sqrt{n}/\varepsilon\right)$ |
| Jiang *et al.* [53] | $\mathbf{A} - \mathbf{A}' = \mathbf{e}_s\mathbf{e}_t^\mathsf{T}$ | $1$ |
| $(\varepsilon, 0)$-differential privacy | No assumption | $\widetilde{O}\left(n\varepsilon^{-1}\log n\right)$ |

Table 12: Comparison of Models for Differentially Private $k$-Rank Approximation ($\beta = O(1)$, $\mathbf{u}$ and $\mathbf{v}$ are unit vectors, $\mathbf{e}_s$ is the $s$-th standard basis, and $d = m + n$).

As in the proof of Theorem 13, there are three terms that contributes to the additive error. The first and the third term are the same as in the proof of Theorem 13. However, the second term differs. In order to compute the additive error incurred due to the second term, we note that $\mathbf{X}$ in Lemma 19 needs to be

$$\mathbf{X} := \widehat{\mathbf{X}} = \underset{r(\mathbf{X})\leq k}{\mathrm{argmin}} \|\mathbf{T}_l(\widehat{\mathbf{A}}\mathbf{R}\mathbf{X}\mathbf{S}\widehat{\mathbf{A}} - \widehat{\mathbf{A}})\mathbf{T}_r\|_F^2.$$

The reason for this value of $\widehat{\mathbf{X}}$ is the same as in the proof of Claim 1. Moreover, all the occurrence of $\mathbf{A}$ is replaced by $\widehat{\mathbf{A}}$. In other words, we get the following by combining Lemma 19, Lemma 20, and Lemma 21.

$$(1 - \alpha)^2 \|\widehat{\mathbf{A}}\mathbf{R}\widehat{\mathbf{X}}(\mathbf{S}\widehat{\mathbf{A}} + \mathbf{N}_4) - \widehat{\mathbf{A}}\|_F^2 \leq (1+\alpha)\Delta_k(\widehat{\mathbf{A}})^2.$$

In other words, the additive error incurred by this expression is $(1 - \alpha)\|\widehat{\mathbf{A}}\mathbf{R}\widehat{\mathbf{X}}\mathbf{N}_4\|_F$. Using equation (32), we have $\gamma_2 = (1 - \alpha)\|\widehat{\mathbf{A}}\mathbf{R}\mathbf{N}^\dagger\left[\mathbf{U_N}\mathbf{U_N}^\mathsf{T}\mathbf{M}\mathbf{V_L}\mathbf{V_L}^\mathsf{T}\right]_k \mathbf{L}^\dagger\mathbf{N}_4\|_F$, where $\mathbf{N}, \mathbf{M}$, and $\mathbf{L}$ are as defined above. This term depends on the singular values of $\widehat{\mathbf{A}}$ and hence can be arbitrarily large.

# N  Problems Studied in Previous Works

Table 12 gives all the previous results under the assumptions made and the problem studied. Below, we define each of these problems and their difference from the problem studied in this paper.

## N.1  Low Rank Approximation With Respect to the Frobenius Norm

Hardt and Roth [47] and Upadhyay [93] studied the following problem.

**Problem 2.** *(Approximation with respect to the Frobenius norm). Given parameters $\alpha, \beta, \tau$, a private $m \times n$ matrix $\mathbf{A}$ (where $m \ll n$) and the target rank $k$, find a rank-$k$ matrix $\widetilde{\mathbf{A}}_k$ such that*

$$\mathsf{Pr}\left[\|\mathbf{A} - \widetilde{\mathbf{A}}_k\|_F \leq (1+\alpha)\|\mathbf{A} - [\mathbf{A}]_k\|_F + \tau\right] \geq 1 - \beta.$$

*Here, two matrices are neighbouring if they differ by single row of unit norm.*

**Difference from this paper**  We consider low-rank factorization while Problem 2 studied only low-rank approximation. Moreover, the granularity of privacy we consider is more general than theirs.

### N.2 Low Rank Approximation With Respect to the Spectral Norm

We can also study low-rank approximation when the approximation metric is spectral norm, which was the focus of Kapralov and Talwar [57], Hardt and Roth [48], and Hardt and Price [46].

**Problem 3.** *(Approximation with respect to the spectral norm). Given parameters* $\alpha, \beta, \tau$*, a private* $m \times n$ *matrix* $\mathbf{A}$ *(where* $m \ll n$*) and the target rank* $k$*, find a rank-*$k$ *matrix* $\widetilde{\mathbf{A}}_k$ *such that*

$$\Pr\left[\|\mathbf{A} - \widetilde{\mathbf{A}}_k\| \leq (1 + \alpha)\|\mathbf{A} - [\mathbf{A}]_k\| + \tau\right] \geq 1 - \beta.$$

*Hardt and Price [46] and Jiang et al. [53] consider two matrices as neighboring if they differ in exactly one entry by at most* 1*. Kapralov and Talwar [57] considered two matrices as neighboring if the difference of their spectral norm is at most* 1*.*

**Difference from this paper**  We consider low-rank factorization with respect to the Frobenius norm while Problem 3 studied only low-rank approximation with respect to the spectral norm. Moreover, granularity of privacy we consider is more general than theirs.

### N.3 Approximating the Right Singular Vectors

Dwork *et al.* [40] studied the following problem.

**Problem 4.** *Given parameters* $\alpha, \beta, \tau$ *and an* $m \times n$ *private matrix* $\mathbf{A}$ *(where* $m \gg n$*)), compute a rank-*$k$ *matrix* $\widetilde{\mathbf{B}}_k$ *such that*

$$\Pr\left[\|\mathbf{A}^\mathsf{T}\mathbf{A} - \widetilde{\mathbf{B}}_k\| \leq \min_{\mathsf{rank}(\mathbf{B}_k) \leq k}\|\mathbf{A}^\mathsf{T}\mathbf{A} - [\mathbf{A}^\mathsf{T}\mathbf{A}]_k\| + \tau\right] \geq 1 - \beta,$$

*where* $\|\cdot\|$ *denotes either the spectral or the Frobenius norm.*

Dwork *et al.* [40] consider two matrices neighbouring if they differ by at most one row. They further assume that the rows are normalized; therefore, their definition of neighbouring matrices is the same as Hardt and Roth [47].

**Difference from this paper**  We consider low-rank factorization of both the right and the left singular vectors while Problem 4 studied low-rank "approximation" of the right singular vectors. Moreover, granularity of privacy we consider is more general than theirs.

## O   Improving the Run-time Efficiency of Factorization

If we want a more efficient algorithm, then we can replace $\mathbf{S}$ by the product of a $v \times v'$ subsampled randomized Hadamard matrix and a $v' \times m$ sparse subspace embedding matrix, also known as count-sketch matrix [23, 71] (here, $v = O(k\alpha^{-3}\log(k/\alpha))$ and $v' = O(k^2\alpha^{-4}\log^6(k/\alpha))$). Both of these matrices satisfies  Definition 7 [22, Lemma 46]. In that case, we have the following claim, which is analogous to the result presented in Lemma 5. In other words, we can use the following claim everywhere where Lemma 5 is used.

**Lemma 22.** *Let* $\mathbf{S}$ *be the product formed by a* $v \times v'$ *subsampled randomized and normalized Hadamard matrix and a* $v' \times m$ *sparse subspace embedding matrix. Then with high probability, for any* $v' \times n$ *matrix* $\mathbf{N}$*, we have* $\|\mathbf{S}^\dagger \mathbf{N}\|_F = \|\mathbf{N}\|_F$.

*Proof.* First note that $\mathbf{S} = (\mathbf{\Pi}_{1..v'}\mathbf{W}\mathbf{D}_1)(\mathbb{H}\mathbf{D}_2)$ , where $\mathbf{D}_1$ is a $v \times v$ diagonal matrix with non-zero entries sampled from $\mathsf{Rad}(1/2)$, $\mathbf{D}_2$ is an $m \times m$ diagonal matrix with non-zero entries sampled from $\mathsf{Rad}(1/2)$, $\mathbf{W}$ is a normalized Hadamard matrix, and $\mathbb{H}$ is a matrix formed using a random hash function $h : [m] \to [v]$ such that entry $\mathbb{H}_{ij} = 1$ if and only if $h(j) = i$ and 0, otherwise. Therefore, $\mathbf{S}^\dagger = ((\mathbf{\Pi}_{1..v'}\mathbf{W}\mathbf{D}_1)(\mathbb{H}\mathbf{D}_2))^\dagger$.

We claim that $\sqrt{(v/m)}\mathbb{H}\mathbf{D}_2$ is a matrix with orthonormal rows. First note that $\mathbf{D}_2$ is a diagonal matrix with only $\pm 1$ non-zero entries; therefore, if we prove that $\sqrt{v/m}\mathbb{H}$ is a matrix with orthonormal rows, then we are done. Now, consider an entry $(\mathbb{H}\mathbb{H}^{\mathsf{T}})_{ij}$. Let $\mathbb{H}_{i:}$ denote the $i$-th row of the matrix $\mathbb{H}$. Note that $\mathbb{H}$ is a full row-rank matrix. Therefore, by the construction of the matrix $\mathbb{H}$, we have the following:

$$(\mathbb{H}\mathbb{H}^{\mathsf{T}})_{ij} = \begin{cases} 0 & i \neq j \\ \|\mathbb{H}_{i:}\|_0 & i = j \end{cases},$$

where $\|\mathbb{H}_{i:}\|_0$ denotes the number of non-zero entries in the row $\mathbb{H}_{i:}$. Now, since $h(\cdot)$ is a random function that maps to every entry in $[v]$ uniformly, $\mathbb{E}[\|\mathbb{H}_{i:}\|_0] = m/v$. Chernoff bound now gives that, with high probability, $\|\mathbb{H}\|_0 = m/v$. Therefore, $v/m(\mathbb{H}\mathbb{H}^{\mathsf{T}})_{ij} = 1$ when $i = j$. In other words, with high probability, $(\mathbb{H}\mathbb{H}^{\mathsf{T}})^{-1} = (v/m)\mathbb{I}$.

Now returning to the proof, conditioned on the event that $\sqrt{v/m}\mathbb{H}\mathbf{D}_2$ has orthonormal rows, we can write $\mathbf{S}^{\dagger} = (\mathbb{H}\mathbf{D}_2)^{\dagger}\left(\boldsymbol{\Pi}_{1..v'}\mathbf{W}\mathbf{D}_1\right)^{\dagger}$. Let $\widetilde{\mathbf{S}} = (\boldsymbol{\Pi}_{1..v'}\mathbf{W}\mathbf{D}_1)$. Then we can invoke the proof of Lemma 5 to say that $\widetilde{\mathbf{S}}\mathbf{N}$ gives a matrix $\widehat{\mathbf{N}}$ with entries of $\mathbf{N}$ permuted according to the permutation $\boldsymbol{\Pi}_{1..v'}$. Therefore, we have $\|\mathbf{S}^{\dagger}\mathbf{N}\|_F = \|(\widehat{\mathbb{H}}\mathbf{D}_2)^{\dagger}\widehat{\mathbf{N}}\|_F$ for some random $v \times n$ Gaussian matrix $\widehat{\mathbf{N}}$ whose entries are sampled i.i.d. from $\mathcal{N}(0,1)$.

Since $\mathbf{D}_2$ is matrix with orthonormal columns, we have $(\mathbb{H}\mathbf{D}_2)^{\dagger}\widehat{\mathbf{N}} = \mathbf{D}_2^{\dagger}\mathbb{H}^{\dagger}\widehat{\mathbf{N}}$. Therefore, with high probability,

$$\|\mathbf{S}^{\dagger}\mathbf{N}\|_F = \|\mathbb{H}^{\dagger}\widehat{\mathbf{N}}\|_F = \|\mathbb{H}^{\mathsf{T}}(\mathbb{H}\mathbb{H}^{\mathsf{T}})^{-1}\widehat{\mathbf{N}}\|_F = \frac{v}{m}\|\mathbb{H}^{\mathsf{T}}\widehat{\mathbf{N}}\|_F.$$

Now, we can write $\frac{v}{m}\|\mathbb{H}^{\mathsf{T}}\widehat{\mathbf{N}}\|_F^2$ as

$$\frac{v}{m}\|\mathbb{H}^{\mathsf{T}}\widehat{\mathbf{N}}\|_F^2 = \frac{v}{m}\sum_{i=1}^{m}\sum_{j=1}^{n}\left(\sum_{k=1}^{v}\mathbb{H}_{k,i}\widehat{\mathbf{N}}_{k,j}\right)^2$$

$$= \frac{v}{m}\sum_{i=1}^{m}\sum_{j=1}^{n}\widehat{\mathbf{N}}_{h(i),j}^2 = \|\widehat{\mathbf{N}}\|_F^2$$

which completes the proof of Lemma 22. $\qquad\square$

# P  Source Codes for Our Experiments

We run our algorithm of random matrices of size that is randomly sampled. In other words, all the private matrices used in our evaluation are dense matrices. As a result, there is no benefit of using the random projection matrices of Clarkson and Woodruff [23]. Therefore, to ease the overload on the code, we use random Gaussian matrices with appropriate variance as all our random projection matrices.

## P.1  Source Code for OPTIMAL-SPACE-LRF

```python
import math
from numpy import linalg as la

def randomGaussianMatrix(rows,columns,variance):
    G = np.zeros((rows, columns))
    for i in range(rows):
        for j in range(columns):
            G[i][j] = np.random.normal(0, variance)
    return G

def computeSingular(singular, Matrix):
    Sigma = np.zeros(np.shape(Matrix))
    i = 0
    while i < len(Matrix) and i < len(Matrix[0]):
        Sigma[i][i] = singular[i]
        i += 1
    return Sigma

def lowrank(k, A):
    a = np.zeros(np.shape(A))
    Left, singular, Right = np.linalg.svd(A,full_matrices=1, compute_uv=1)
    singularmatrix = computeSingular(singular,A)
    top_right = Right[:k,:]
    top_left = Left[:,:k]
    top_singular = singularmatrix[:k,:k]
    a += np.dot(np.dot(top_left, top_singular),top_right)
    return a

def Upadhyay(Ycolumn, Yrow, Z, S, T):
    U, sc, Vc = np.linalg.svd(Ycolumn)
    Ur, sr, V = np.linalg.svd(Yrow)
    t = len(Yrow)
    v = len(S)
    U = U[:, :t]
    V = V[:t, :]
    TopU = np.dot(S, U)
    TopV = np.dot(V, T)
    Us, Ss, Vs = np.linalg.svd(TopU)
    Ut, St, Vt = np.linalg.svd(TopV)
    inn = np.dot( Us.T, np.dot( Z, Vt.T) )
    innlow = lowrank(k,inn)
    SingularS = computeSingular( np.reciprocal(Ss), TopU )
    SingularT = computeSingular( np.reciprocal(St), TopV )
    outerT = np.dot(innlow ,np.dot(SingularT.T, Ut.T) )
    outerS = np.dot(Vs.T,np.dot( SingularS.T, outerT))
    B = np.dot(U, np.dot(outerS ,V))
    return B

def Initialization(A,m,n,k,alpha):
    Ak = lowrank(k, A)
    t = int(k/alpha)
    v = int(k/(alpha ** 2))
    Phi = randomGaussianMatrix(n,t,1/t)
    Psi = randomGaussianMatrix(t,m,1/t)
```

```
    S = randomGaussianMatrix(v,m,1/v)
    T = randomGaussianMatrix(n,v,1/v)
    Ycolumn = np.dot(A, Phi)
    Yrow = np.dot(Psi, A)
    Z = np.dot(S, np.dot(A, T))
    B =Upadhyay(Ycolumn, Yrow, Z, S, T)
    frobupadhyay = la.norm(A - B, 'fro')
    actual = la.norm(A - Ak, 'fro')
    print (m, n, k,alpha, frobupadhyay,actual)

i=1
n=50
while n< 200:
    m=9*n+int(50*np.random.rand())
    alpha=0.25
    k=10
    Initialization(randomMatrix(m,n,1,5000), m,n,k,alpha)
    Initialization(5000*np.random.rand(m, n), m,n,k,alpha)
    m = 14 * n + int(50*np.random.rand())
    Initialization(randomMatrix(m,n,1,5000), m,n,k,alpha)
    Initialization(5000*np.random.rand(m, n),m,n,k,alpha)
    n=50 + (i * int(20 * np.random.rand()))
    i=i + 1
```

## P.2 Source Code for Comparing Private Algorithms

In this section, we present our source code to compare our algorithm with that of Hardt and Roth [47].

```
import math
from numpy import linalg as la

def randomGaussianMatrix(rows,columns,variance):
    G = np.zeros((rows, columns))
    for i in range(rows):
        for j in range(columns):
            G[i][j] = np.random.normal(0, variance)
    return G

def randomMatrix(rows,columns,a,b):
    A = np.zeros((rows, columns))
    for i in range(rows):
        for j in range(columns):
            A[i][j] = np.random.randint(a, b)
    return A

def computeSingular(singular, Matrix):
    Sigma = np.zeros(np.shape(Matrix))
    i = 0
    while i < len(Matrix) and i < len(Matrix[0]):
        Sigma[i][i] = singular[i]
        i += 1
    return Sigma

def lowrank(k, A):
    a = np.zeros(np.shape(A))
    Left, singular, Right = np.linalg.svd(A,full_matrices=1, compute_uv=1)
    singularmatrix = computeSingular(singular,A)
    top_right = Right[:k,:]
    top_left = Left[:,:k]
    top_singular = singularmatrix[:k,:k]
    a +=  np.dot(np.dot(top_left, top_singular),top_right)
    return a

def Upadhyay(Ycolumn, Yrow, Z, S, T):
```

```python
    U, sc, Vc = np.linalg.svd(Ycolumn)
    Ur, sr, V = np.linalg.svd(Yrow)
    t = len(Yrow)
    v = len(S)
    U = U[:, :t]
    V = V[:t, :]
    TopU = np.dot(S, U)
    TopV = np.dot(V, T)
    Us, Ss, Vs = np.linalg.svd(TopU)
    Ut, St, Vt = np.linalg.svd(TopV)
    inn = np.dot( Us.T, np.dot( Z, Vt.T) )
    innlow = lowrank(k,inn)
    SingularS = computeSingular( np.reciprocal(Ss), TopU )
    SingularT = computeSingular( np.reciprocal(St), TopV )
    outerT = np.dot(innlow ,np.dot(SingularT.T, Ut.T) )
    outerS = np.dot(Vs.T,np.dot( SingularS.T, outerT))
    B = np.dot(U, np.dot(outerS ,V))
    return B

def Initialization(A,m,n,k,alpha):
    Ak = lowrank(k, A)
    actual = la.norm(A - Ak, 'fro')
    HR = HardtRoth(A,m,n,k)
    frobHR = la.norm(A - HR, 'fro')
    m = m+n
    epsilon = 1
    delta = 1/(m)
    t = int(k/alpha)
    v = int(k/(alpha ** 2))
    Phi = randomGaussianMatrix(n,t,1/t)
    Psi = randomGaussianMatrix(t,m,1/t)
    S = randomGaussianMatrix(v,m,1/v)
    T = randomGaussianMatrix(n,v,1/v)
    sigma = 4*math.log(1/delta)*math.sqrt(t*math.log(1/delta)) / epsilon
    Ahat = A
    scaledI = sigma*np.identity(n)
    for i in range(n):
        Ahat = np.vstack([Ahat, scaledI[i]])
    for i in range(n):
        A = np.vstack([A, scaledI[i] - scaledI[i]])
    rho1 = math.sqrt(-4*math.log(delta)/epsilon**2)
    rho2 = math.sqrt(-6*math.log(delta)/epsilon**2)
    Ycolumn = np.dot(Ahat, Phi) + randomGaussianMatrix(m,t,rho1)
    Yrow = np.dot(Psi, Ahat)
    Z = np.dot(S, np.dot(Ahat, T)) + randomGaussianMatrix(v,v,rho2)
    L = np.dot(Yrow, T)
    N = np.dot(S, Ycolumn)
    B =Upadhyay(Ycolumn, Yrow, Z, S, T)
    frobupadhyay = la.norm(Ahat - B, 'fro')
    print (m, n, k,alpha,frobupadhyay,frobHR,actual)

def HardtRoth(A,m,n,k):
    Omega = randomGaussianMatrix(n,2*k,1)
    Y = np.dot(A, Omega)
    epsilon = 1
    delta = 1/(m)
    sigma = - 32 * k * math.log(delta)/epsilon**2
    N = randomGaussianMatrix(m, 2*k , sigma)
    Y += N
    Q = np.zeros(Y.shape)
    for i in range(Y.shape[1]):
        avec = Y[:, i]
        q = avec
        for j in range(i):
            q = q - np.dot(avec, Q[:, j]) * Q[:, j]
```

```
        Q[:, i] = q / la.norm(q)
    a = Q[0][0]
    for i in range(Y.shape[0]):
        for j in range(Y.shape[1]):
            if a < Q[i][j]:
                a = Q[i][j]
    rho =   32*(a**2)*k*math.log(8*k/delta)*math.log(1/delta)/epsilon**2
    inner = np.dot(Q.T, A) + randomGaussianMatrix(2*k,n,rho)
    output = np.dot(Q, inner)
    return output

i=1
n=50
while n < 200:
    m = 9 * n + int(50*np.random.rand())
    alpha = 0.25
    k = 10
    Initialization(randomMatrix(m,n,1,5000), m,n,k,alpha)
    Initialization(5000*np.random.rand(m, n), m,n,k,alpha)
    m = 14 * n + int(50*np.random.rand())
    Initialization(randomMatrix(m,n,1,5000), m,n,k,alpha)
    Initialization(5000*np.random.rand(m, n),m,n,k,alpha)
    n = 50 + (i * int(20 * np.random.rand()))
    i = i + 1
```

**P.3  Source Code for Analyzing Additive Error for OPTIMAL-SPACE-PRIVATE-**LRF

```
import numpy as np
import math
from numpy import linalg as la

def randomGaussianMatrix(rows, columns, variance):
    G = np.zeros((rows, columns))
    for i in range(rows):
        for j in range(columns):
            G[i][j] = np.random.normal(0, variance)
    return G

def randomMatrixUnit(rows, columns, a, b):
    return (b-a) ** np.random.rand(rows, columns)

def randomMatrix(rows, columns,k, a, b):
    A = np.zeros((rows, columns))
    for i in range(rows):
        a = 0
        for j in range(k):
            A[i][j] = np.random.randint(a, b)
    return A

def computeSingular(singular, Matrix):
    Sigma = np.zeros(np.shape(Matrix))
    i = 0
    while i < len(Matrix) and i < len(Matrix[0]):
        Sigma[i][i] = singular[i]
        i += 1
    return Sigma

def lowrank(k, A):
    a = np.zeros(np.shape(A))
    Left, singular, Right = np.linalg.svd(A, full_matrices=1, compute_uv=1)
    singularmatrix = computeSingular(singular, A)
    top_right = Right[:k, :]
    top_left = Left[:, :k]
```

```python
        top_singular = singularmatrix [:k, :k]
        a += np.dot(np.dot(top_left, top_singular), top_right)
        return a

def Upadhyay(A, m, n, k, alpha, epsilon, delta):
    m = m + n
    t = int(0.05 * k / alpha)
    v = int(0.05 * k / (alpha ** 2))
    Phi = randomGaussianMatrix(n, t, 1 / t)
    Psi = randomGaussianMatrix(t, m, 1 / t)
    S = randomGaussianMatrix(v, m, 1 / v)
    T = randomGaussianMatrix(n, v, 1 / v)
    sigma=4*math.log(1 / delta)*math.sqrt(t*math.log(1 / delta))/epsilon
    Ahat = A
    scaledI = sigma * np.identity(n)
    for i in range(n):
        Ahat = np.vstack([Ahat, scaledI[i]])
    for i in range(n):
        A = np.vstack([A, scaledI[i] - scaledI[i]])
    rho1 = math.sqrt(-4 * math.log(delta) / epsilon ** 2)
    rho2 = math.sqrt(-6 * math.log(delta) / epsilon ** 2)
    Ycolumn = np.dot(Ahat, Phi) + randomGaussianMatrix(m, t, rho1)
    Yrow = np.dot(Psi, Ahat)
    Z = np.dot(S, np.dot(Ahat, T)) + randomGaussianMatrix(v, v, rho2)
    U, sc, Vc = np.linalg.svd(Ycolumn)
    Ur, sr, V = np.linalg.svd(Yrow)
    U = U[:, :t]
    V = V[:t, :]
    TopU = np.dot(S, U)
    TopV = np.dot(V, T)
    Us, Ss, Vs = np.linalg.svd(TopU)
    Ut, St, Vt = np.linalg.svd(TopV)
    inn = np.dot(Us.T, np.dot(Z, Vt.T))
    innlow = lowrank(k, inn)
    SingularS = computeSingular(np.reciprocal(Ss), TopU)
    SingularT = computeSingular(np.reciprocal(St), TopV)
    outerT = np.dot(innlow, np.dot(SingularT.T, Ut.T))
    outerS = np.dot(Vs.T, np.dot(SingularS.T, outerT))
    B = np.dot(U, np.dot(outerS, V))
    expected =sigma * math.sqrt(n) + math.sqrt(-k *m * math.log(delta))
    return (B, Ahat, expected)

def HardtRoth(A, m, n, k, epsilon, delta):
    Omega = randomGaussianMatrix(n, 2 * k, 1)
    Y = np.dot(A, Omega)
    epsilon = 1
    delta = 1 / (m)
    sigma = - 32 * k * math.log(delta) / epsilon ** 2
    N = randomGaussianMatrix(m, 2 * k, sigma)
    Y += N
    Q = np.zeros(Y.shape)
    for i in range(Y.shape[1]):
        avec = Y[:, i]
        q = avec
        for j in range(i):
            q = q - np.dot(avec, Q[:, j]) * Q[:, j]

        Q[:, i] = q / la.norm(q)
    a = Q[0][0]
    for i in range(Y.shape[0]):
        for j in range(Y.shape[1]):
            if a < Q[i][j]:
                a = Q[i][j]
    rho = 32*(a**2)*k*math.log(8*k/delta)*math.log(1/delta)/epsilon**2
    inner = np.dot(Q.T, A) + randomGaussianMatrix(2 * k, n, rho)
```

```
        output = np.dot(Q, inner)
        return output

def Initialization(m, n, k, alpha):
    epsilon = 1
    delta = 1 / (m ** 2)
    A = randomMatrix(m,n,k,1,20)
    Ak = lowrank(k, A)
    actual = la.norm(A - Ak, 'fro')

    HR = HardtRoth(A, m, n, k, epsilon, delta)
    frobHR = la.norm(A - HR, 'fro')

    (Upadhyay16, Ahat, expected)=Upadhyay(A,m,n,k,alpha,epsilon,delta)
    frobupadhyay = la.norm(Ahat - Upadhyay16, 'fro')
    print(expected, frobupadhyay, frobHR)

#Use this part of the code for varying dimension
i = 1
n = 50
while n < 200:
    m = 9 * n + int(50 * np.random.rand())
    alpha = 0.1
    k = 10
    Initialization(randomMatrix(m, n, 0, 5), m, n, k, alpha)
    m = 14 * n + int(5 * np.random.rand())
    Initialization(randomMatrix(m, n, 0, 5), m, n, k, alpha)
    n = 50 + (i * int(5 * np.random.rand()))
    i = i + 1

#Use this part of the code for varying alpha
n=200
alpha = 0.10
while alpha < 0.25:
    m = 9 * n
    k = 10
    Initialization(m,n,k,alpha)
    m = 14 * n
    Initialization(m,n,k,alpha)
    alpha = alpha + 0.02

#Use this part of the code for varying k
n=50
k = 10
alpha = 0.25
while k<20:
    m = 9 * n
    Initialization(m,n,k,alpha)
    m = 14 * n
    Initialization(m,n,k,alpha)
    k +=1
```

## P.4   Source Code for PRIVATE-LOCAL-LRF

```
import math
from numpy import linalg as la

def randomGaussianMatrix(rows,columns,variance):
    G = np.zeros((rows, columns))
    for i in range(rows):
        for j in range(columns):
            G[i][j] = np.random.normal(0, variance)
    return G
```

```python
def randomMatrix(rows, columns, a, b):
    A = np.zeros((rows, columns))
    for i in range(rows):
        for j in range(columns):
            A[i][j] = np.random.randint(a, b)
    return A

def lowrank(k, A):
    a = np.zeros(np.shape(A))
    Left, singular, Right = np.linalg.svd(A, full_matrices=1, compute_uv=1)
    top_left = Left[:,:k]
    a += np.dot(np.dot(top_left, top_left.T),A)
    return a

def Local(A,Ycolumn, L, Z, S):
    N = np.dot(S,Ycolumn)
    Un, Sn, Vn = np.linalg.svd(N)
    Ul, Sl, Vl = np.linalg.svd(L)
    inner = np.dot(Un,np.dot(Un.T,np.dot(Z,np.dot(Vl.T,Vl))))
    innerlowrank = lowrank(k,inner)
    output = np.dot(la.pinv(N),np.dot(innerlowrank,la.pinv(L)))
    U, s, V = np.linalg.svd(output)
    Y = np.dot(Ycolumn, U)
    Q = np.zeros(Y.shape)
    for i in range(Y.shape[1]):
        avec = Y[:, i]
        q = avec
        for j in range(i):
            q = q - np.dot(avec, Q[:, j]) * Q[:, j]
        Q[:, i] = q / la.norm(q)
    B = np.dot(Q, np.dot(Q.T,A))
    return B

def Initialization(A,m,n,k,alpha):
    t = int(k / (alpha))
    v = int(k / (alpha ** 2))
    Ak = lowrank(k, A)
    actual = la.norm(A - Ak, 'fro')
    epsilon = 0.1
    delta = 1/(m**10)
    Phi = randomGaussianMatrix(n, t, 1 / t)
    Psi = randomGaussianMatrix(t, m, 1 / t)
    S = randomGaussianMatrix(v, m, 1 / v)
    T = randomGaussianMatrix(n, v, 1 / v)
    rho1 = math.sqrt(-4 * math.log(delta)/epsilon**2)
    rho2 = m * math.sqrt(-6 * math.log(delta)/epsilon**2)
    Ycolumn = np.dot(A, Phi) + randomGaussianMatrix(m,t,rho1)
    Yrow = np.dot(np.dot(Psi, A), T) + randomGaussianMatrix(t,v,rho2)
    Z = np.dot(S, np.dot(A, T)) + randomGaussianMatrix(v,v,rho2)
    B = Local(A, Ycolumn, Yrow, Z, S)
    frobupadhyay = la.norm(A - B, 'fro')
    print (m, n, k, alpha, frobupadhyay, actual)

i=1
n=50
while n < 200:
    m = 9 * n + int(50*np.random.rand())
    alpha = 0.25
    k = 10
    Initialization(randomMatrix(m,n,1,5000), m,n,k,alpha)
    Initialization(5000*np.random.rand(m, n), m,n,k,alpha)
    m = 14 * n + int(50*np.random.rand())
    Initialization(randomMatrix(m,n,1,5000), m,n,k,alpha)
    Initialization(5000*np.random.rand(m, n),m,n,k,alpha)
    n = 50 + (i * int(20 * np.random.rand()))
```

```
        i = i + 1
```

## P.5 Source Code for Analyzing Additive Error for PRIVATE-LOCAL-LRF

```python
import math
from numpy import linalg as la

def randomGaussianMatrix(rows, columns, variance):
    G = np.zeros((rows, columns))
    for i in range(rows):
        for j in range(columns):
            G[i][j] = np.random.normal(0, variance)
    return G

def randomMatrix(rows, columns, k, a, b):
    A = np.zeros((rows, columns))
    for i in range(rows):
        for j in range(k):
            A[i][j] = np.random.randint(a, b)
    return A

def lowrank(k, A):
    a = np.zeros(np.shape(A))
    Left, singular, Right = np.linalg.svd(A, full_matrices=1, compute_uv=1)
    top_left = Left[:,:k]
    a += np.dot(np.dot(top_left, top_left.T), A)
    return a

def Local(A, Ycolumn, L, Z, S):
    N = np.dot(S, Ycolumn)
    Un, Sn, Vn = np.linalg.svd(N)
    Ul, Sl, Vl = np.linalg.svd(L)
    inner = np.dot(Un, np.dot(Un.T, np.dot(Z, np.dot(Vl.T, Vl))))
    innerlowrank = lowrank(k, inner)
    output = np.dot(np.linalg.pinv(N), np.dot(innerlowrank, np.linalg.pinv(L)))
    U, s, V = np.linalg.svd(output)
    Y = np.dot(Ycolumn, U)
    Q = np.zeros(Y.shape)
    for i in range(Y.shape[1]):
        avec = Y[:, i]
        q = avec
        for j in range(i):
            q = q - np.dot(avec, Q[:, j]) * Q[:, j]

        Q[:, i] = q / la.norm(q)
    B = np.dot(Q, np.dot(Q.T, A))
    return B

def Initialization(m, n, k, alpha):
    t = int(0.05*k / (alpha))
    v = int(0.02*k / (alpha ** 2))
    A = randomMatrix(m, n, k, 1, 20)
    Ak = lowrank(k, A)
    actual = la.norm(A - Ak, 'fro')
    epsilon = 0.1
    delta = 1/(m**2)
    Phi = randomGaussianMatrix(n, t, 1 / t)
    Psi = randomGaussianMatrix(t, m, 1 / t)
    S = randomGaussianMatrix(v, m, 1 / v)
    T = randomGaussianMatrix(n, v, 1 / v)
    rho1 = math.sqrt(-4 * math.log(delta)/epsilon**2)
    rho2 = m * math.sqrt(-6 * math.log(delta)/epsilon**2)

    Ycolumn = np.dot(A, Phi) + randomGaussianMatrix(m, t, rho1)
```

```
        Yrow = np.dot(np.dot(Psi, A), T) + randomGaussianMatrix(t,v,rho2)
        Z = np.dot(S, np.dot(A, T)) + randomGaussianMatrix(v,v,rho2)
        B = Local(A, Ycolumn, Yrow, Z, S)
        frobupadhyay = la.norm(A - B, 'fro')
        expected = v * math.sqrt(-m * math.log(delta))/epsilon
        print (frobupadhyay,   expected)

#Use this part of the code for varying dimension
i = 1
n = 50
while n < 200:
    m = 9 * n + int(50 * np.random.rand())
    alpha = 0.1
    k = 10
    Initialization(randomMatrix(m, n, 0, 5), m, n, k, alpha)
    m = 14 * n + int(5 * np.random.rand())
    Initialization(randomMatrix(m, n, 0, 5), m, n, k, alpha)
    n = 50 + (i * int(5 * np.random.rand()))
    i = i + 1

#Use this part of the code for varying alpha
i=1
n=200
alpha = 0.10
while alpha < 0.25:
    m = 9 * n
    k = 10
    Initialization(m,n,k,alpha)
    m = 14 * n
    Initialization(m,n,k,alpha)
    alpha = alpha + 0.02

#Use this part of the code for varying k
n=50
k = 10
alpha = 0.25
while k<20:
    m = 9 * n
    Initialization(m,n,k,alpha)
    m = 14 * n
    Initialization(m,n,k,alpha)
    k +=1
```

## Footnotes

[1]A spectral decomposition of a symmetric matrix $\mathbf{A}$ is the representation of a matrix in form of its eigenvalues and eigenvectors: $\sum_i \lambda_i \mathbf{v}_i \mathbf{v}_i^\mathsf{T}$, where $\lambda_i$ are the eigenvalues of $\mathbf{A}$ and $\mathbf{v}_i$ is the eigenvector corresponding to $\lambda_i$.