[Reviews · NeurIPS 2018]

Reviewer 1



Low rank matrix factorization under privacy constraint is a well-studied problem in Differentially private Machine Learning. This problem has been studied in various models and tight bounds are known for various settings. The current work looks at this problem in distribute d, streaming, and continual release settings. It studies two natural notions of neighborhood from the point of view of defining privacy. The main contribution is a sketching based approach that gets tight bounds for differentially private PCA under these privacy models, whi le getting fast and simple sketching based algorithms. In a little more detail, the privacy models both consider two matrices to be neighboring if their differnce has Frobenious norm 1. In the first model priv_1, this difference is also rank 1, where priv_2 can handle arbitrary Frobenius norm 1 differences. There has been recen t work on these problems using sketching algorithms that gets good bounds in terms of space complexity, runtime, and communication. The c urrent paper shows that these algorithms based on sketching can be made differentially private, to yield near-optimal algorithm for vario us settings, without incurring a significant overhead in terms of space or time complexity. The authors show that this approach gives a streaming DP algorithm inthe turnstile model, where one can continually output an estimate of he best rank-k approximation to the matrix. The overhead in terms of error in Frobenius norm is $O(\sqrt{kd})$ for a $d \times d$ matrix , wiht a constant multiplicative error. Overall, I find that the paper makes progress on an interesting and important problem in DP Machine learning. I would therefore support a ccepting it.

Reviewer 2



SUMMARY: This is mostly a theory paper proposing new algorithms for low-rank matrix factorisation (LRF) that satisfy approximate (delta > 0) differential privacy (DP) and are also suitable for settings where the input matrix is distributed, or streamed, or the output is produced continually. The paper analyses the runtime, space requirements and the multiplicative and additive errors of the proposed algorithms. There is an attempt in the paper to provide the reader with sufficient background to understand how the proposed algorithms compare to prior work and what are the key new ideas. Unfortunately, the paper is at times difficult to read due to unclear writing. MAIN COMMENTS: 0/ [suitability] The main paper (8 pages) is essentially an advertisement for the actual, substantial journal paper appearing in the Appendix. Although I realize this appears to be common practice nowadays due to the page limitation, submitting to a journal instead would provide the benefit of the actual technical content of this work (currently in the appendices) being peer-reviewed. 1/ [significance] This could appear as another paper that takes an existing task X and simply looks at what happens when one wants to solve it under DP, proposing a new DP-X algorithm. However, I think X=LRF can be a particularly interesting problem to look at under the DP constraint, for the following reason. In DP we are constantly looking for ways to reduce the amount of noise required in order to reach a given privacy guarantee, and one folk wisdom general advice seems to be to avoid privatising high-dimensional objects (especially when some dimensions can carry little useful statistical information, but still need to be privacy-protected as we don't know a priori which dimensions these are). In studying DP-LRF one is forced to look at this question whether interest in a lower-dimensional description of the data allows one to get away with less noise being added, and this paper answers this question in the positive for the settings studied. I wonder if there might be some substance in this view and if it could help differentiate this paper from other DP-X papers. 2/ [significance] Is the proposed algorithm practical, or is it theoretical at this stage? Specifically, could you compute exact values of epsilon and delta when the algorithm is run? I couldn't spot the values of epsilon or delta in either the main text or in Appendix K. 3/ [originality] To my knowledge the paper is novel, and it is well discussed against existing work, describing the differences in assumptions, theoretical results and algorithmic ideas used. However, there is a bit of confusion especially in the Abstract about whether the proposed algorithms are the first ones to consider a given setting under the privacy constraint (lines 7-9), or whether they are improvements on top of existing ones (line 19). 4/ [clarity] The clarity of writing could be substantially improved at places (for example, in all of the important Section 2). The submission would also really benefit from just careful proof-reading for grammar (although I'm not penalising in my review for bad grammar, fixing even singular vs plural in many places would have made reading the submission easier). The submission has probably not even gone through a spell-checker. Please see also minor comments below for some concrete suggestions. 5/ [significance] Motivation for the newly introduced neighbouring relation definitions Priv1 and Priv2 currently only appears in Appendix B, but I think a summary would deserve to appear in the main paper (space can definitely be gained e.g. in the final Empirical evaluation section), so that the significance of targeting these new privacy settings can be better assessed. 6/ [quality] Is the code for reproducing experiments (available in the supplementary material upload) planned to be released to the public? STRENGTHS: [+] LRF seems to be an interesting problem to study under DP, and it appears the particular settings considered here had either not been studied before, or the algorithms proposed here perform better. WEAKNESSES: [-] The quality of the writing (and grammar) unfortunately leaves a lot to be desired. I don't think the paper could be published in the exact same form as it has been submitted for review. [-] Being a theory paper first and foremost, a more suitable venue than NIPS could be a journal where the reviewing process scope would be such that the proofs (currenty in the Appendix) are also carefully peer-reviewed. MINOR COMMENTS: (Problem 1) Missing range for the variable gamma. (Definition 1) If the output of M is not discrete, a rigorous definition would only consider measurable sets S, rather than all subsets of the range. (36) The sentence is false as written. (73-74) Please provide context about the problem considered by Dwork et al. and Blum et al. (90) Does Big-O with tilde only hide poly log factors in the variable n, or also in other variables (e.g. poly log k)? (170) What is "Question 3"? (205) Do you mean "Algorithm 1" instead of "Figure 1"? (Figure 1) Using different marker shapes would help plot readability when printed in black and white. ===== Update after author response and reviewer discussion ===== Thank you for the responses to our reviews. I'm happy with the answers to our technical comments, and where appropriate I would like to suggest perhaps incorporating some of these clarifications into the next revision. I don't think the quality of writing was good enough for NIPS in the submitted version of the paper, but I believe this can be fixed in the next revision. Please do have your paper proof-read for style and grammar issues. Assuming these changes are made, I've increased my score from 5 to 7.

Reviewer 3



This is a thorough study on differentially private (DP) low-rank factorisation under almost all possible design decisions such as in which order the matrices are updated or whether the output is produced at the end of all updates or not. The authors propose two new definitions of adjacent matrices which are stricter than the ones used in the literature. Using these, the DP arbitrary update algorithm has comparable time and space complexity to the non-private algorithm. The paper has a clear contribution, but it is too exhaustive and long. It can be organised much better and it would be better if the main text is made more self-contained. The message is not very clear.